# Dynamic control of gene expression by ISGF3 and IRF1 during IFNβ and IFNγ signaling

Aarathy Ravi Sundar Jose Geetha[1,2,6], Katrin Fischer [1,2,6], Olga Babadei [1,2], Georg Smesnik[1,2], Alex Vogt[3], Ekaterini Platanitis[1,2], Mathias Müller [4], Matthias Farlik [5] & Thomas Decker [1,2✉]

## Abstract

**Type I interferons (IFN-I, including IFNβ) and IFNγ produce overlapping, yet clearly distinct immunological activities. Recent data show that the distinctness of global transcriptional responses to the two IFN types is not apparent when comparing their immediate effects. By analyzing nascent transcripts induced by IFN-I or IFNγ over a period of 48 h, we now show that the distinctiveness of the transcriptomes emerges over time and is based on differential employment of the ISGF3 complex as well as of the second-tier transcription factor IRF1. The distinct transcriptional properties of ISGF3 and IRF1 correspond with a largely diverse nuclear protein interactome. Mechanistically, we describe the specific input of ISGF3 and IRF1 into enhancer activation and the regulation of chromatin accessibility at interferon-stimulated genes (ISG). We further report differences between the IFN types in altering RNA polymerase II pausing at ISG 5′ ends. Our data provide insight how transcriptional regulators create immunological identities of IFN-I and IFNγ.**

**Keywords** Macrophage; Interferons; STAT; IRF; Transcription
**Subject Categories** Chromatin, Transcription & Genomics; Immunology

## Introduction

Interferons (IFNs) are major determinants of cell-autonomous immunity as first reported for their ability to interfere with viral replication (Isaacs and Lindenmann, 1957). Type I IFN (IFN-I; here represented by IFNβ) are particularly important for innate immunity against viral infections. The main function of IFNγ, the type II IFN, is to enhance immunity to nonviral pathogens by activating macrophages (MacMicking, 2012; McNab et al, 2015; Schneider et al, 2014). Despite their distinct activities in mammalian immune responses, IFNs share a set of common attributes, such as the ability to establish the antiviral state or the enhancement of inflammation.

The activities of IFNs require profound transcriptome changes. The deployment of different signal transducer and activator of transcription (STAT) complexes is held responsible for different immunological effects of IFN-I and IFNγ. IFN-I receptor-associated Janus kinases JAK1 and TYK2 phosphorylate STAT1 and STAT2, causing their heterodimerization and translocation to the nucleus. Interferon regulatory factor (IRF) 9 assembles with the STAT1/2 complex at promoter interferon-stimulated response elements (ISRE), forming the interferon-stimulated gene factor 3 (ISGF3). A minor fraction of STAT1 homodimers is formed in IFN-I-treated cells, but ISGF3 predominates transcriptional activation of interferon-stimulated genes (ISGs) (Ivashkiv and Donlin, 2014; Schindler et al, 2007; Kessler et al, 1990). IFNγ on the other hand activates receptor-bound JAK1 and JAK2 kinases to phosphorylate STAT1, leading to homodimerization and formation of the gamma-interferon activated factor (GAF). GAF stimulates ISG transcription via association with the gamma-interferon-activated site (GAS) (De Weerd and Nguyen, 2012; Decker et al, 1997). The JAK-STAT paradigm of IFN signaling thus posits that ISGF3 dominates transcriptional responses to IFN-I, whereas GAF is critical for the generation of an IFNγ-specific transcriptome. However, recent investigations of the early response to IFN-I and IFNγ cast some doubt on the exclusive assignment of the ISGF3 complex to IFN-I signaling. A large fraction of ISG promoters was associated with ISGF3 following IFNγ receptor engagement, and the transcriptomes of macrophages treated for a brief period with IFNβ or IFNγ were remarkably similar and sensitive to *Irf9* deletion (Platanitis et al, 2019, 2022). These findings prompted us to revisit the factors responsible for the partitioning of IFNβ and IFNγ-induced transcriptomes and their propagation of diverse biological responses.

Transcription factor IRF1 binds to ISRE sequences and contributes to the IFN-induced transcriptional response. The *Irf1* gene is induced during the primary response to both IFN types via association of STAT1 homodimers to a GAS element of its proximal promoter (Pine et al, 1994). IRF1 generates a secondary wave of ISG transcription by cooperating with transcription factors such as GAF or IRF8 (Langlais et al, 2016; Ramsauer et al, 2007; Michalska et al, 2018). Thus, the IRF1 regulome might be important for an increasing divergence between the IFNβ- and

[1]Max Perutz Labs, Vienna Biocenter Campus (VBC), Vienna 1030, Austria. [2]University of Vienna, Center for Molecular Biology, Department of Microbiology, Immunobiology and Genetics, Vienna 1030, Austria. [3]Evotec, Hamburg, Germany. [4]Institute of Animal Breeding and Genetics, University of Veterinary Medicine Vienna, Vienna 1210, Austria. [5]Department of Dermatology, Medical University of Vienna, Vienna 1090, Austria. [6]These authors contributed equally: Aarathy Ravi Sundar Jose Geetha, Katrin Fischer. ✉E-mail: thomas.decker@univie.ac.at

IFNγ-induced transcriptomes at delayed stages of the transcriptional response. However, this assumption has not been rigorously tested, and the gene sets affected by the lack of IRF1 in IFN-I-versus IFNγ-induced transcription have not been globally defined.

Additional contributions to the transcriptome divergence produced by IFN-I or IFNγ signaling may originate from noncanonical versions of the ISGF3 complex such as STAT2-IRF9 which compensates some of the ISGF3 activity in cells lacking STAT1 (Abdul-Sater et al, 2015; Blaszczyk et al, 2015; Majoros et al, 2016; Mariani et al, 2019; Nan et al, 2018; Platanitis et al, 2019; Majoros et al, 2017). Moreover, studies in non-hematopoietic cells support the concept of unphosphorylated U-ISGF3 complexes and their activity in delayed stages of the IFN response (Cheon and Stark, 2009; Sung et al, 2015; Cheon et al, 2013).

Based on the current state of the literature and particularly the surprising role of ISGF3 in the early transcriptional response to IFNγ, our study investigates how IFN-induced transcriptomes arrive from similar states early after treatment at clearly divergent states that produce the striking biological differences of type I IFN versus IFNγ-treated macrophages in the innate immune response. We used nascent transcript sequencing to determine how ISG expression is temporally controlled by the ISGF3 complex or IRF1. We further show that cooperativity between STAT1/ISGF3 and IRF1 correlates with different nuclear interactomes. Finally, we present evidence that in addition to transcription factor recruitment, temporal control of ISG expression includes enhancer activation, dynamic changes of chromatin accessibility and control of RNA pol II pausing. Together, these factors cause continuous segregation of genes and gene groups induced by the two IFN types that are compatible with IFNβ being a better inducer of typical antiviral genes and with the higher potency of IFNγ as a macrophage-activating cytokine.

# Results

## IFNβ and IFNγ-driven transcriptional responses diverge over time

To explore the temporal control of ISG transcription, we performed nascent transcript sequencing in murine bone marrow-derived macrophages (BMDMs) stimulated with IFNβ or IFNγ for up to 48 h (Fig. 1A). In agreement with earlier studies, this treatment duration did not affect the viability of BMDM (Stockinger et al, 2002). Consistent with published data (Platanitis et al, 2022; Liu et al, 2012), early IFNβ and IFNγ-induced transcriptomes were similar, and their similarity decreased over time (Figs. 1B,C and EV1A). The IFNβ-specific transcriptional response was most pronounced after 4 h, while IFNγ produced the strongest increase during later stages (24 and 48 h). Hierarchical clustering of the top 1000 differentially expressed genes in each timepoint identified 11 clusters, defined by similar transcriptional trends over time and treatment (Fig. 1D; Dataset EV1). Representative genes showing different temporal patterns of transcription between the two IFN types are shown in Fig. EV1B. Limitation to the top 1000 genes reflects the low inducibility scores of any genes beyond this point which would result in a diluted representation of the main trends of inducibility.

Transcription of most genes forming clusters 1 and 2 peaks transiently after stimulation with both IFN types; however, the

magnitude of transcriptional induction by IFNβ was higher. Despite its gradual reduction, transcription was maintained above steady state up to 48 h. These clusters reflect the large transcriptional overlap between IFNβ and IFNγ during the early response. Genes in clusters 4 and 9 were similarly induced rapidly by both IFN types, but the expression was sustained specifically by IFNγ, particularly for cluster 9. Further differences between the IFN types were noted for clusters 3 and 10, with IFNβ causing delayed induction in Cluster 3 and IFNγ in Cluster 10. None of the clusters of differentially induced genes contained IFN genes, suggesting that feed-forward loops through induced IFN synthesis are not a factor in sustained ISG responses. In addition to induced gene expression, we observed clusters showing transient (Cluster 5, 7, and 11), delayed (Cluster 6) or sustained (Cluster 8) gene repression upon IFNβ and/or IFNγ stimulation.

Gene ontology analysis identified the major functional categories for each cluster (Figs. 1E and EV1C). Early genes in clusters 1, 2, 4, and 9 showed enrichment in gene functions linked to immune response or interferon signaling while clusters with delayed induction showed enrichment in genes involved in processes such as lipid localization, cell to cell adhesion (cluster 3) and regulation of proteolysis and peptidase activity (cluster 10). Clusters with repressed genes showed enrichment for genes belonging to DNA replication/cell division, metabolic and cytoskeletal reorganization and also other gene categories unrelated to immune responses. The ISG core, including the majority of typical antiviral effector genes (Mostafavi et al, 2016), was mainly found in clusters 1 and 2 (86 out of 95) and a few in cluster 9 (9 out of 95) (Fig. EV1D,E).

Taken together, this kinetic analysis of transcriptional responses to both IFN types demonstrated a strong overlap during the early stages, and increasing divergence during later stages, caused by maintenance of ongoing and/or induction of de novo gene expression.

## Chromatin accessibility and Pol II pausing contribute to the temporal control of ISG transcription

Increases in the accessibility of regulatory elements allow for interactions of transcription factors with promoters and enhancers as well as their cooperative enhancement of transcriptional responses (Maniatis et al, 1987; Smale and Kadonaga, 2003). Previous data showed that the early response to IFN includes profound changes of accessibility at regulatory regions of ISG (Platanitis et al, 2022). Hence, we determined whether waning ISG expression towards termination is similarly accompanied by chromatin accessibility changes. ATAC-Seq in BMDMs stimulated with IFNβ or IFNγ for either a short (1.5 h) or long (48 h) period revealed a striking similarity between transcription and the general trends of chromatin accessibility at regions up to 1000 bp upstream of the TSS (Fig. 2A) and correlation plots of IFN-induced effects on transcription and chromatin accessibility (Appendix Fig. S1). The data suggest continuous chromatin remodeling during the transcriptional response. Gbp2 and Slfn1 represent genes responding to both type I IFN and IFNγ in clusters 9 and 2, respectively. Both belong to the class of genes showing good agreement between the kinetics of chromatin accessibility changes and the transcriptional profile after IFN treatment (Fig. 2B). In contrast, promoter accessibility of the *Dennd6b* ISG from cluster 2, preexists and is

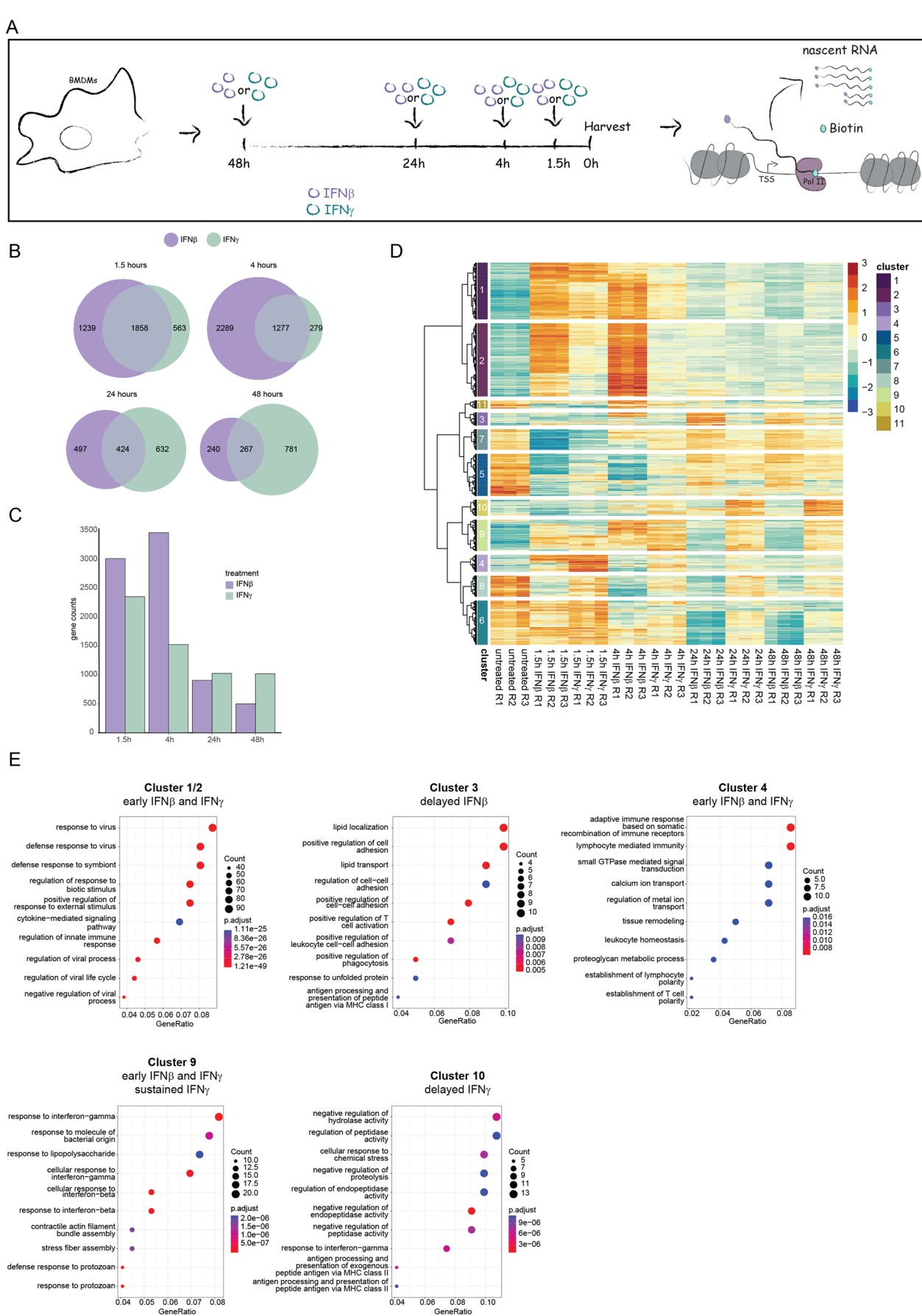

**Figure 1.** IFNβ and IFNγ-driven transcriptional responses diverge over time.

(A) Schematic illustration of the experimental setup used for the PRO-Seq experiment. (B–E) Bone marrow-derived macrophages (BMDM) in three independent replicates were treated with either IFNβ or IFNγ for 48, 24, 4, or 1.5 h or left unstimulated before harvest for nuclear run-on (PRO-Seq). Data were derived from the PRO-Seq analysis. Venn diagram (B)/Bar graph (C) showing the number of upregulated genes in IFNβ and IFNγ treated versus untreated cells at indicated timepoints. The overlap in (B) represents genes that are upregulated by both IFN types. (D) Heatmap showing hierarchical clustering using a pool of the top 1000 genes that are significantly expressed (absolute log2 fold change (log2FC) >=1, adjusted P value (Padj) <0.01) in each timepoint in comparison to the untreated condition. Eleven clusters were defined. (E) Gene ontology of genes belonging to Cluster 1 and 2, 3, 4, 9, and 10 was analyzed using overrepresentation analysis in ClusterProfiler.

maintained independently of interferon treatment. The gray lines representing individual gene promoters in Fig. 2A reflect this variability.

RNA pol II (Pol II) pausing is a regulatory step controlling transcription of mammalian genes (Guo and Price, 2013; Min et al, 2011). We used our PRO-seq data to determine whether Pol II pausing is more generally involved in ISG regulation. Read densities were quantified in promoter-proximal regions versus gene bodies and the respective pausing index was calculated. The log-transformed ratio of pausing indices during each treatment to that of the untreated control per cluster is visualized for both IFN types. The magnitude of index changes is in agreement with findings by others (Steinparzer et al, 2019; Mostafavi et al, 2016) and in agreement with maximal pausing changes achieved by depletion of the pausing factor NELF (Gilchrist et al, 2012). Surprisingly, changes of Pol II pausing were more pronounced during IFNβ signaling (Fig. 2C). Pausing indices changed significantly in most clusters when transcription was strongly induced or repressed during IFNβ stimulation (Dataset EV2). In contrast, Pol II pausing does not seem to have a pronounced effect on transcription during IFNγ signaling, not excluding that it contributes to the regulation of a minority of genes (Steinparzer et al, 2019; Platanitis et al, 2022).

Taken together, the data suggest chromatin accessibility and RNA Pol II pausing as factors controlling the dynamics of ISG transcription. However, the extent to which Pol II pausing contributes to differences between the IFN-I and IFNγ-induced transcriptomes remains to be further investigated.

## ISGF3 and IRF1 exert distinct, gene cluster-specific control of transcription

Next, we investigated how ISGF3 and IRF1 shape immediate, delayed, and sustained profiles of IFNβ- and IFNγ-induced gene expression. IRF1 amounts in IFNγ-treated BMDMs (Fig. 3A) or RAW 264.7 cells (Fig. EV2A) were high and persistent, in agreement with *Irf1* transcription in BMDM (Fig. EV2B). In contrast, IRF1 declined rapidly after IFNβ stimulation. This is in line with previous publications (Michalska et al, 2018). Principal component analysis (PCA) of nascent transcript sequencing (Fig. 3B) showed a larger similarity of IRF1-deficient samples to wild-type than was observed with IRF9-deficient samples. We computed the impact of IRF9 and IRF1 on genes in the previously defined clusters across all timepoints and visualized results as trend lines showing z-score-normalized read counts (Fig. 3C). The observations are summarized in Appendix Table S1. In keeping with our interest in IFN-induced genes, subsequent analyses are focused on clusters 1, 2, 3, 4, 9, and 10.

Genes in clusters 1 and 2 showed no effect of IRF1 deficiency but constant dependence on the ISGF3 complex. Contrasting IFNβ

treatment, cluster 1 gene expression was reduced, but not abrogated by the lack of IRF9 in IFNγ-treated cells, suggesting a potentially larger contribution of STAT1 homodimers. To investigate whether IRF9 dependence reflected the canonical ISGF3 complex, we performed site-directed chromatin immunoprecipitation (ChIP) using antibodies against IRF9, STAT1 and STAT2 (Fig. EV2C). ChIP data confirmed the IFNβ-induced binding of ISGF3 components to the promoter of *Mx2*, a representative cluster 2 gene up to 24 h while binding in response to IFNγ was more transient, consistent with the more robust response of cluster 2 genes to type I IFN. Ratios between STAT1 and STAT2 binding were unchanged between the early and delayed response to IFNβ, suggesting constant employment of the canonical ISGF3 complex (Fig. EV2D). Treatment of BMDMs with the Janus kinase inhibitor ruxolitinib as well as site-directed ChIP using antibodies against phosphorylated and total STAT proteins, produced no evidence for U-STAT complexes maintaining delayed levels of transcription. Gene expression required constant JAK activity (Fig. EV2E). The data suggest a strong involvement of the canonical ISGF3 complex in early induction and maintenance of cluster 1 and 2 gene transcription for both IFN types.

Transcriptional induction of the majority of clusters 3 and 10 genes was delayed. Cluster 3 genes responded better to IFNβ and required IRF9 for both IFNβ and IFNγ inducibility. A minor effect of IRF1 deficiency was observed for IFNγ, but not for IFNβ. In contrast, cluster 10 genes responded much better to IFNγ, but required IRF1 for both IFN types.

The majority of cluster 9 genes showed transient IFNβ-, but sustained IFNγ-induced expression. Transcription in response to IFNβ required both IRF9 and IRF1, but IFNγ-induced gene expression showed constant IRF1 dependence. Binding of IRF1 to the *Gbp2* promoter (cluster 9) showed strong correlation to the trend of transcription, being transient during IFNβ and sustained during IFNγ stimulation (Fig. EV2G). Delayed effects of ISGF3 and IRF1 might be caused by their sustained activity or, alternatively, result from induced synthesis of additional transcription factors. To distinguish between these possibilities, we determined the binding of IRF9 and IRF1 to promoters of clusters 2, 3, 9, and 10 genes (Fig. 3D). Contrasting the strong binding of IRF9 and IRF1 to the *Mx2* (cluster 2) and *Gbp2* (cluster 9) promoters, respectively, we saw no evidence for direct binding to *Lrp11* (cluster 3) and *Ptgs2* (cluster 10). This suggests both direct and indirect effects of ISGF3 and IRF1 in maintaining transcription. Among clusters of induced genes cluster 4 was peculiar in not showing clear effects of IRF9 or IRF1 deficiency. Possibly, secreted factors responding to IFN treatment and ISGF3/IRF1 activity cause higher-order transcriptional waves via induction of different transcription factors. Importantly, genes associated with pattern recognition or effector mechanisms of activated macrophages showed delayed or sustained responsiveness, a stronger response to IFNγ and transcriptional dependence on IRF1 (Fig. EV3).

A

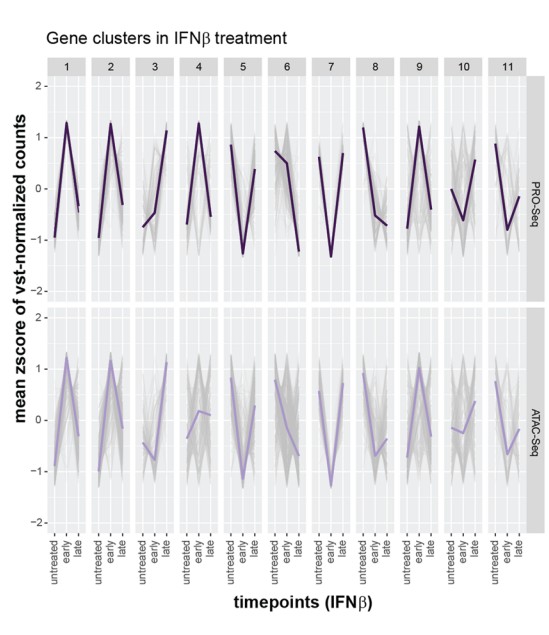

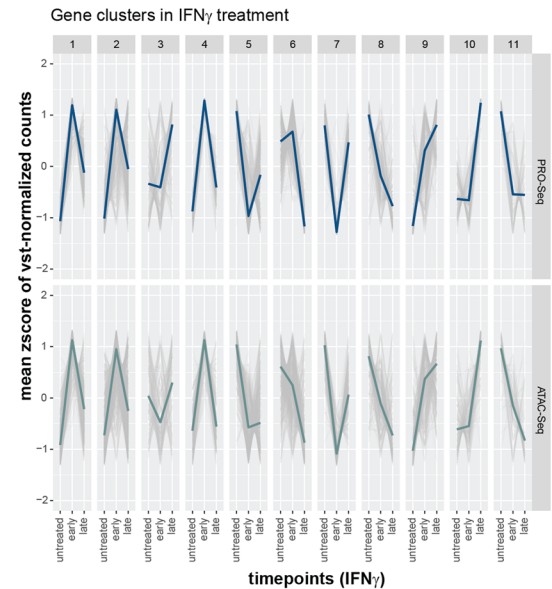

B

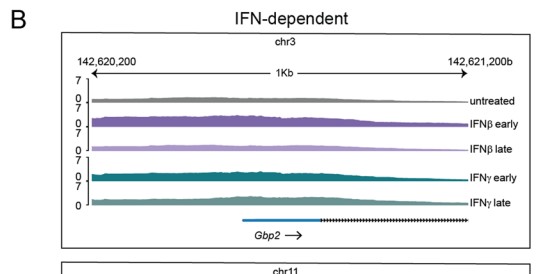

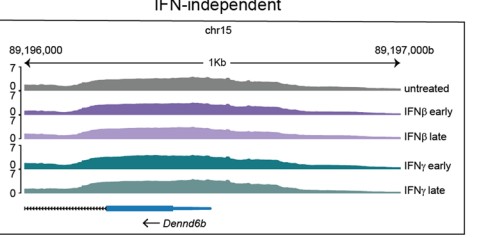

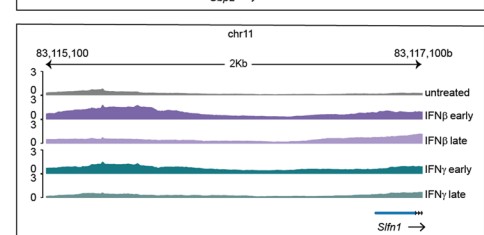

C

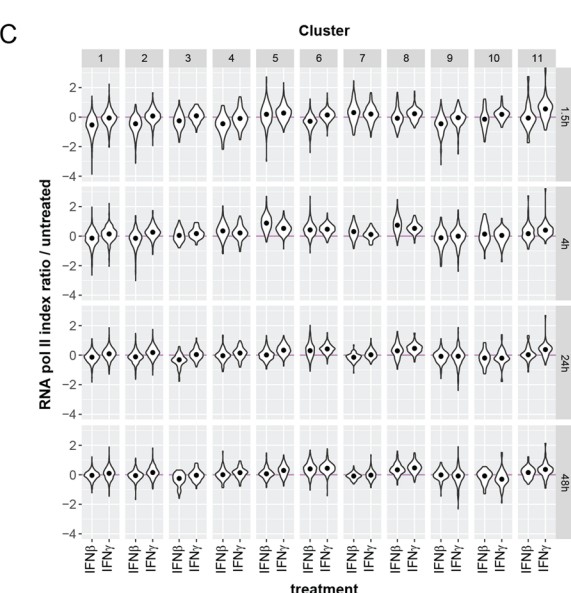

Figure 2.   Chromatin accessibility and Pol II pausing are involved in regulating transcriptional dynamics during IFN signaling.

(A) Trend lines of the mean z-score of vst-normalized PRO-Seq and ATAC-Seq counts (gray) representing three independent replicates of IFN-treated BMDMs separate by cluster and treatment (IFNβ or IFNγ). Early = 1.5 h for PRO-Seq, 2 h for ATAC-Seq; Late = 48 h; Single-colored lines represent the median across all genes. (B) Browser tracks of ATAC-Seq-derived samples showing chromatin accessibility of *Gbp2* and *Slfn1* (IFN-dependent genes) and *Dennd6b* (IFN-independent gene). Early = 2 h; Late = 48 h. (C) Log ratio of Pol II pausing indices during IFNβ and IFNγ stimulation at each timepoint to those of the untreated condition. Data represent triplicate samples from BMDM, as described in the legend of Fig. 1. Read counts used for calculating pausing indices were derived from the PRO-Seq.

In conclusion, the data show the importance of the ISGF3 complex for responses to both IFN types. IFNγ inducibility of cluster 2 genes is entirely ISGF3-dependent and lesser effects of IRF9 deficiency were noted in clusters 1 and 3. ISGF3 deficiency reduced the responsiveness of all induced gene clusters to IFNβ, except clusters 4 and 10. IRF1 contributed to the IFNβ inducibility of genes in clusters 9 and 10. IRF1's main impact consisted of sustaining IFNγ-induced genes, thus contributing largely to the gradual diversification of IFNβ and IFNγ-induced transcriptomes.

## IRF9 and IRF1 regulate the activation of a subset of ISG enhancers

The activation of enhancers and promoter-proximal regulatory elements is associated with their bidirectional transcription and the production of enhancer RNAs (eRNAs) (Andersson et al, 2014; Tyssowski et al, 2018). We sought to determine whether the activation of regulatory elements and production of eRNAs reflected the transcription control of ISG observed in Fig. 3, particularly whether clusters 3 and 10 contained ISG not requiring the binding of ISGF3 and IRF1. To this end, we identified active transcriptional regulatory elements (TREs) using our PRO-Seq data (Wang et al, 2019; Danko et al, 2015). We estimated the eRNA read counts in TREs within the 50 kb range from the annotated TSS for genes of interest and, in addition, followed the trend of transcription in the respective clusters in the wild-type situation as described (Chu et al, 2018). To avoid transcriptional interference due to the loss of ISGF3 or IRF1 (Fig. 4A), intragenic regions were excluded. We visualized z-score normalized, vst-transformed eRNA read counts from enhancer regions in wild-type, $Irf9^{-/-}$ and/ or $Irf1^{-/-}$ cells in the gene expression clusters defined above (Fig. 4B).

The decrease of eRNA signals resulting from the loss of IRF9 in enhancers regulating clusters 1 and 2 was strong, but it was comparatively moderate at cluster 3, 9, and 10 regulatory elements. This strengthens our notion that additional transcription factors may regulate the expression of the corresponding ISG. To cross-validate our results, we intersected ChIP-Seq data from published (Platanitis et al, 2019; Langlais et al, 2016) as well as a newly generated dataset with the identified enhancers. Browser tracks showing eRNA-based positions of regulatory elements and the corresponding association with IRF9 or IRF1 for the *Mx2* (cluster 2) and *Gbp4* (cluster 9) genes are presented in Fig. 4C. A large number of enhancers were bound by IRF9 in clusters 1 and 2 (5006 out of 8997; 56%) and cluster 3 (184 out of 486; 38%). IRF1 on the other hand showed binding at a minor number of enhancers in cluster 9 (518 out of 2245; 23%) and cluster 10 (181 out of 787; 23%), which might reflect the involvement of additional transcription factors (Fig. 4D,E). To further strengthen this assumption, we performed motif enrichment analysis using the respective enhancer regions, including intragenic regions for each cluster. In addition to STAT and IRF binding sites, binding motifs of transcription factors

belonging to the ETS and bZIP family were significantly enriched. The ranking of these motifs differed between the clusters (Dataset EV3). Motifs representing ISRE sequences were ranked highest in clusters 1, 2, 3, and 9. In cluster 10, the bZIP factor motifs were ranked highest. With exception of AP2 motifs in clusters 3 and 10, no other motifs showed specificity for clusters with delayed or sustained genes. Moreover, bZIP factor genes such as *Atf3* (log2fc IFNβ = 2.5; log2fc IFNγ = 1.7), *c-Jun* (log2fc IFNβ = 1.3; log2fc IFNγ = 0.96) and *Junb* (log2fc IFNβ = 1.1; log2fc IFNγ = 0.62) were rapidly upregulated by IFN stimulation, supporting our notion that secondary transcription factors other than IRF1 may contribute to delayed and/or sustained ISG transcription.

## IRF1 regulates chromatin accessibility of a specific subset of genes

IRF1 contributes to Pol II recruitment to the TSS of ISG (Ramsauer et al, 2007), but its role in determining the accessibility of its target promoters in IFN-stimulated macrophages has not been examined. To this end, promoter accessibility of genes requiring IRF9 or IRF1 for rapid induction was determined following the workflow outlined in Fig. 5A. Figure 5B,C depicts IRF1 and IRF9-dependent genes, respectively, with purple dots. Genes showing IRF1 or IRF9-dependent accessibility changes in addition to requiring the transcription factors for expression are marked with yellow dots. In accordance with our previous results (Platanitis et al, 2022), IRF9 deficiency resulted in profound effects on both gene expression and promoter accessibility after both IFNβ and IFNγ treatment (Fig. 5C). IRF1 on the other hand regulated transcription of smaller subsets of genes after IFN treatment (Fig. 5B) or in resting cells (Fig. EV4). Regulation of gene expression and chromatin opening did not necessarily coincide as shown for the *Ifi44* and *Gbp2* genes (Figs. 5D and EV3). Expression required IRF1 for both, but accessibility of the *Ifi44* regulatory region depended upon IRF1, whereas that of *Gbp2* acquired an open conformation independently of IRF1. The data show that compared to ISGF3, IRF1 affects chromatin accessibility of a smaller gene subset which overlaps only partially with and is considerably smaller than the pool of genes requiring IRF1 for transcription. The impact of IRF1 on basal, ISRE-driven ISG expression agrees with a recent report showing a contribution of IRF1 to the intrinsic antiviral resistance of a human hepatocyte line (Ikeda et al, 1998).

## STAT1 and IRF1 interact with proteins relevant for transcription, histone modification, and chromatin remodeling

To gain further insight into molecular interactions involving STATs or IRF1, we determined their interactomes. STAT1 was chosen for these experiments because it represents both the transcriptionally active homodimers as well as the ISGF3 complex. Published evidence (Ramsauer et al, 2007) and the data shown above suggest that ISG

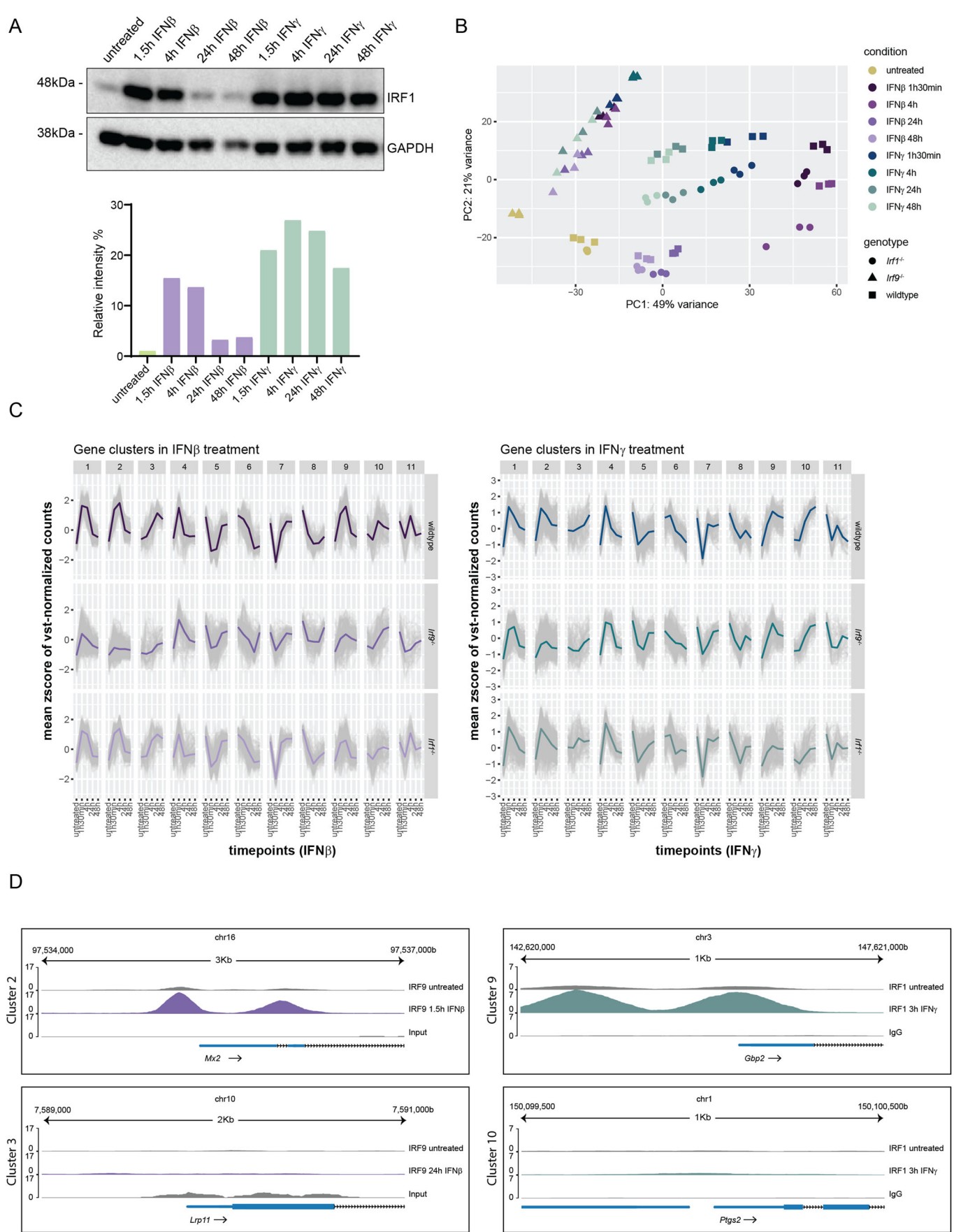

**Figure 3.   ISGF3 complex and IRF1 have distinct, cluster-specific roles in regulating transcription.**

(A) Bone marrow-derived macrophages (BMDM) were treated with IFNβ or IFNγ for either 1.5, 4, 24, or 48 h and protein levels of IRF1 and GAPDH in whole-cell lysates were measured using western blotting. GAPDH was used as a loading control. The representative blot (of three independent replicates) was quantified using Image Lab and shown in the lower panel. Relative intensities of the bands were normalized to their corresponding GAPDH levels. (B, C) Wild-type, *Irf9*<sup>−/−</sup> and *Irf1*<sup>−/−</sup> BMDMs were treated with IFNβ or IFNγ for 1.5, 4, 24, or 48 h and nascent RNA transcripts derived from the PRO-Seq were analyzed. Principal component analysis (PCA) using 500 most variable genes in DESeq2 (B). Trend lines representing the mean z-score of vst-normalized counts (gray) across genotype and treatment conditions separated by clusters. Single-colored lines represent the median (C). (D) Browser tracks of ChIP-Seq-derived samples showing binding of IRF9 at the promoter of *Mx2* (untreated versus 1.5 h IFNβ) and *Lrp11* (untreated versus 24 h IFNβ) and IRF1 binding at the promoter of *Gbp2* (untreated versus 3 h IFNγ) and *Ptgs2* (untreated versus 3 h IFNγ). Published IRF1 ChIP-Seq data (Langlais et al, 2016) were re-analyzed. Source data are available online for this figure.

promoters under IRF1 control are subject to cooperativity with both STAT1 dimers and the ISGF3 complex. While ISGF3 and IRF1 may bind to the same regulatory element (ISRE subsets that form IRF-E), cooperation between STAT1 dimers and IRF1 occurs via binding to spatially separated elements (IRF-E versus GAS) (Decker et al, 1997; Tanaka et al, 1993; Fujii et al, 1999).

To examine whether both STAT1 and IRF1 are present in the same protein complexes, we performed co-immunoprecipitation studies with STAT1 and IRF1 antibodies in BMDMs (Fig. EV5A,B) and RAW 264.7 cells (Fig. EV5C,D), failing to observe interaction between the two proteins. To examine STAT1/IRF1 proximity or transient interaction and to address their interplay with histone modifiers and the transcriptional machinery (Bonev and Cavalli, 2016; Guo and Price, 2013; Maniatis et al, 1987; Min et al, 2011; Schoenfelder and Fraser, 2019; Smale and Kadonaga, 2003), we performed proximity labeling. In a recent study with cells treated for a brief period with IFNs, the prototypical BioID technology failed to report an interaction between ISGF3 subunits and IRF1 in whole-cell lysates (Platanitis et al, 2019). Here, we modified the experimental system in several aspects to increase sensitivity. First, the much shorter biotinylation period of the TurboID system (10 min) reduces the experimental background. Second, we enriched for nuclear interactions by analyzing nuclear extracts. Third, increasing IFN treatment to 3 h allowed to interrogate secondary transcriptional effects. Clones of RAW 264.7 cells were engineered to express doxycycline (Dox)-inducible, V5-tagged STAT1 or IRF1 fused to the modified biotin ligase BirA* in their respective knockouts (Appendix Fig. S2A,B). Cells expressing a V5-tagged nuclear localization signal (NLS) fused to BirA* were used for normalization.

Mass spectrometry of biotinylated proteins identified numerous STAT1 interactors (184) and comparatively few of IRF1 (37) in resting and/or IFN-treated cells (Fig. 6A,B). The larger number of interactors in the STAT1 pull-down may reflect a real difference but could also be influenced by a higher signal-to-noise ratio, hence larger number of statistically significant proximities. Importantly, the analysis revealed STAT1-BirA* and IRF1-BirA* self-biotinylation and demonstrated STAT1 proximity to the ISGF3 subunits STAT2 and IRF9 as well as to IRF1. Interactors of the IRF1 bait included STAT1, and the interaction increased after IFN treatment, but it did not conform to our statistical cutoff. The reduced proximity of IRF1 to the STAT1 bait might result from steric constraints of the tagged proteins due to the limitation of the 10 nm distance reached by the biotin ligase.

To maximize for interactions of our bait proteins in the transcriptionally active state, we excluded STAT1 interactors found exclusively at steady state. Accordingly, we identified 127 STAT1-specific and 30 IRF1-specific interactors, 7 were in common

between both baits (Fig. 6C). The heatmap visualizes the list of all selected interactors grouped by functional annotation (Fig. 6D). Specifically, interactors included proteins representing transcription, histone modification, cell division, DNA replication/methylation and regulation of small GTPase as well as several ISGs in the STAT1-BirA* analysis, while IRF1 interactors represented transcription, ISGs and chromatin remodeling. This was confirmed by gene ontology analysis (Dataset EV4). Consistent with our previous study (Ramsauer et al, 2007), STAT1 interactors included subunits of histone and nucleosome-modifying complexes, particularly of the NuA4 (Tip60) complex (e.g., KAT5 = TIP60, EP400, TRRAP), a histone acetyltransferase complex responsible for acetylation of histone H4 and H2A (Fig. 6E) (Judes et al, 2015; Dhar et al, 2017; Keogh et al, 2006; Babiarz et al, 2006; Doyon et al, 2004). In addition, STAT1 interactors included subunits of the histone-acetylating SAGA and ATAC (TRRAP, YEATS2) as well as SRCAP/SWR1 (e.g., VPS72 = YL1) complexes which are involved in the exchange of the histone variant H2AZ. Many nuclear interactors of STAT1 are well-documented ISG, some of which localize to nuclear sub-compartments such as members of the speckled family, thought to confer transcriptional regulation (Fraschilla and Jeffrey, 2020). Their link to the transcriptional activity of STAT1 is unclear. Similarly, proximity to proteins involved in RNA processing such as decapping (Edc3, Edc4), splicing (Gemin5), nuclear spindle assembly (Haus complex) or ADP ribosylation (Parp9, Parp12, Parp14) is of unclear functional consequences. Of interest, PARP14 is linked to the nuclear accumulation of ISG (Caprara et al, 2018). Several interactors of STAT1 were found in common with the recently published BioID screens of STAT1 (IFI203, STAT2, Al607873, TRIM14, and AHNAK) and IRF9 (PARP14, STAT2, Al607873, EP400, and c/EBPδ,) using whole-cell lysates (Platanitis et al, 2019). IRF1 interactors showed enrichment for components of the PBAF complex, which plays a role in chromatin remodeling (Fig. 6F) (Sima et al, 2019), in line with the importance of IRF1 for chromatin accessibility (Fig. 5B,C). Interestingly, the ISGF3 component STAT2 was also found in proximity to IRF1-BirA*.

To determine whether transcription factors showing proximity to STAT1 and/or IRF1 qualify as potential secondary or tertiary drivers of ISG expression in the sustained and/or delayed clusters (Cluster 3, 9, and 10), we analyzed the IFN-induced transcriptional responses of genes encoding interactors of STAT1 and/or IRF1 (Fig. 6D; labeled in bold). Transcription factor ATF3 (log2fc IFNβ = 2.5, log2fc IFNγ = 1.7), a common interactor of both STAT1-BirA* and IRF1-BirA*, was among rapidly induced genes. Binding sites for ATF3 were among the enriched motifs shown in delayed and sustained ISG (Dataset EV3).

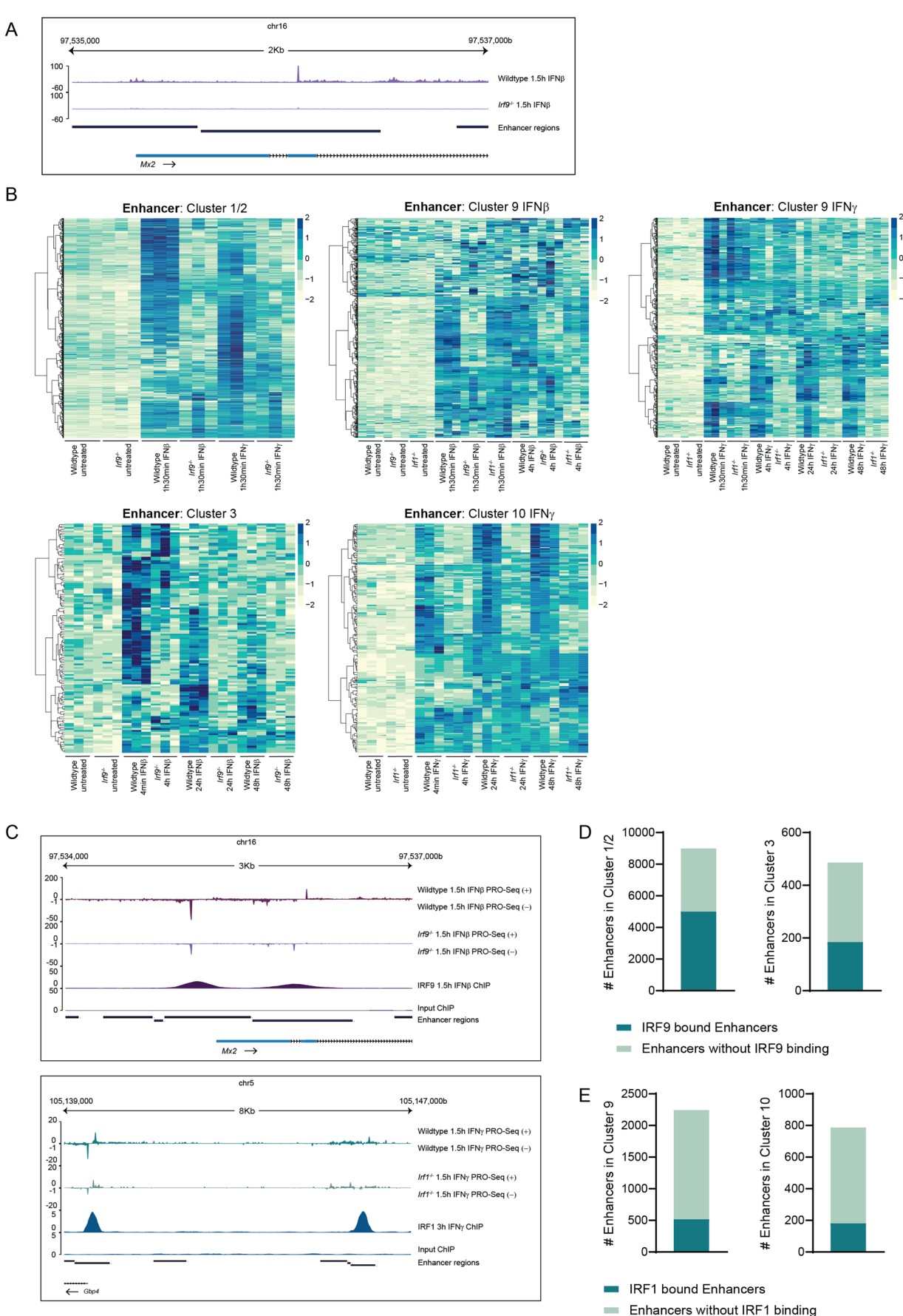

**Figure 4. IRF9 and IRF1 regulate the activation of a subset of ISG enhancers.**

(A) Browser tracks representing nascent transcripts (sense strand) in wild-type and *Irf9*−/− BMDMs at dREG-derived enhancer regions at *Mx2* loci during 1.5 h of IFNβ stimulation. (B) Heatmap of vst-normalized read counts (DESeq2) across indicated genotype and treatment conditions in enhancers of respective clusters. The enhancer regions represented here were identified using dREG transcriptional regulatory element peak calling, and further filtered by exclusion of gene-regions using bedtools 2.30.0. (C) Browser tracks representing nascent RNA transcripts (derived from PRO-Seq) in sense and anti-sense strands of wild-type and *Irf9*−/− (top panel) or *Irf1*−/− (bottom panel) BMDMs, together with browser tracks of published ChIP-Seq data for IRF9 and re-analyzed published ChIP-Seq data for IRF1 (Langlais et al, 2016) at enhancer regions of *Mx2* and *Gbp4*, respectively. (D, E) Bar plot representing total number of enhancers identified in triplicate samples for genes in the indicated clusters (derived from PRO-Seq) and number of enhancers that are in addition bound by IRF9 (D) and IRF1 (E) colored in dark green derived from re-analyzed published ChIP-Seq data (Platanitis et al, 2019; Langlais et al, 2016) as well as newly generated IRF9 ChIP-Seq data at 24 h.

In conclusion, we identified protein complexes relevant for the establishment of a transcription-permissive chromatin structure as STAT1 and to a lesser extent, IRF1 interactors. The findings are in agreement with a role of STAT1 and IRF1 in the regulation of histone modification and remodeling. Candidate transcription factors for delayed and sustained responsiveness to IFN, such as ATF3, were among IFN-induced immediate–early genes.

## Discussion

Macrophages are major targets of the IFN system. While both IFN types induce many genes encoding intracellular effector proteins or secreted immunoregulators (Schoggins, 2019; Sadler and Williams, 2008; Liu et al, 2012), IFN-I are the more important mediators of antiviral immunity and the polarization toward an antimicrobial or M1 state is mainly a functional attribute of IFNγ (MacMicking, 2012). Mutations abrogating IFNγ responsiveness in both mice and humans provide clear evidence for this (Bastos et al, 2007; Van den Broek et al, 1995; Newport, 1997; Jouanguy et al, 1999).

Analyses of nascent RNA transcripts confirm the similarities of most of the immediate–early genes induced by the two IFN types observed earlier (Platanitis et al, 2022, 2019), while also showing that on average these early transcripts were more abundant after IFN-I stimulation. In line with expectations, ISGF3/IRF9 played a major role in the transcriptional response to type I IFN with the exception of genes in clusters 4 and 10. Whereas the early response of cluster 4 genes might be caused by STAT1 homodimers, this is unlikely for the delayed genes of cluster 10 due to the kinetics of STAT1 homodimer formation. IRF9-independent responsiveness to type I IFN was also observed in transformed fibroblasts and B cells from a human patient with IRF9 deficiency (Hernandez et al, 2018). Intriguingly, this patient suffered from severe pulmonary influenza, but was able to control other respiratory viruses, suggesting ISGF3-independent deployment of antiviral activity.

Divergence of the transcriptomes occurred progressively through IFN type-specific delays of gene induction or maintenance of high transcription rates. For example, expression of the *Gbp* cluster, which contributes resistance to nonviral pathogens (Praefcke, 2018; Tretina et al, 2019), or the M1 polarization marker genes *Cd86* and *Nos2* (Taylor et al, 2005) is sustained after IFNγ, but transient upon IFN-I treatment. Likewise, chemokine genes responding to both IFN types such as *Cxcl9* and *Cxcl10* (Forero et al, 2019; Mundra et al, 2016), show sustained responses only after IFNγ treatment. The antimicrobial *Gbp2b* gene, (Sohrabi et al, 2018), was found to be induced in a delayed manner, specifically in the IFNγ response. In summary, our data suggest that the temporal control of transcription, rather than inducibility per se, plays an

important role in diversifying IFN-I and IFNγ-induced transcriptomes. Consistent with the large-scale binding of the ISGF3 complex on gene promoters during IFNγ stimulation (Platanitis et al, 2019), we show that many ISG and their associated enhancer transcripts require IRF9, further emphasizing ISGF3 regulation of genes induced rapidly by both IFN types. However, delayed waves of transcription require higher-order transcriptional activities, such as the second-tier transcription factor IRF1 (Langlais et al, 2016; Ramsauer et al, 2007). We found IRF1 to play a major role in delayed and sustained IFNγ responses and thus to contribute to the macrophage-activating activity of IFNγ. Our results are in excellent agreement with a recent study showing that IRF1 loss-of-function in humans results in susceptibility to mycobacterial disease, a condition including decreased macrophage activation, whereas IRF1 was largely redundant for antiviral activity (Rosain et al, 2023). In addition to the confirmed role of IRF1 we identified candidate transcription factors which may further contribute to the temporal dynamics of delayed/sustained responses. We recently reported synergistic regulation of ISG by AP-1 and STAT complexes in macrophages concomitantly exposed to stress and IFN (Boccuni et al, 2022). In line with this, AP-1 family transcription factors ATF3 and BATF2 were identified by STAT1 and/or IRF1 proximity labeling. Binding motifs for these transcription factors were enriched in ISG regulatory elements (Gargiulo et al, 2013; Li et al, 2012). Interestingly, ATF3 was identified as a negative regulator of a subset of type I IFN-induced genes in an earlier study (Labzin et al, 2015). This suggests it may diversify IFN-induced transcriptomes by selective suppression. Our data suggest this may involve direct interactions and/or functional cooperation with both STAT1/ISGF3 and IRF1.

Several studies reported the STAT-mediated recruitment of histone-acetyltransferases (HAT) such as EP300 and CREBBP (Zhang et al, 1996; Bhattacharya et al, 1996). At the *Gbp*2 promoter, STATs were responsible for IFN-induced histone acetylation, but IRF1 did not contribute (Ramsauer et al, 2007). In agreement with this, nuclear STAT1 interactors represented complexes involved in histone acetylation and exchange, especially the ones belonging to the NuA4/Tip60 chromatin remodeling complex family (Judes et al, 2015). Interestingly, some versions of these complexes mediate the H2 exchange against the variant H2AZ which has been associated with ISG repression (Au-Yeung and Horvath, 2018). Regarding IRF1, interaction with chromatin remodeling SWI/SNF family members was previously shown (Ren et al, 2015). Consistently, TurboID revealed IRF1 proximity to the SWI/SNF family PBAF complex and ATAC-seq demonstrated that IRF1 increases the accessibility of ISG promoters during steady state as well as IFN stimulation. In summary, the turbo-ID approach corroborates predictions from this or earlier studies about functional

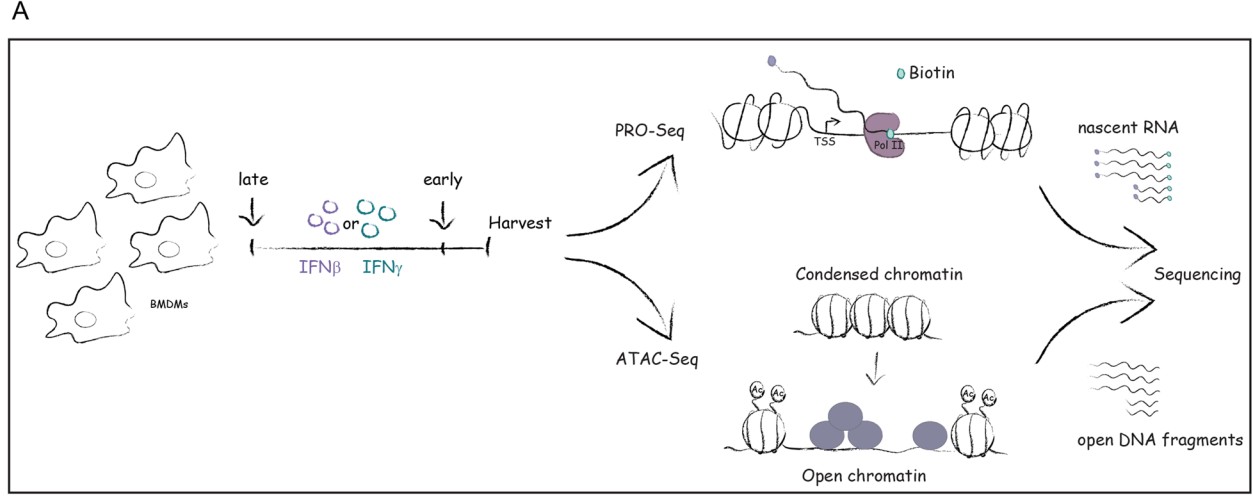

**Figure 5.  IRF1 regulates chromatin accessibility of a specific subset of genes.**

(A) Schematic representation of the treatment conditions and workflow for processing ATAC-Seq samples in comparison to PRO-Seq samples. (B, C) Volcano plots of nascent transcripts in wild-type BMDMs stimulated with IFNβ (left panel) or IFNγ (right panel) for 1.5 h ($n = 3$). Each dot represents a gene. The log2-transformed fold change and $-$log10-transformed *Padj* values are shown on the *x* and *y* axis, respectively. Only genes with significant changes in accessibility within the range of 2000 bp from the TSS were considered. Dark-blue dots represent genes that were significantly upregulated after either IFN treatment (log2FC ≥ 1, Padj ≤0.05) according to PRO-Seq. The violet dots represent genes significantly downregulated in *Irf1*$^{-/-}$ (B) or *Irf9*$^{-/-}$ (C) BMDMs according to PRO-Seq (log2FC ≤ -1, Padj ≤0.05). The yellow dots represent genes which in addition showed a significant decrease in chromatin accessibility (log2FC ≤ 1, Padj ≤0.05) in *Irf1*$^{-/-}$ (B) or *Irf9*$^{-/-}$ (C) BMDMs compared to their wild-type counterpart, either during IFN treatment (1.5 h) or homeostatic condition according to ATAC-seq data. (D) Browser tracks showing chromatin accessibility at representative gene loci (*Ifi44* and *Gbp2*) in wild-type and *Irf1*$^{-/-}$ BMDMs derived from ATAC-Seq.

cooperativity between transcription factors, chromatin-modifying or remodeling complexes and STAT1/ISGF3 or IRF1. The experiments also yielded a number of additional interactors of interest, such as a surprisingly large number of nuclear ISG products. We did not validate these interactions with an independent approach, as co-precipitation-based technologies that do not report transient interactions or proximity would result in a different interactor spectrum. In future studies, it may be of interest to test functional consequences of such interactions with targeted CRISPR screens.

The transcriptional activity of IRF1 and its ability to contribute to chromatin remodeling affected overlapping, but nonidentical sets of genes. This might be explained by IRF1-independent chromatin rearrangements by pioneer factors such as PU.1 (Mayran and Drouin, 2018; Heinz et al, 2010; Mancino et al, 2015). A recent study in a human monocyte cell line reported IRF1 as an accessibility factor for a large number of genes downstream of TLR (Song et al, 2021). The larger number compared to the IFN response (Platanitis et al, 2022) may result from the contribution of the ISGF3 complex in IFN-treated cells, rendering IRF1's activity unnecessary. In addition to showing steady state or immediate effects on promoter accessibility, our data further suggest that the waning of ISG transcription correlates with a gradual decrease of accessibility. This may either indicate that the restoration of nucleosomes impedes transcription initiation or that it is a consequence of reduced transcriptional activity.

IFN alter the 3D arrangement of ISG loci (Platanitis et al, 2022). The lack of direct binding of ISGF3 and IRF1 to a large fraction of bona-fide IFN-responsive enhancers could be partially explained by this phenomenon. Further relevant in this context is the ability of a fraction of ISG promoters to act as Epromoters, i.e., to exert enhancer activity on neighboring ISG (Santiago-Algarra et al, 2021). By and large, the occupancy of ISRE-containing ISG promoters with canonical ISGF3 complexes corresponded to the kinetics of the transcriptional response. This finding disagrees with the idea of STAT2-IRF9 complexes sustaining expression of the investigated ISG as reported for STAT1-deficient cells (Majoros et al, 2017; Abdul-Sater et al, 2015; Majoros et al, 2016). Likewise, the IFN-I response of macrophages showed no evidence of the U-STAT signaling reported for other cell types (Cheon and Stark, 2009; Sung et al, 2015). Our results rather agree with an early report in a human B lymphocyte line suggesting a direct relationship between the levels of tyrosine-phosphorylated STATs and ISG expression (Lee et al, 1997).

Residency periods of RNA pol II downstream of the TSS were found to differ between the IFN types for a subset of ISG. Our data thus assert a role of Pol II pausing in the IFN-induced transcriptional response. Changes in RNA Pol II pausing occurred in response to both IFN types but were more pronounced after

IFN-I treatment. This is in agreement with the general importance of RNA Pol II pausing for rapid responses to environmental changes (Guo and Price, 2013; Min et al, 2011) and with recent evidence for a contribution of Pol II pausing to the regulation of IFNγ-induced transcription in murine fibroblasts (Steinparzer et al, 2019).

In conclusion, our findings confirm the initial similarity and ISGF3 dependence of IFNγ and IFN-I-induced transcriptomes, while also demonstrating their progressive divergence and the emergence of gene sets with distinct transcriptional dynamics. We define a role for IRF1 in transcriptome diversification and suggest that other higher order transcription factors may contribute. Further studies are needed to discern their link to the mechanisms associated with differences in transcriptional induction by IFNγ and IFN-I.

## Methods

### Cell culture and differentiation

Bone marrow-derived macrophages (BMDMs) from C57BL/6 mice of either sex were isolated from femur and tibia by either crushing or flushing the bones followed by filtration and centrifugation at 350 rcf. Cells were differentiated for 9–10 days in Dulbecco's modified Eagle's medium (DMEM) (Sigma-Aldrich) supplemented with 10% fetal calf serum (FCS), 100 units/ml Penicillin and 100 units/ml Streptomycin (Pen/Strep) (Sigma-Aldrich) and recombinant M-CSF (a kind gift from L. Ziegler-Heitbrock, Helmholtz Center, Munich, Germany) on 15-cm Petri dishes. Cells were incubated at 37 °C and 5% $CO_2$ atmosphere. RAW 264.7 cells (ATCC TIB-71) were cultured in DMEM supplemented with 10% FCS and Pen/Strep at 37 °C and 5% $CO_2$ atmosphere. RAW 264.7 were regularly tested for mycoplasma contamination. BMDMs and RAW 264.7 cells were either stimulated with IFNβ (250 IU/ml, PBL Assay Science; Catalog # 12400-1) or IFNγ (10 ng/ml, eBioscience; Catalog # 14-8311-63).

### Genome-editing via CRISPR–Cas9

Wild-type RAW 264.7 cells were edited using a guide RNA for murine *Irf1* (TTAATTCCAACCAAATCCCA (GGG; PAM sequence)) that was designed using CHOPCHOP web tool (Labun et al, 2019). Oligos were ligated into LentiCRISPRv2 vector (Addgene catalog # 52961; a kind gift from Gijs Versteeg lab, Max Perutz Labs, Vienna, Austria) and transduced into RAW 264.7 cells. After transduction, single clones were selected. The guide RNA for murine *Stat1* (GGGGCCATCACATTCACAT (GGG; PAM sequence) was cloned into the lentiviral vector 1358_sgRNA

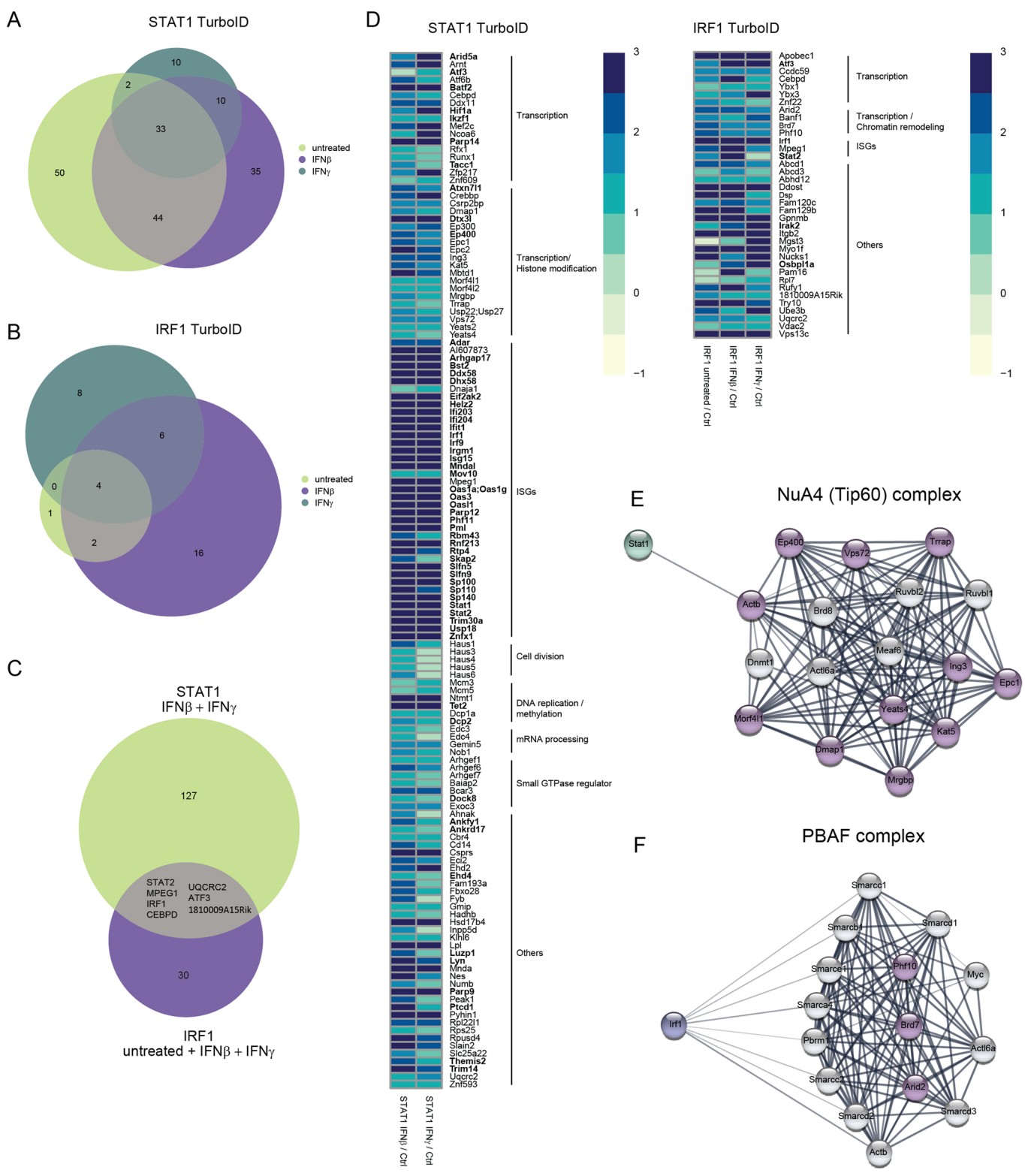

V2 U6-IT-mPgk-iRFP720 (a kind gift from Zuber Lab, IMP, Vienna, Austria) and transduced into Dox-inducible Cas9-GFP RAW 264.7 cells (a kind gift from Gijs Versteeg lab, Max Perutz Labs, Vienna, Austria).

## Jak inhibitor

In total, $1 \times 10^6$ BMDMs were seeded on day 9 of differentiation, treated with IFNβ for 48 h and additionally incubated with 1.5 μM

◄ **Figure 6. STAT1 and IRF1 interact with factors relevant for transcription, histone modification, and chromatin remodeling.**

(A–F) Interactors of STAT1-BirA* and IRF1-BirA* in Stat1$^{-/-}$ and Irf1$^{-/-}$ RAW 264.7 cells, respectively. Dox-treated cells in three independent replicates were either left untreated or stimulated with IFNβ or IFNγ for 3 h ($n = 3$). Interactors were filtered for log2FC >= 0.5 (IRF1-BirA*) or log2FC >= 1 (STAT1-BirA*) enrichment above background (BirA*-NLS control) with a $P$adj value of <0.05. (A) Venn diagram of STAT1 interactors at steady state and upon IFNβ or IFNγ treatment. (B) Venn diagram of IRF1 interactors at steady state and upon IFNβ or IFNγ treatment. (C) Venn diagram of unique and common STAT1 and IRF1 interactors. (D) Heatmap showing log2FC of STAT1 and IRF1 interactors, grouped by functional annotation. Proteins labeled in bold were encoded by genes upregulated early after IFN treatment (1.5 h) in our PRO-Seq screen. Color-coded values represent log2FC with regard to the NLS control. (E) NuA4 (Tip60) complex components. Proteins labeled in purple were found in the proximity labeling screen using STAT1-BirA*. (F) PBAF complex components. Proteins labeled in purple were found in the proximity labeling screen using IRF1-BirA*.

Jak inhibitor ruxolitinib (InvivoGen, catalog # INCB018424) for 3, 2, 1, or 0.5 h before lysing for RNA isolation.

## RNA isolation, cDNA synthesis, and RT-qPCR

Total RNA from BMDMs and RAW 264.7 cells was isolated using the NucleoSpin RNA II kit (Macherey-Nagel, Catalog #740955). The cDNA of pre-mRNA was synthesized by the use of random hexamer primers (Thermo-Fisher Scientific) and RevertAid Reverse Transcriptase (Thermo-Fisher Scientific). Real-time quantitative PCR was run on the Eppendorf Mastercycler (SybrGreen) using the LUNA Universal qPCR Master Mix (New England Biolabs, Catalog # M3003). Primers used for real-time qPCR can be found in Appendix Table S2.

## Western blot

Cells were lysed in Laemmli buffer (120 mM Tris-HCl pH 8, 2% SDS and 10% glycerol) and protein concentration was measured using Pierce™ BCA Protein Assay kit (Thermo-Fisher Scientific). In all, 20 µg of protein was resuspended with SDS-loading dye (50% β-mercaptoethanol and 0.02% Bromphenol blue). Samples were boiled and loaded on a 10% SDS polyacrylamide gel. Proteins were blotted on a nitrocellulose membrane in carbonate buffer (3 mM $Na_2CO_3$, 10 mM $NaHCO_3$ and 20% ethanol) for 16 h at 200 mA and 2 h at 400 mM at 4 °C. Membranes were blocked for 1–2 h at room temperature in 5% milk powder in TBS-T (0.2 M Tris-HCl pH 7.5, 1.5 M NaCl and 1% Tween-20). After washing three times with TBS-T, primary antibody was added, and membranes were incubated overnight at 4 °C while shaking. Afterward, membranes were washed three times with TBS-T and further incubated with the appropriate secondary antibody for 1 to 2 h at room temperature while shaking. After another three washing steps with TBS-T, membranes were incubated with SuperSignal West Pico Chemiluminescent substrate (Thermo-Fisher Scientific) and developed using the ChemiDoc™ Imaging system from Bio-Rad. Primary antibodies: GAPDH (Millipore, Catalog # ABS16, 1:3000), STAT1 (Cell signaling, Catalog # 14995, RRID:AB_2716280 1:1000), IRF1 (Cell signaling, Catalog # 8478, 1:2000, RRID:AB_10949108). Secondary antibodies: Jackson ImmunoResearch Inc., Catalog # 111-035-003, 1:6000 and Catalog # 115-035-144, 1:6000).

## Immunoprecipitation

Cells were seeded in 10 cm dishes and treated with either IFNβ (150 IU/ml, PBL Assay Science; Catalog # 12400-1) or IFNγ (10 ng/ml, eBioscience; Catalog # 14-8311-63) for 1.5 h. Cells were washed three times with ice-cold PBS and lysed in 1 ml IP lysis buffer (10 mM Tris-HCl pH 7.5, 50 mM NaCl, 30 mM NaPPi, 50 mM NaF, 2 mM EDTA, 1% Triton-X, 1 mM DTT, 0.1 mM PMSF and

1× complete EDTA-free protease inhibitor cocktail (Sigma-Aldrich)). After 5 min incubation on ice, cell lysate was centrifuged for 5 min at 12.000 rcf. The supernatant was transferred to a fresh tube, and 200 µl magnetic beads (Dynabeads protein G, Life Technology, Catalog # 10003D) were added to preclear unspecific binding to the beads. Lysate was rotated for 30 min at room temperature. After the removal of the beads, protein concentration of the precleared sample was measured using Pierce™ BCA Protein Assay kit (Thermo-Fisher Scientific). Equal amounts of proteins were used from each condition and filled up to 200 µl final volume. Samples were incubated with the following antibodies overnight at 4 °C while rotating: IgG (Cell signaling, Catalog # 3900 S, 1 µl), STAT1 (Cell signaling, Catalog # 9172, 5 µl) and IRF1 (Cell signaling, Catalog # 8478, 4 µl). 50 µl of magnetic beads per sample were added for 3 h at 4 °C while rotating. After bead binding, beads were washed five times for 5 min at 4 °C with 1 ml of IP lysis buffer (w/o inhibitors). Proteins were eluted from the beads for 10 min at 95 °C and 1200 rpm using SDS sample buffer (250 mM Tris-HCl pH 6.8, 20% Glycerol, 1.6% SDS, 20% β-Mercaptoethanol and 0.002% Bromophenol blue) on a thermoshaker. Samples were transferred to fresh tubes and loaded on a 10% SDS acrylamide gel.

## Site-directed ChIP and ChIP-Seq/chipmentation

Overall, $1.5 \times 10^7$ BMDMs were seeded on day 9 of differentiation in TC-treated 15-cm dishes (DMEM supplemented with 10% FCS, Pen/Strep and M-CSF). Cells were stimulated for 48, 24, and 1.5 h with either IFNβ or IFNγ. On day 11, cells were washed once with ice-cold PBS and cross-linked with 1% formaldehyde (Thermo-Fisher Scientific, Catalog # 28906) in PBS for 10 min at room temperature while shaking. For quenching, glycine was added to a final concentration of 0.125 M and cells incubated for another 10 min at room temperature while shaking. Afterwards, cells were washed twice with ice-cold PBS and harvested in PBS supplemented with 0.1 mM PMSF (two plates were pooled of each condition). Cells were centrifuged for 5 min at 1350 rcf, pellets snap-frozen in liquid nitrogen and further stored at −80 °C for up to 6 months. Pellets were thawed on ice for 1 h before resuspending them in 5 ml of LB1 (50 mM HEPES, 140 mM NaCl, 1 mM EDTA, 10% Glycerol, 0.5% NP-40 and 0.25% Triton X-100). Samples were rotated for 10 min at 4 °C and then centrifuged for 5 min at 1350rcf at 4 °C. Supernatant was removed and pellets resuspended in 5 ml of LB2 (10 mM Tris, 200 mM NaCl, 1 mM EDTA and 0.5 mM EGTA). Samples were again rotated for 10 min at 4 °C before pelleting the cells for 5 min at 1350rcf at 4 °C. Pellets were mixed with 3 ml of LB3 (10 mM Tris, 100 mM NaCl, 1 mM EDTA, 0.5 mM EGTA, 0.1% deoxycholate and 0.5% N-lauroylsarcosine) and split into two in 15-ml polypropylene tubes suitable for the Bioruptor® Pico (Diagenode, Catalog # C01020031). One lid of a 1.5-ml tube was filled with sonication beads and added to each sample. Settings for the Bioruptor:

Power = high; "On" interval = 30 s; "OFF" interval = 45 s; 6 cycles. After chromatin shearing, samples were centrifuged for 10 min at 16.000 rcf at 4 °C to remove cellular debris. Chromatin concentration was measured and 300 μl of 10% Triton X-100 added to each sample. In all, 25 μg of chromatin was used for each IP. As an input control, 25 μg of chromatin was transferred to a fresh tube and stored at 4 °C until the elution step. Antibodies were added to the lysates and filled up with dilution buffer (16.5 mM Tris pH 8, 165 mM NaCl, 1.2 mM EDTA, 1% Triton X-100, 0.1% SDS, 0.1 mM PMSF and complete EDTA-free protease inhibitor cocktail (Sigma-Aldrich)) to a final volume of 1 ml. Antibodies used: STAT1 (Cell signaling, Catalog #14995, 10 μl), STAT2 (Cell signaling, Catalog # 72604, 15 μl), IRF9 (6F1-H5 hybridoma supernatant, 150 μl, (Platanitis et al, 2019)), IRF1 (Cell signaling, Catalog # 8478, 5 μl), phospho-STAT1 (Tyr701; Cell signaling, Catalog # 9167, 1:1000, RRID:AB_561284), phospho- STAT2 (Tyr689, Merck Millipore, Catalog # 07-224, RRID:AB_2198439) and IgG (Cell signaling, Catalog # 3900, 1 μl). Samples were incubated overnight at 4 °C while rotating. In addition, 50 μl of magnetic beads (Dynabeads protein G, Life Technologies, Catalog # 10003D) per sample were washed twice with dilution buffer and blocked overnight with dilution buffer supplemented with 1% BSA at 4 °C while rotating. The day after, 50 μl magnetic beads were added to each sample and incubated for 3 h at 4 °C while rotating. Next, beads were washed once with RIPA buffer (50 mM Tris-HCl pH 8, 150 mM NaCl, 1% NP-40, 0.1% SDS and 0.5% sodium deoxycholate), twice with high salt buffer (50 mM Tris-HCl pH 8, 500 mM NaCl, 1% NP-4,0 and 0.1% SDS), twice with lithium chloride buffer (50 mM Tris-HCl pH 8, 250 mM LiCl, 1% NP-40, and 0.5% sodium deoxycholate) and once with TE buffer (10 mM Tris-HCl pH 8 and 1 mM EDTA) every 10 min at 4 °C. Afterward, samples (including Input) were eluted from the beads in freshly prepared elution buffer (2% SDS, 100 mM NaHCO₃ and 10 mM DTT) for 1 h on a thermoshaker at room temperature and 1400 rpm. To reverse cross-linking between proteins and DNA, NaCl to a final concentration of 200 mM was added and samples incubated at 65 °C and 300 rpm for a maximum of 16 h on a thermoshaker. Next, Proteinase mix (final concentration: 0.1 mg/ml Proteinase K, 40 mM Tris-HCl pH 8 and 10 mM EDTA) was added and samples incubated for on a thermoshaker for 1 to 2 h at 55 °C and 850 rpm. Samples were transferred to 5Prime phase-lock gel tubes and mixed properly 1:1 with phenol-chloroform-isoamyl alcohol (PCI). Samples were centrifuged for 5 min at 12.000 rcf, supernatant transferred into a fresh tube and mixed with 800 μl ethanol, 40 μl of 3 M CH₃COONa pH 5.3 and 1 μl glycogen. Samples were stored for a minimum of 3 h at −20 °C (or overnight). DNA was precipitated by centrifugation at 16.000 rcf for 45 min at 4 °C, washed in ice-cold 70% ethanol, shortly air-dried and diluted in H₂0. DNA was incubated for 10 min at 65 °C before performing RT-qPCR using the KAPA SYBR FAST qPCR Kit from KAPA Biosystems. Primers used for real-time qPCR can be found in Appendix Table S2.

Chipmentation was performed according to the previously described protocol (Schmidl et al, 2015), with minor adaptions. Until sonication, steps of ChIP were followed. Antibody incubation was in RIPA buffer conditions (final concentration: 10 mM Tris-HCl pH 8.0, 1 mM EDTA pH 8.0, 140 mM NaCl, 1% Triton X-100, 0.1% SDS, 0.1% sodium deoxycholate, 1× protease inhibitors (Sigma) and 1 μM PMSF) up to 1 mL per immunoprecipitation. For washing, RIPA buffer, RIPA-500 (10 mM Tris-HCl pH 8.0, 1 mM EDTA pH 8.0, 500 mM NaCl, 1% Triton X-100, 0.1% SDS and 0.1% DOC), RIPA-LiCl (10 mM Tris-HCl pH 8.0, 1 mM EDTA pH 8.0,

250 mM LiCl, 1% Triton X-100, 0.5% DOC and 0.5% NP-40) and Tris pH 8.0 were used twice each. Beads were resuspended in 25 μl tagmentation reaction mix (10 mM Tris pH 8.0, 5 mM MgCl₂, 10% v/v dimethylformamide) containing 1 μl Tagment DNA Enzyme (Nextera DNA Sample Prep Kit (Illumina)), followed by incubation at 37 °C for 10 min. Further washes of the beads with RIPA-LS and Tris pH 8.0 were followed by removal of supernatant and incubation with 10.5 μl 20 mM EDTA. Sample was heated for 30 min at 50 °C, added 10.5 μl 20 mM MgCl2 + 25 μl 2× KAPA (preheated 20 s to 98 °C) and incubated for 5 min at 72 °C, 10 min at 95 °C and cooled on ice. Amplification and sequencing were performed as described for Chipmentation (Schmidl et al, 2015). The libraries were sequenced by the Biomedical Sequencing Facility at the Center for Molecular Medicine (CeMM), Vienna using the Illumina HiSeq3000/4000 platform and the 25-bp paired-end configuration. For library generation, the NEBNext Ultra II DNA Library Prep Kit for Illumina from NEB (New England Biolabs (NEB), Catalog # E7645) was used according to the manufacturer's protocol. Quality checking and sequencing was performed at the Vienna Biocenter Core Facilities NGS Unit.

## TurboID: cloning and constructs

Full-length mouse *Stat1* (NM_001205313.1) and *Irf1* (NM_008390.2) coding sequence was cloned together with V5-TurboID adapted from Addgene plasmid V5-TurboID-NES_pCDNA3 (Addgene # 107169) into the pCW57.1 mCherry vector (Catalog # 41393, adapted from Versteeg lab, Max Perutz Labs). As a control, V5-TurboID was cloned together with an NLS sequence into the pCW57 mCherry vector. N-terminal V5-TurboID-tagged *Irf1* and N-terminal V5-TurboID-tagged NLS constructs were transduced into *Irf1*⁻/⁻ RAW 264.7 cells. C-terminal TurboID-V5-tagged Stat1 and N-terminal V5-TurboID-tagged NLS were transduced into *Stat1*⁻/⁻ RAW 264.7 cells. Cells were sorted for mCherry, and single clones selected.

## TurboID: biotinylation, nuclear extraction, and Strep IP

Cells were seeded in 15-cm dishes 2 days before harvesting. Single-cell clones of RAW 264.7 *Stat1*⁻/⁻ cells transduced with STAT1-TurboID-V5 as well as V5-TurboID-NLS were treated for 24 h with Doxycycline (final concentration: 2 μg/ml). Single-cell clones of RAW 264.7 *Irf1*⁻/⁻ cells transduced with V5-TurboID-IRF1 as well as V5-TurboID-NLS were treated with 2 μg/ml Doxycycline for 4 h. Cells were either treated with IFNβ or IFNγ for 3 h and biotinylation was induced by adding 500 μM Biotin for 10 min. Single-cell clones of RAW 264.7 *Stat1*⁻/⁻ cells transduced with V5-TurboID-STAT1 as well as V5-TurboID-NLS were treated with 2 μg/ml Doxycycline for 24 h. Cells were either treated with IFNβ or IFNγ for 3 h and biotinylation was induced by adding 500 μM Biotin for 10 min. The experiment was performed in biological triplicates. Cells were washed 3 times in ice-cold PBS, cell number counted, and aliquots (2 × 10⁶ cells per tube) snap-frozen in liquid nitrogen before storing them at −80 °C. In total, 2 × 10⁶ cells were lysed for 15 min on ice in CEB buffer (10 mM HEPES, 10 mM KCl, 1.5 mM MgCl₂, 0.1% NP-40, and complete EDTA-free protease inhibitor cocktail). Cells were centrifuged for 5 min at 500 rcf and 4 °C. Nuclear pellet was washed three times with wash buffer A (10 mM HEPES, 10 mM KCl, 1.5 mM MgCl₂, 0.1% NP-40, and

EDTA-free protease inhibitor cocktail). The pellet was lysed in 200 μl NEB (25 mM HEPES pH 7.9, 1.5 M MgCl$_2$, 0.1 M EDTA) supplemented with EDTA-free protease inhibitor cocktail, 1 mM PMSF, 1 mM Na$_2$VO$_3$ and 550 mM NaCl. Nuclei were vortexed 5 times every 10 min for 10 s while being kept on ice, centrifuged for 5 min at 16.000 rcf at 4 °C and transferred to a fresh protein low-bind tube. Protein concentration was measured using Pierce™ BCA Protein Assay kit (Thermo-Fisher Scientific). Overall, 200 μg protein was added to 100 μl acetylated beads (Hollenstein et al, 2022) (Pierce Streptavidin, Catalog # 88817) and filled up with NEB to reach a final concentration of 200 mM NaCl. Samples were incubated overnight at 4 °C while rotating. Next day, beads were washed once for 5 min with NEB, three times for 5 min with RIPA buffer (0.1% SDS, 0.5% sodium deoxycholate, 1% NP-40 substitute, 50 mM Tris, 150 mM NaCl, 1 mM Na$_3$VO$_4$, 2 mM NaF, 2 mM PMSF, Protease Inhibitor) and six times with TBS (50 mM Tris, 150 mM NaCl pH 7.5). Proteome analyses were performed by the Mass Spectrometry Facility at Max Perutz Labs using the VBCF instrument pool. Identified interactors were filtered for log2FC >= 0.5 (IRF1-BirA*) or log2FC >= 1 (STAT1-BirA*) enrichment above background (BirA*-NLS control) with a Padj value of <0.05.

## Sample preparation for mass spectrometry analysis

The beads were resuspended in 50 μL 1 M urea and 50 mM ammonium bicarbonate. Disulfide bonds were reduced with 2 μL of 250 mM dithiothreitol (DTT) for 30 min at room temperature before adding 2 μL of 500 mM iodoacetamide and incubating for 30 min at room temperature in the dark. The remaining iodoacetamide was quenched with 1 μL of 250 mM DTT for 10 min. Proteins were digested with 150 ng LysC (mass spectrometry grade, FUJIFILM Wako chemicals) in 1.5 μL 50 mM ammonium bicarbonate at 25 °C overnight. The supernatant without beads was digested with 150 ng of trypsin (Trypsin Gold, Promega) in 1.5 μL 50 mM ammonium bicarbonate followed by incubation at 37 °C for 5 h. The digest was stopped by the addition of trifluoroacetic acid (TFA) to a final concentration of 0.5%, and the peptides were desalted using C18 Stagetips (Rappsilber et al, 2007).

## Liquid chromatography-mass spectrometry analysis

Peptides were separated on an Ultimate 3000 RSLC nano-flow chromatography system (Thermo-Fisher), using a pre-column for sample loading (Acclaim PepMap C18, 2 cm × 0.1 mm, 5 μm, Thermo-Fisher), and a C18 analytical column (Acclaim PepMap C18, 50 cm × 0.75 mm, 2 μm, Thermo-Fisher), applying a segmented linear gradient from 2% to 35% and finally 80% solvent B (80% acetonitrile, 0.1% formic acid; solvent A 0.1% formic acid) at a flow rate of 230 nL/min over 120 min.

Eluting peptides were analyzed on an Exploris 480 Orbitrap mass spectrometer (Thermo-Fisher) coupled to the column with a FAIMS pro ion-source (Thermo-Fisher) using coated emitter tips (PepSep, MSWil) with the following settings: The mass spectrometer was operated in DDA mode with two FAIMS compensation voltages (CV) set to -45 or -60 and 1.5 s cycle time per CV. The survey scans were obtained in a mass range of 350–1500 $m/z$, at a resolution of 60k at 200 $m/z$, and a normalized AGC target at 100%. The most intense ions were selected with an isolation width of 1.2 $m/z$, fragmented in the HCD cell at 28% collision energy, and the spectra recorded for max.

100 ms at a normalized AGC target of 100% and a resolution of 15k. Peptides with a charge of +2 to +6 were included for fragmentation, the peptide match feature was set to preferred, the exclude isotope feature was enabled, and selected precursors were dynamically excluded from repeated sampling for 45 s.

## Data analysis of mass spectrometry

MS raw data split for each CV using FreeStyle 1.7 (Thermo-Fisher), were analyzed using the MaxQuant software package (version 2.1.4.0) (Tyanova et al, 2016) with the Uniprot mouse reference proteome (version 2022.03, www.uniprot.org), as well as a database of most common contaminants. The search was performed with full trypsin specificity and a maximum of two missed cleavages at a protein and peptide spectrum match false discovery rate of 1%. Carbamidomethylation of cysteine residues was set as fixed, oxidation of methionine, and N-terminal acetylation as variable modifications. For label-free quantification, the "match between runs" only within the sample batch and the LFQ function were activated—all other parameters were left at default.

MaxQuant output tables were further processed in R 4.2.1 (R Core Team, 2018) using Cassiopeia_LFQ (https://github.com/moritzmadern/Cassiopeia_LFQ). Reverse database identifications, contaminant proteins, protein groups identified only by a modified peptide, protein groups with less than two quantitative values in one experimental group, and protein groups with less than 2 razor peptides were removed for further analysis. Missing values were replaced by randomly drawing data points from a normal distribution model on the whole dataset (data mean shifted by −1.8 standard deviations, a width of the distribution of 0.3 standard deviations). Differences between groups were statistically evaluated using the LIMMA 3.52.1 (Ritchie et al, 2015) with batch correction at 5% FDR (Benjamini–Hochberg).

## Downstream analysis of proteomic data

Gene ontology of identified interactors of STAT1 and IRF1 was analyzed using Enrichr (Xie et al, 2021; Kuleshov et al, 2016; Chen et al, 2013). Interactive network was visualized using STRING database and Cytoscape application (Szklarczyk et al, 2015; Shannon et al, 2003). Venn diagram showing overlap of enriched interactors between conditions were created using Amica application (Didusch et al, 2022).

## Proteomics data deposition

The mass spectrometry proteomics data have been deposited to the ProteomeXchange Consortium via the PRIDE partner repository (Perez-Riverol et al, 2019) with the dataset identifier PXD040337.

## ATAC-Seq

Bone marrow-derived macrophages were seeded in non-treated six-well plates and stimulated for either 1.5, 4, or 48 h with either IFNβ or IFNγ, washed twice with ice-cold PBS and counted (viability above 90%). Overall, 1 × 10$^6$ cells per condition were pelleted by centrifugation at 1200 rcf for 5 min at 4 °C using a V-bottom 2-ml tube. All the following steps were performed according to the in-house protocol of the VBC NGS Facility. The supernatant was removed and cells

resuspended in 500 µl nuclear isolation buffer (NIB; 0.32 M sucrose, 3 mM CaCl$_2$, 2 mM magnesium acetate, 0.1 mM EDTA, 10 mM Tris-HCl pH 8, 0.6% NP-40 and freshly added 1 mM DTT) and incubated for 5 min on ice. Cells were centrifuged for 5 min at 700 rcf at 4 °C. Supernatant was removed without disturbing the pellet and another 500 µl NIB was added following a 3-min incubation on ice. Samples were centrifuged for 5 min at 700 rcf and 4 °C. Supernatant was removed and nuclei gently resuspended in 20 µl ice-cold nuclear resuspension buffer (NRB; 50 mM Tris-HCl pH 8, 40% glycerol, 5 mM MgCl$_2$, 0.1 mM EDTA). Nuclei were counted and kept on ice (or for longer storage at −80 °C). Overall, 50.000 nuclei per sample were centrifuged at 1000 rcf for 5 min at 4 °C and resuspended in transposition mix (12.5 µl TD buffer, 5 µl TD enzyme and 7.5 µl H$_2$O; Illumina, Catalog # 20034197). Nuclei were incubated at 37 °C for 45 min and 800 rpm. Afterward, DNA was purified using NEB Monarch PCR & DNA cleanup kit (New England Biolabs, catalog #T1030) according to the manufacturer's instructions and eluted in 13 µl H$_2$O. For the library amplification, 12.5 µl of the eluted DNA was mixed with 5 µl I7 index primer (10 µM, dual indexing), 5 µl I5 index primer (10 µM, dual indexing), 2.5 µl Evagreen and 2x Q5 PCR Master mix (NEB, Catalog #50492 L) and end point PCR was performed using following program: 5 min at 72 °C, 1 min at 98 °C, 6–7 cycles with 10 s at 98 °C, 30 s at 65 °C and 60 s at 72 °C. PCR reactions were purified by adding 15 µl of SPRI beads which were prepared by the NGS facility (MBSpure beads) and incubated for 5 min at room temperature. 50 µl of MBSpure beads were added to the supernatant, mixed well, and incubated for 5 min at room temperature on a magnet. Beads were washed twice with 150 µl of 80% ethanol. The supernatant was removed, beads dried for 30 s, and DNA eluted in 20 µl of H$_2$O. To check the quality of the libraries, samples were run on a bioanalyzer to determine the size distribution. Sequencing was performed on a NovaSeq6000 S2 PE50 or NovaSeq SP PE50 at the Vienna Bioscience (VBC) Next Generation Sequencing Facility.

## PRO-Seq

Overall, $1.5 \times 10^7$ bone marrow-derived macrophages were seeded on day 9 of differentiation in 15-cm non-treated dishes and stimulated for either 1.5, 4, 24, or 48 h with either IFNβ or IFNγ. The experiment was performed in triplicates using three mice per genotype. Cells were washed twice with ice-cold PBS and cell number counted (viability above 90%). All the following steps were performed according to the protocol from Kwak et al (Kwak et al, 2013) including adaptations from Mahat et al (Mahat et al, 2016) and Ursula Schöberl from Rushad Pavri lab, IMP, Vienna. Cells were pelleted by centrifugation at 1200 rcf for 5 min at 4 °C using a V-bottom 2-ml tube. The supernatant was removed and cells resuspended in 1 ml nuclear isolation buffer (NIB; 0.32 M sucrose, 3 mM CaCl$_2$, 2 mM magnesium acetate, 0.1 mM EDTA, 10 mM Tris-HCl pH 8, 0.6% NP-40 and freshly added 1 mM DTT) supplemented with 0.2 U/µl recombinant RNase Inhibitor (Takara, Catalog # 2313B) and incubated for 5 min on ice. Cells were centrifuged for 5 min at 700 rcf at 4 °C. Supernatant was removed without disturbing the pellet and another 500 µl NIB was added following a 3-min incubation on ice. Samples were centrifuged for 5 min at 700 rcf and 4 °C. Supernatant was removed and nuclei gently resuspended in 200 µl ice-cold nuclear resuspension buffer (NRB; 50 mM Tris-HCl pH 8, 40% glycerol, 5 mM MgCl$_2$, 0.1 mM EDTA) supplemented with 0.2 U/µl recombinant RNase Inhibitor

(Takara, Catalog # 2313B). Nuclei were counted and kept on ice (or for longer storage at −80 °C). Nuclear run-on was performed with $1 \times 10^7$ nuclei in 100 µl NRB supplemented with 0.2 U/µl RNase inhibitor mixed with 30 °C preheated 100 µl of nuclear run-on Master Mix (10 mM Tris-HCl pH 8, 5 mM MgCl$_2$,1 mM DTT, 300 mM KCl, 50 µM ATP, 50 µM GTP, 50 µM Biotin-11-CTP (Jena Bioscience, Catalog # NU831-biox), 50 µM Biotin-11-UTP (Jena Bioscience, Catalog # NU821-biox), 1% Sarkosyl and 0.4 U/µl recombinant RNase Inhibitor) by pipetting 15 times up and down and incubating at 30 °C for exactly 3 min. Nuclear run-on was stopped by adding 500 µl TRIzol LS and incubation at room temperature for 5 min. Samples were snap-frozen in liquid nitrogen and kept at −80 °C until further use. Nascent RNA was isolated using TRIzol LS and 130 µl of Chloroform (vortex 15 s, incubate at room temperature for 2–3 min, centrifuged for 15 min at 20.000 rcf and 4 °C). In all, 1 µl of Glycoblue was added to the aqueous phase plus 2.5× volume of 100% ethanol. Samples were incubated for 10 min at room temperature and centrifuged at 20.000 rcf for 15 min at 4 °C. RNA pellet was washed with 80% ethanol, air-dried for around 2–3 min, and finally resuspended in 20 µl of H$_2$O before being heat denatured at 65 °C for 40 s. Base hydrolysis was performed using 0.2 N NaOH on ice for 20 min. The reaction was neutralized by adding 1 volume of 1 M Tris-HCl pH 6.8. Buffer exchange was performed using Bio-Rad P-30 columns according to the manufacturer's instructions (Bio-Rad, Catalog #7326231). After elution, RNase inhibitor was added to a final concentration of 0.2 U/µl. Enrichment of the fragmented nascent RNA was performed using Streptavidin M280 beads that were beforehand equilibrated by washing once with bead washing buffer 1 (0.1 N NaOH, 50 mM NaCl), twice with bead washing buffer 2 (100 mM NaCl) and resuspension in binding buffer (10 mM Tris-HCl pH 7.4, 0.3 M NaCl and 0.1% Triton X-100). Beads were added to the samples and incubated for 20 min at room temperature while rotating. Beads were washed twice with high salt buffer (50 mM Tris-HCl pH 7.4, 2 M NaCl and 0.5% Triton X-100), twice with binding buffer (see above) and once with low salt buffer (5 mM Tris-HCl pH 7.4 and 0.1% Triton X-100). RNA isolation from the beads was performed using 300 µl of TRIzol reagent in two rounds, pooling the aqueous phase of both before starting ethanol precipitation. RNA pellet was resuspended in 4 µl of 12.5 µM reverse 3' RNA adaptor (Rev3a; 5'-5Phospho rNrNrNrNrNrNrNr NrGrAr-UrCrGrUrCrGrGrArCrUrGrUrArGrArArCrUrCrUrGrAr ArC-/inverted dT/-3'), incubated for 20 s at 65 °C, placed on ice, and 6 µl of ligation mix (1 µl 10× T4 RNA ligase buffer, 1 µl 10 mM ATP, 2 µl 50% PEG, 1 µl RNase Inhibitor, 1 µl T4 RNA ligase I (New England Biolabs)) was added before ligating overnight at 16 °C. RNA was isolated using Streptavidin M280 beads, extracted and finally precipitated using TRIzol (as described above). RNA pellet was then resuspended in 5 µl H$_2$0 and 5' Cap and triphosphate repair initiated by adding 5 µl of Cap-Clip™ mix (1 µl 10X Cap-Clip™ Acid Pyrophosphatase Reaction Buffer, Cap-Clip™ Acid Pyrophosphatase (5 U/µl) in 50% glycerol, 0.5 µl RNase inhibitor and 3 µl H$_2$O). Samples were incubated at 37 °C for 1 h. 5' hydroxyl repair was performed by adding 90 µl of PNK mix (2.5 µl T4 PNK, 10 mM ATP, 10 µl 10x PNK buffer and 66.5 µl H$_2$O) to the samples and incubation for another hour at 37 °C. RNA extraction and precipitation was performed using 500 µl of TRIzol and 100 µl of Chloroform. Dried RNA pellet was dissolved in 4 µl of 12.5 µM reverse 5' RNA adaptor (VRA5a; 5'- rCrCrUrUrGr

GrCrArCrCrCrGrArGrArArUrUrCrCrArNrNrNrN-3′), incubated at 65 °C for 20 s and placed on ice. In all, 6 μl of ligation mix (see above) was used and ligation performed overnight at 16 °C. RNA was again isolated using Streptavidin M280 and precipitated using TRIzol and Chloroform. The pellet was dissolved in 10 μl of $H_2O$ before reverse transcribing the nascent RNA into cDNA libraries using Superscript III (Invitrogen, Catalog # 18080) and RP1 primer. Library amplification was performed using KAPA HiFi Real-time PCR library amplification Kit and Illumina primers containing standard barcodes (forward: RP1; reverse: RPI1-82). The quality of the amplified libraries was checked using Bioanalyzer and amplicons from 170 to 600 bp were excised using PIPPIN-Prep (2%). Samples were sequenced on Illumina NovaSeq S4 PE150 XP (VBC NGS facility, Vienna, Austria).

## PRO-Seq read alignment and estimation of transcript abundance

Pre-processing and alignment of PRO-seq reads were performed using pro-seq-2.0 pipeline (Chu et al, 2019). The pipeline automates removal of the adapter sequences, trimming based on base quality and deduplication of the reads based on the UMI barcodes. Further, sequencing reads are mapped to the reference genome using BWA and aligned BAM files are converted into bigWig format. We used the *Mus musculus* GRCm38 reference genome and the associated GENCODE (M29) gene annotations. The bigwig files obtained from pro-seq.2.0 were further used to generate a count matrix corresponding to annotated genes as well as enhancers as described in the Tf-target package (Chu et al, 2018).

## PRO-Seq quantification of gene loci and downstream analyses

For estimating read counts in genes, we omitted the first 500 bases downstream of the TSS to avoid a bias from promoter-proximal polymerase pausing. In addition, genes with gene body <1 kb were excluded from the analysis. Principal component analysis (PCA) and downstream differential analysis were performed using the DESeq2 package (version 1.36.0) (Love et al, 2014). Variance-stabilized reads were used for performing PCA. For differential analysis in wild-type cells, each treatment condition was compared to the homeostatic condition (untreated sample) and for analysis including the knockouts, treatment conditions in $Irf9^{-/-}$ or $Irf1^{-/-}$ were compared to the respective conditions in wild-type cells. Differentially expressed genes were selected based on the following criteria: absolute log2FoldChange >=1 and adjusted $P$ value < 0.01 by Wald test. Following the differential analysis in wild-type cells, the first 1000 differentially expressed genes in each treatment condition (ordered based on adjusted $P$ values) were further selected. Z-score normalized counts from these genes in wild-type cells were estimated and used for performing hierarchical clustering by "Ward. D2" method. To create the heatmap, we used the pheatmap package from R (Kolde, 2015) and the cutree_rows option to separate 11 clusters with strikingly different patterns of gene expression based on visual exploration. The numbering was autogenerated by the program. The scale in the heatmap represents the z-scores of variance-stabilized reads, calculated across all genotype and treatment conditions, separately for each IFN type. A

total of 3126 genes were used for this analysis. 11 distinct clusters were defined based on their transcriptional profile across timepoints in each IFN treatment. Further, GO Enrichment Analysis of each cluster was performed by overrepresentation analysis of GO terms in biological process ontology, using the clusterProfiler (v4.4.4) package (Yu et al, 2012). For visualizing the effect on transcription in the aforementioned clusters upon the loss of IRF9 or IRF1, corresponding z-score normalized read counts in the cells derived from wildtype, $Irf9^{-/-}$ and $Irf1^{-/-}$ were plotted using ggplot2 package (Wickham, 2016).

## Enhancer analysis

We used dREG package (Danko et al, 2015; Wang et al, 2019) for identifying active transcriptional regulatory elements (TREs) at each timepoint. To increase the sensitivity of dREG, we merged the bigWigs of biological replicates under each condition. The replicate bigWig files were first combined based on the condition and then according to clustering in PCA, using the script https://github.com/Danko-Lab/proseq2.0/blob/master/mergeBigWigsprovidedinproseq-2.0. Using the resulting merged bigWig files, transcription regulatory elements (TREs) were identified by employing the dREG package. The TREs identified in common as well as specific for each merged bigWig files were further combined using bedtools v2.30.0 and used for further analysis. A count matrix was created by adding reads from both strands in the region, defined by bed files obtained from dREG corresponding to TREs as mentioned in the TF-target package. Differential analysis was performed similarly to that of genic regions using the DESeq2 package. Those regions that fell within 50 kb on either side of the TSS of annotated genes and in addition following the transcriptional trend of the genes in consideration (log2FoldChange >=1 and padj <0.01 by Wald test (DESeq2) were described as putative enhancers in our analysis.

## Pol II pausing analysis

Reads in the region up to 500 bp downstream of annotated TSS (TSS region) and rest of the gene body except for the last 500 bp (gene body region) were calculated and normalized by counts-per-million and by region length (cpm/bp) for each gene. Pausing Index (PI) was calculated as the ratio of normalized reads in the TSS region (cpm/bp) to normalized reads in the gene body region(cpm/bp). Mann–Whitney $U$ test was performed comparing untreated to the most relevant timepoint. Further, we calculated the $\log_e$ transformed ratio of pausing indices between each treatment and untreated condition. All computations were performed using the R statistical package.

## ChIP-Seq analysis

A ChIP-Seq and ChIPmentation experiment was generated with IRF9 antibody in BMDMs, both for short (1.5 h) and prolonged (24 h) interferon treatments. These dataset as well as published ChIP-seq data (Platanitis et al, 2019; Langlais et al, 2016) were analyzed using the ChIP-seq pipeline from the nf-core framework (nf-core/chipseq v1.1.0) (https://zenodo.org/record/3529400#.YA8YGmRKjZk). The Illumina iGenome *Mus musculus* GRCm38 was used as the reference genome. For genome browser tracks of IRF9 binding, newly generated dataset was used.

## ChIP-Seq integration with PRO-Seq

For IRF9, bed files corresponding to peaks obtained from ChIP-Seq data on early and late timepoints (representative of ChIPmentation and ChIP-Seq) and published ChIP-Seq (Platanitis et al, 2019) were combined using BEDOPS v2.4.41 and intersected with selected enhancer regions defined before, to estimate IRF9 binding to the potential enhancer regions for relevant clusters. Similarly for IRF1, the peak files were generated from the published dataset (Langlais et al, 2016).

## ATAC-seq data processing

The ATAC-seq pipeline from the nf-core framework (nf-core/atacseq v1.2.1, https://zenodo.org/record/3965985#.YA8a8GRKjZk) was used for ATAC-seq preliminary analysis of ATAC-Seq data. Illumina iGenome *Mus musculus* GRCm38 was used as the reference genome. Downstream analysis of ATAC-seq data was performed using MACS2 narrow-peak calling and differential chromatin accessibility analysis (included in nf-core/atacseq pipeline).

## ATAC-seq integration with PRO-Seq

For integrating ATAC-seq and PRO-Seq, we used the respective differential analyses from DESeq2. For ATAC-seq, each called interval (available output from the nf-core pipeline) was annotated to a gene by using the HOMER script *annotatePeaks.pl.* (Heinz et al, 2010). Thereby, peaks were annotated to genes and could be compared with the PRO-Seq analysis. Further, a distance filtering based on absolute distance of the ATAC-Seq peak region to the annotated TSS < = 2000 was used in this analysis. pyGenomeTracks was used for visualizing the genome browser tracks and bed files corresponding to enhancers and transcription factor binding peaks (Ramírez et al, 2018; Lopez-Delisle et al, 2021). We calculated the z-scores for vst-normalized read counts across untreated and IFN treatment conditions (for early and late hours of treatment) independently in ATAC-seq and PRO-seq samples for each gene, followed by correlation between samples and representation of the trends as mean z-scores in PRO-seq and ATAC-seq in the respective treatment conditions.

## Statistical information

R and GraphPad Prism (Version 8) were used for statistical analysis. We used Wald test for differential analysis using DESeq2 to compare between untreated and each of the IFN-treated sample groups. For comparison of pausing indices between untreated and treated sample groups, the Mann–Whitney $U$ test was used. Heatmaps were produced with the pheatmap package in R. Barplots representing qPCR-derived pre-mRNA expression data and ChIP data show the mean values and standard deviation. Statistical analysis was performed using unpaired Student's $t$ test using GraphPad Prism (GraphPad) software. Asterisks denote statistical significance: not significant (ns), $*P < 0.05$; $**P \leq 0.01$; $***P \leq 0.001$; $****P \leq 0.0001$.

## Data availability

Raw data processed in this paper are available under SRA BioProject: PRJNA947574 (IRF9 ChIP-seq, Chipmentation), BioProject: PRJNA694816 and PRJNA937204 (ATAC-seq), BioProject: PRJNA931836 (PRO-seq), PRIDE: PXD040337 (Proteomics).

## Peer review information

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

## Acknowledgements

The authors thank Carlo Pecoraro and Physalia Courses for NGS data analysis training. We are especially grateful to Tinyi Chu and Sebastian Didusch for sharing their expertise and enthusiasm for data visualization, PRO-seq, and mass spectrometry analysis with us. Illumina sequencing was performed by the VBCF NGS Unit and Biomedical Sequencing Facility at CeMM. Proteomics analyses were performed by the Mass Spectrometry Facility at Max Perutz Labs using the VBCF instrument pool. Selection of cell lines by FACS was performed by the Max Perutz Labs FACS facility. Help and suggestions for PRO-seq and the establishment of CRISPR–Cas9 knockout cells by Ursula Schöberl (Rushad Pavri group, IMP), Henry Fabian Thomas (Christa Bücker group, Max Perutz labs) and Sara Scinicariello (Gijs Versteeg group, Max Perutz labs) is gratefully acknowledged. Funding was provided by the Austrian Science Fund (FWF) through projects SFB F6101, F6106, and F6107 to TD, MF, and MM. KF was supported by the FWF through the doctoral program W1261 Signaling Mechanisms in Cell Homeostasis.

## Author contributions

**Aarathy Ravi Sundar Jose Geetha**: Conceptualization; Formal analysis; Investigation; Writing—original draft. **Katrin Fischer**: Conceptualization; Formal analysis; Investigation; Writing—original draft. **Olga Babadei**: Investigation. **Georg Smesnik**: Investigation. **Alex Vogt**: Investigation. **Ekaterini Platanitis**: Investigation. **Mathias Müller**: Conceptualization; Resources; Funding acquisition. **Matthias Farlik**: Conceptualization. **Thomas Decker**: Conceptualization; Supervision; Funding acquisition; Writing—original draft; Writing—review and editing.

## Disclosure and competing interests statement

# Expanded View Figures

**Figure EV1.   Transcriptional response of interferon-stimulated genes in wild-type BMDM.** ▶

(**A**) Venn diagram showing numbers of significantly upregulated genes at indicated timepoints during IFNβ and IFNγ signaling in three independent replicates of BMDM (log2FC > 1) and padj < 0.01). (**B**) Log2FC of *Nos2, Cd86, Cxcl9, Cxcl10, Mx2, Ifit3 and Rsad2* across the denoted timepoints separated by IFNβ and IFNγ stimulation. (**C**) Bubble plot visualizing gene ontologies resulting from overrepresentation analysis performed per indicated clusters using clusterProfiler (*P* value cutoff = 0.05). (**D**) Heatmap of log2FC of genes belonging to the ISG core (Mostafavi et al, 2016) at respective timepoints during IFNβ and IFNγ stimulation. (**E**) Pie chart showing respective number of ISG-core genes in clusters 1, 2 and 9.

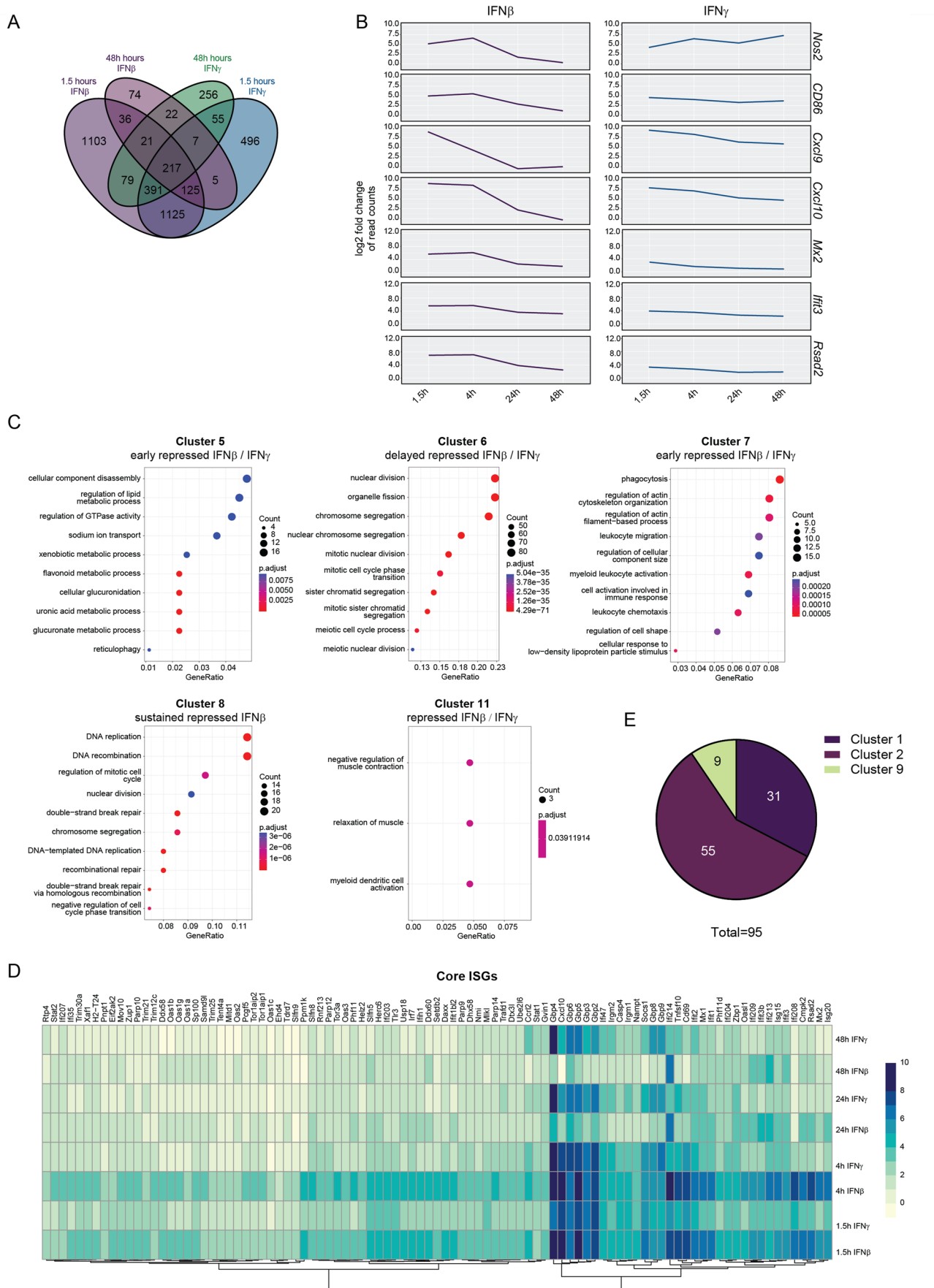

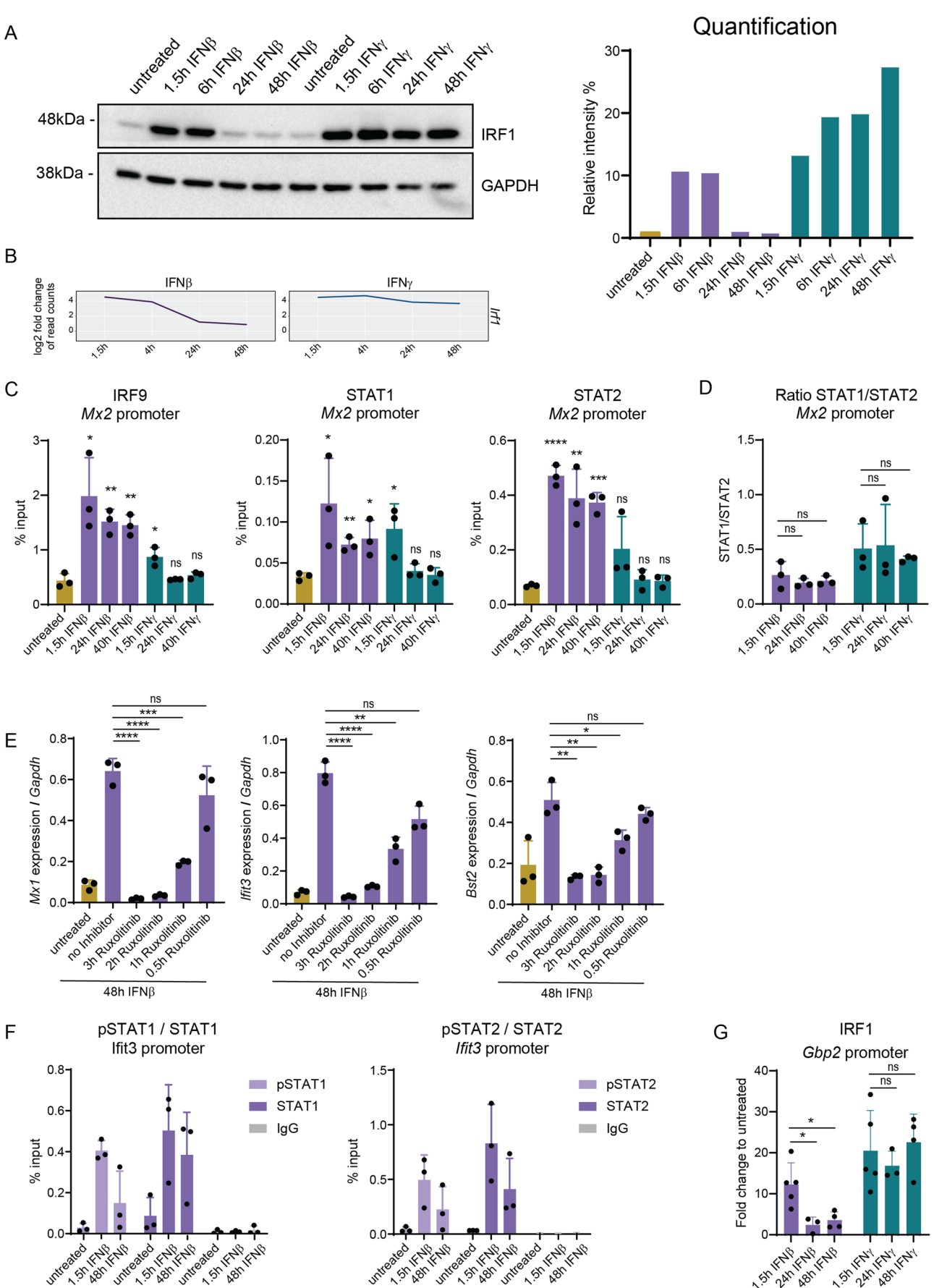

**Figure EV2. Promoter binding and phosphorylation requirement of transcription factors controlling ISG expression.**

(A) RAW 264.7 cells were treated with IFNβ or IFNγ for either 1.5 h, 4 h, 24 h or 48 h and protein levels of IRF1 and GAPDH were measured using western blotting ($n = 2$). GAPDH was used as a loading control. Quantification of the representative blot on the left was performed using Image Lab and is shown in the panel on the right. Relative intensities of the bands were normalized to their corresponding GAPDH levels. (B) Log2FC of *Irf1* derived from PRO-Seq data described in the legend to Fig. 1 across the denoted timepoints after IFNβ and IFNγ stimulation, respectively. (C) ChIP was performed in biological triplicates using antibodies against IRF9, STAT1 and STAT2 in IFNβ or IFNγ-treated wild-type BMDMs (1.5, 24 and 40 h). Graph represents RT-qPCR of genomic *Mx2*. (D) Graph represents ratio of binding of STAT1/STAT2 to the promoter of *Mx2* during early (1.5 h) and prolonged (24 h, 40 h) responses to IFNβ- and IFNγ stimulation of BMDMs. (E) Graph representing pre-mRNA levels of *Mx1*, *Ifit3* and *Bst2* in IFNβ-treated BMDMs (48 h). Additionally, cells were treated with ruxolitinib for indicated times ($n = 3$). Standard deviation and unpaired Student's *t* test statistics were calculated for each of the conditions indicated. *P* values are indicated as not significant (ns), *$P < 0.05$; **$P \le 0.01$; ***$P \le 0.001$; ****$P \le 0.0001$). (F) ChIP was performed using antibodies against STAT1, p(Y)STAT1, STAT2, p(Y)STAT2 and IgG in IFNβ-treated wild-type BMDMs (1.5 and 48 h). The graph represents RT-qPCR of genomic *Ifit3*. (G) Site-directed ChIP was performed using antibodies against IRF1 in IFNβ or IFNγ-treated wild-type BMDMs (1.5, 24 and 48 h). The graph represents RT-qPCR of genomic *Gbp2*. Input normalized values were used to calculate fold changes caused by interferon treatment relative to untreated cells. Standard deviation and unpaired Student's *t* test statistics were calculated for each of the conditions indicated. *P* values are indicated as not significant (ns), *$P < 0.05$; **$P \le 0.01$; ***$P \le 0.001$; ****$P \le 0.0001$). Source data are available online for this figure.

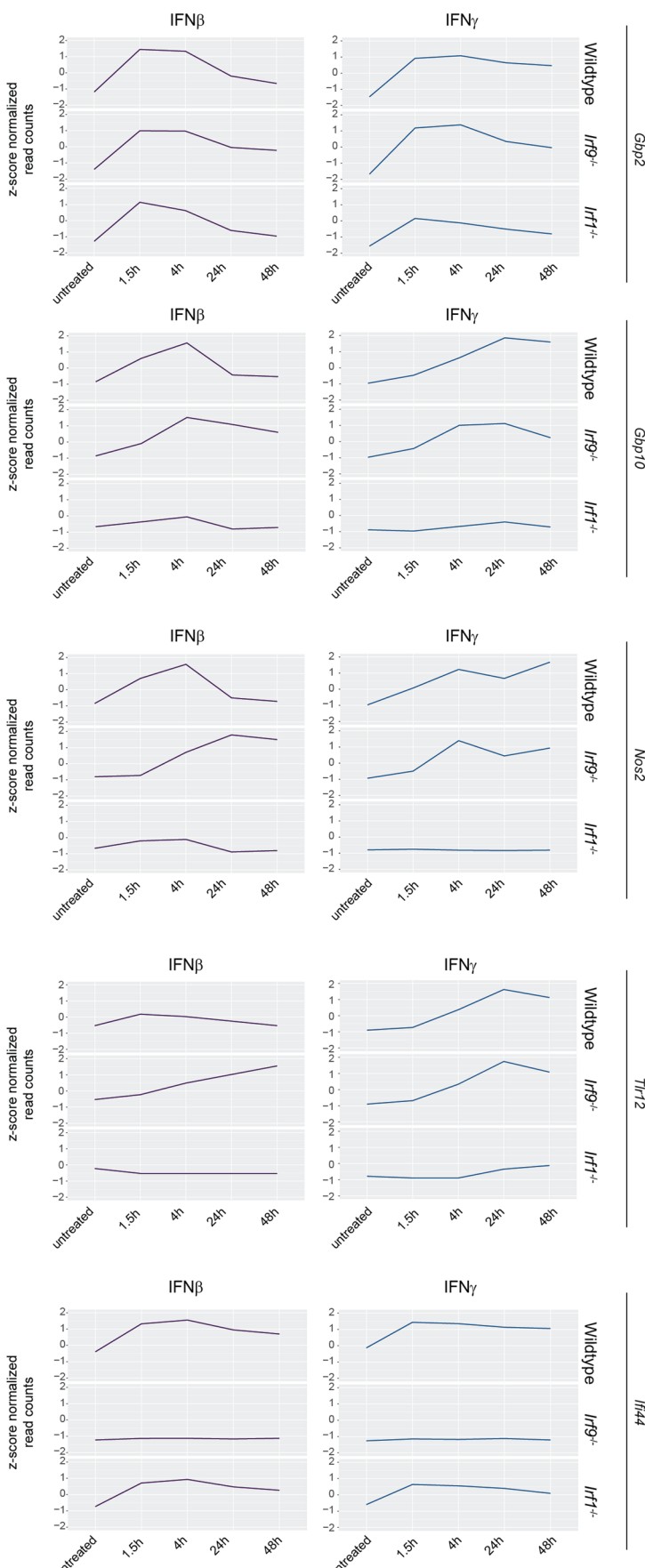

◀    **Figure EV3.   Transcriptional response to interferons in wild-type compared to IRF1- or IRF9-deficient BMDM.**

Z-score normalized read counts of *Gbp2, Gbp10, Nos2, Tlr12* and *Ifi44*, calculated across treatment times and genotypes. Counts were derived from PRO-Seq data described in the legend to Fig. 3.

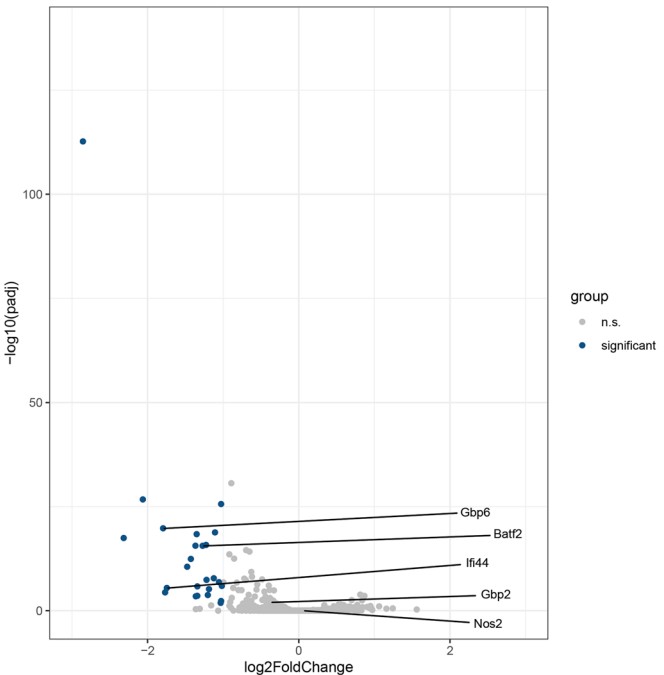

**Figure EV4.  IRF1-dependent chromatin accessibility of ISG promoters at steady state.**

Volcano plot of genes derived from ATAC-Seq of *Irf1*[−/−] and wild-type BMDMs as described in the legend to Fig. 5 at steady state. The log2-transformed fold change and −log10-transformed padj are shown on the *x* and *y* axis, respectively. Genes depicted in blue are significantly (log2FC <= 1, *P*adj < 0.05) downregulated in *Irf1*[−/−] BMDMs compared to their wild-type control.

A

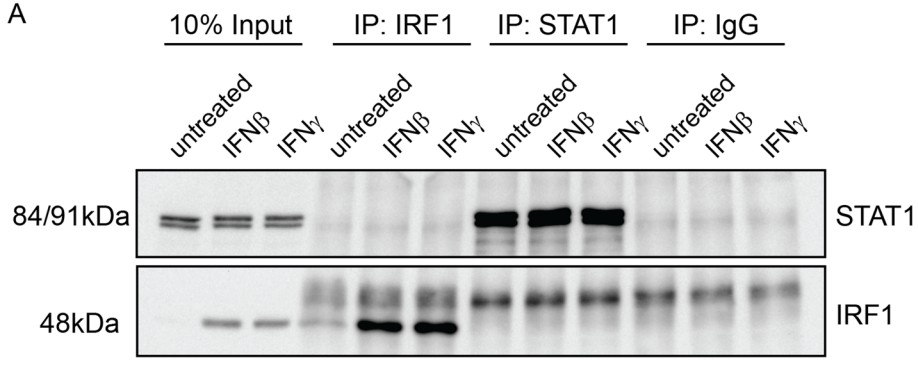

B

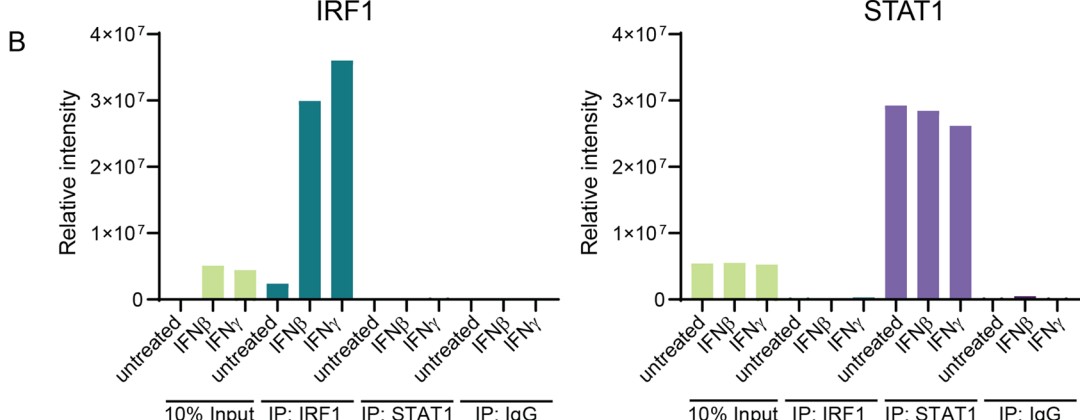

C

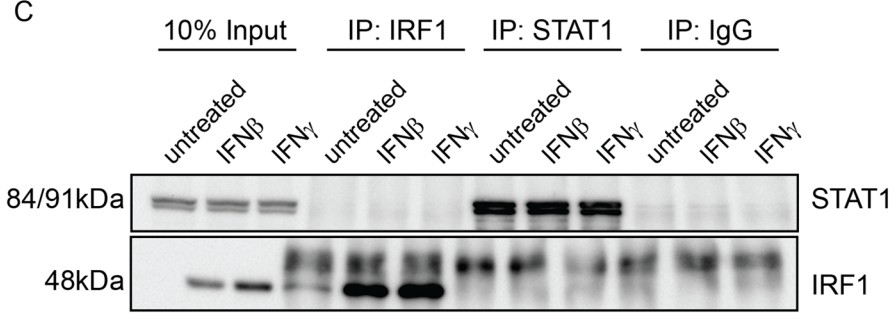

D

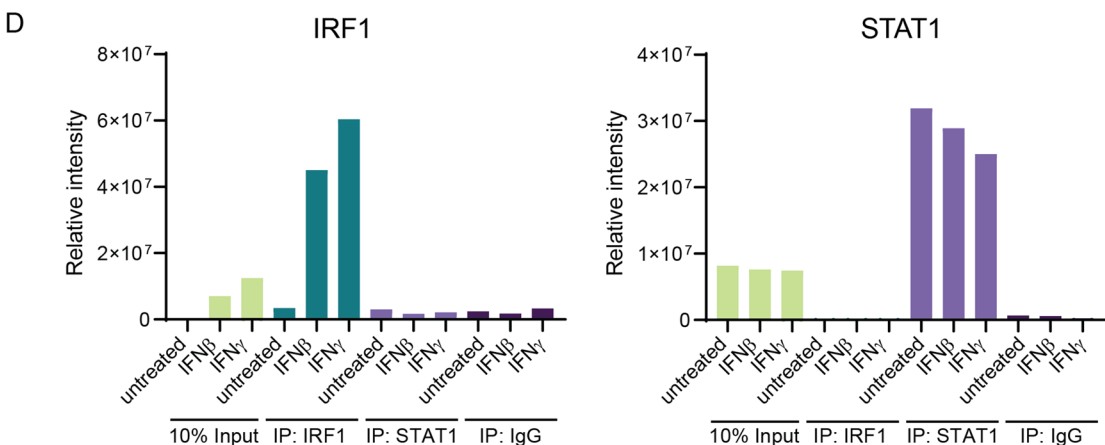

◀ **Figure EV5. Immunoprecipitation of IRF1 and STAT1.**

(A–D) BMDMs (**A**, **B**) and RAW 264.7 cells (**C**, **D**) were treated with IFNβ or IFNγ for 1.5 h. STAT1-IRF1 complexes were analyzed by immunoprecipitation (IP) using antibodies against IRF1, STAT1 or an IgG control, followed by western blotting ($n = 3$). Input controls represent 10% of the total lysate that was used for the IP. The representative blot in (**A**) was quantified using Image Lab (**B**). The representative blot in (**C**) was quantified using Image Lab (**D**). Source data are available online for this figure.

