## [Peer Review File · The EMBO Journal]

Dynamic control of gene expression by ISGF3 and IRF1 during IFN β and IFN γ signaling

Aarathy Ravi Sundar Jose Geetha, Katrin Fischer, Olga Babadei, Georg Smesnik, Alexander Vogt, Ekaterini Platanitis, Mathias Müller, Matthias Farlik, and Thomas Decker

Corresponding author(s): Thomas Decker (thomas.decker@univie.ac.at)

Review Timeline:

Submission Date:	20th Jun 23
Editorial Decision:	6th Jul 23
Revision Received:	15th Nov 23
Editorial Decision:	29th Jan 24
Revision Received:	6th Feb 24
Editorial Decision:	7th Mar 24
Revision Received:	11th Mar 24
Accepted:	13th Mar 24

Review
COMMONS

Editor: Cornelius Schneider

Transaction Report: This manuscript was transferred to The EMBO JOURNAL following peer review at Review Commons.

Review #1

1. Evidence, reproducibility and clarity:

Evidence, reproducibility and clarity (Required)

The authors examine the differences between the genes induced by type I IFNs and IFN γ by examining nascent transcripts over a prolonged period of time. The overall question being asked is very broad and the authors generate a massive amount of data that is hard to understand and interpret. The authors also fail to take into consideration the secondary genes induced by products of primary genes. For example, IFN β will induce other type I IFNs that will act in cis or trans to induce secondary ISGs. Also, it is not clear what effect cell death has on gene expression especially at later time points. The differential roles of ISFG3 (IRF9) and IRF1 are interesting but the biological meaning or outcomes of differences in gene expression at 24 and 48 hours is not entirely clear to this reviewer.

Major Issues:

- Figure 2. Difficult to interpret data as it is presented. Consider quantifying figure 2C in order to make "changes in Pol II pausing were more pronounced during IFN β signaling" statement more apparent.
- How are you distinguishing autocrine signaling in the BMDMs driven by IFN treatment from late transcripts (for example, at 48 hours are differential genes due to autocrine cytokine signaling or are they truly late transcripts)?
- Figure 3D. Authors choose Gbp2 (as positive control for IFN γ driven gene), but don't show that Gbp2 is a IFN β independent gene. Consider using IRF1 KO BMDMs in this data as well.

Minor Issues:

- Simplify figure 4B- consider focusing on most differentially expressed genes between clusters
- Define known IFN γ and IFN β driven genes when they are introduced in figure 2 rather than in discussion

- Clarify which cell types (IRF1 KO vs IRF9 KO) are used in figure 5 A/B.
- Unclear whether IRF1 expression in figure 3A is from whole cell lysate or nuclear fraction.
- Authors suggest IFN β treatment induces less IRF1 at later time points, however loading control also seems slightly lower than other considerations. Is it possible that IFN β treated cells are dying at later time points, given that type I IFN signaling can be pro-apoptotic.

2. Significance:

Significance (Required)

I accepted the request to review the paper based on the abstract and the idea that this was mechanistic investigation of the role of ISGF3 and IRF1 in regulation of genes induced by type I IFN and IFN γ . However, after thorough reading of the paper, I feel that I am not aptly qualified to evaluate all aspects of the manuscript. Many of the data are bioinformatic analysis and I do not have sufficient expertise to either understand the analysis or offer my interpretation of the conclusions drawn by the authors. I suggest that the best path forward is to find another suitable reviewer.. Hopefully, other reviewers have already offered you their suggestions to make an informed decision.

3. How much time do you estimate the authors will need to complete the suggested revisions:

Estimated time to Complete Revisions (Required)

(Decision Recommendation)

Cannot tell / Not applicable

No

Review #2

1. Evidence, reproducibility and clarity:

Evidence, reproducibility and clarity (Required)

- This interesting study addresses underlying molecular mechanisms that distinguish transcriptomes induced by type I and II IFNs. It is widely accepted that they induce distinct and overlapping genes. The IFN field has shown that type I IFNs can induce both ISGF3 (STAT1/STAT1/IRF9) and GAS (STAT1/STAT1) activation, while type II IFNs only induce GAS elements. The current study adds to some of these observations, including the cooperation of ISGF3 and IRF1 at later time points. They show that ISGF3 and IRF1 can affect enhancers and modify chromatin accessibility. While some of these observations are incremental, this study would significantly interest the interferon community.
- Biological significance is limited as this study is largely descriptive and they do not test the hits obtained from BioID.
- The sequencing and BioID data are not submitted to public databases.

2. Significance:

Significance (Required)

This interesting study addresses underlying molecular mechanisms that distinguish transcriptomes induced by type I and II IFNs. It is widely accepted that they induce distinct and overlapping genes. The IFN field has shown that type I IFNs can induce both ISGF3 (STAT1/STAT1/IRF9) and GAS (STAT1/STAT1) activation, while type II IFNs only induce GAS elements. The current study adds to some of these observations, including the cooperation of ISGF3 and IRF1 at later time points. They show that ISGF3 and IRF1 can affect enhancers and modify chromatin accessibility. While some of these observations are incremental, this study would significantly interest the interferon community.

While this reviewer does not suggest additional experimentation, this manuscript would be suitable as a resource paper.

My expertise is in innate immunity.

3. How much time do you estimate the authors will need to complete the suggested revisions:

Estimated time to Complete Revisions (Required)

(Decision Recommendation)

Between 3 and 6 months

Yes

Review #3

1. Evidence, reproducibility and clarity:

Evidence, reproducibility and clarity (Required)

****Summary:****

This manuscript by Geetha et colleagues addresses the differences and similarities in gene expression control between Type I IFN (IFN β) and Type II IFN (IFN γ) signaling. The authors aim to determine the factors responsible for the partitioning of

IFN β and IFN γ -induced transcriptomes and their propagation of diverse biological responses. The authors mention the JAK-STAT paradigm of IFN signaling, which posits that ISGF3 dominates transcriptional responses to IFN-I, whereas GAF is critical for the generation of an IFN γ -specific transcriptome. However, recent investigations suggest that the ISGF3 complex may also play a role in IFN γ signaling. The authors investigate the contributions of IRF1, ISGF3, and noncanonical versions of the ISGF3 complex to the transcriptome divergence produced by IFN-I or IFN γ signaling. They used nascent transcript sequencing to determine how ISGs expression are temporally controlled by the ISGF3 complex or IRF1 and show that temporal control of ISG expression includes transcription factor recruitment, enhancer activation, changes of chromatin accessibility, and control of RNA pol II pausing. The authors also investigate cooperativity between STAT1 and IRF1 correlates with different nuclear interactomes. In its current form, the manuscript suffers from a lack of independent experiment replications and nuance in the interpretation of the results.

****Major comments:****

1. In Fig. 1d is difficult to interpret and misleading for many reasons. First, the cluster numbering is disconnected from the cluster order; why not numbering them based on the hierarchical clustering and writing the cluster number besides the cluster itself? Second, having a 2-color gradient is misleading; negative values shouldn't be in the same color tone than the positive values. Third, the authors did not provide adequate rationale behind using only the top 1,000 most expressed gene? Why not using all the differentially expressed genes in at least one of the condition to provide a comprehensive analysis? Could this potentially lead to bias in the data, and is there any information lost by not using the - lower - expressed genes fraction? Fourth, it is not clear what the color scale is representing and how the data was transformed. Was a mean centering of the expression values of the log₂FC applied to the RNA-seq data to facilitate clustering? Mean centering and z-scoring is a common technique used to adjust expression data, but it can potentially exaggerate differences between samples. More information about the data and analysis should be provided, as it is difficult to determine whether this was a valid approach or not.

2. Fig 2c. The authors claim that RNA Pol II pausing is a major factor in controlling the dynamics of ISG transcription. However, they did not provide sufficient explanation of the results, and in all fairness there is not much variation between the clusters to sustain the claim that this is a major factor in ISG transcriptional control.

3. The large standard deviation bars in the claim that ChIP data confirmed the binding of ISGF3 components to the promoter of Mx2 cast doubt on the validity of the results and conclusions. The authors should consider additional experiments or

complementary analyses to validate their findings. Or alternative, to adjust their claims accordingly.

4. On p.5, the authors mention "Representative browser tracks from the Gbp2 and Slfn1 genes further validate this observation" but they are simply referring to genome browser snapshot, i.e., specific genomic examples, extracting from the same single dataset. Without using an independent dataset, this can not "further validate" the initial findings.

5. IRF1 was successfully pulled down with STAT1 bait but not in the reciprocal experiment. The author should discuss this point as it is important for the conclusions. Could it potentially indicate issues with the technique used, and if this could introduce any bias into the results. The statement, "In contrast, interactors of the IRF1 bait did not include STAT1. This discrepancy could result from steric constraints of the tagged proteins due to the limitation of the 10nm distance reached by the biotin ligase," does not seem to be sufficient to explain this discrepancy.

6. The authors interpret their ATAC-seq and ChIP-seq results based on a 2kb window to the TSS of genes, not considering relatively close enhancers or longer range cis-regulatory interactions in their interpretation. For example, they mention on p.7 "Contrasting the strong binding of IRF9 and IRF1 to the Mx2 (cluster 2) and Gbp2 (cluster 9) promoters, respectively, we saw no evidence for direct binding to Lrp11 (cluster 3) and Ptgs2 (cluster 10)", but on Fig 3d they show only the proximal regions. No scale bars are shown either. Moreover, exploring the same published IRF1 ChIP-seq dataset, there is a clear IRF1 binding site at the promoter of Ptgs2, while the authors report none.

7. Lack of statistical analysis on chromatin accessibility claims: The authors claim that ATAC-seq data in BMDMs stimulated with IFN β or IFN γ for a short (1.5 hours) or long (48 hours) period reveals a striking similarity between transcription and the general trends of chromatin accessibility at regions up to 1000 bp upstream of the TSS (Fig. 2a), suggesting continuous chromatin remodeling during the transcriptional response. However, I would like to know if this conclusion is well-supported by the correlation between the chromatin accessibility from ATAC-seq data from only one sample and the PRO-seq data. The need for additional experiments to verify claims such as the dependence of Ifi44 on IRF1 for gaining ATAC signal, as stated in the claim, "Expression required IRF1 for both, but accessibility of the Ifi44 regulatory region depended upon IRF1 whereas that of Gbp2 acquired an open structure independently of IRF1 (Fig. 5c)."

8. In the figure legends, there is missing information about the number of times

experiments were replicated, suggesting that some were done a single time. Moreover, some graphs are missing statistical analysis, e.g., in Fig S3cS3e, S3f, the ChIP-qPCR experiments were done on biological triplicates, there is no mention of statistical test performed, it is not mentioned what the error bars represents (SD, SEM, etc.) and the variance is large, but the authors still interpret these results as significant enrichment of the transcription factors to the Mx2 promoter. Another example are the RNA Pol II pausing index ratios, which show minor variations and not are supported by statistics to support a possible significance. Proper description, replication and statistical analyses of the results are critical.

9. The authors used CRISPR-Cas9 genome editing to generate knockout cell lines. However, they did not verify the knockouts at the protein level. Further experiments could confirm that the targeted proteins are not expressed in the knockout cell lines.

10. On p.9, it is mentioned "IRF1 affects chromatin structure ...". Here chromatin structure is related to minor changes in chromatin accessibility, this can not be qualified as changes in chromatin structure.

11. The authors have not adequately addressed the methodological limitations in their discussion, which extends beyond the aforementioned comments. It is suggested they include a comprehensive discussion of the claims made pertaining to the necessity of IRF1 for accessibility and the potential biases in the interactomes, along with their associated consequences.

12. Although the TurboID experiments identify known STAT1 and IRF1 interactors, the proposed new interactors are numerous and none are validate through independent co-IP experiments. Moreover, the results are very noisy, with little differences between untreated BMDMs (where IRF1 is barely expressed) and IFN-treated conditions.

13. The work should be discussed in the context of the demonstrated physiopathological evidence of the IRF1 and IRF9 functions. IRF9 (Hernandez et al., JEM 2018) and more recently IRF1 (Rosain et al Cell, 2023) were identified as causing non overlapping phenotypes in human patients carrying loss-of-function mutations for these genes. The authors must interpret their results in this context.

****Minor comments:****

- In most graphs the expression values or log2FC are shown separately for IFN β and IFN γ , however in the heatmaps (Fig 1d, S1d) the IFN β and IFN γ results are

intercalated keeping them side-by-side for each time point, which makes them more difficult to interpret. Suggestion to show the IFN β data first and followed by the IFN γ results.

- Fig 1e. The color scales on the GO enrichment graphs are misleading since they use the same blue-to-red gradient for adj p-values ranging from 10⁻²⁵ to 10⁻⁴⁹ and 0.008 to 0.016, which could be considered non significant.
- The inconsistency in the title referring to IFN β as Type 1 but using IFN γ instead of Type 2 nomenclature, perhaps consistency is best.
- The incomplete schema in Figure 1a, which only focuses on PRO-seq and does not include the ATAC-seq element.
- Figure 6d includes a color scale of -1 to +3, but it is unclear what these values represent and how they were calculated per interactor. The figure legend should be revised to clarify this information.
- The clearer labeling of Figure 5a and 5b.
- The statement that "IFN-I are the more important mediators of antiviral immunity" is not entirely accurate and may be an oversimplification, as there are certainly articles which suggest a larger role for type II IFN elements than type I (ref: Yamane D et al., 2019 Nature microbiology). While yes, IFN-I plays a critical role in the innate immune response to viral infections, IFN γ also has antiviral activity and is involved in the adaptive immune response to viral infections, and in some instances to a larger extent than IFN I.
- The authors claim that a significant portion of ISG promoters is associated with ISGF3 upon IFN γ receptor engagement and that the transcriptomes of macrophages treated briefly with IFN β or IFN γ exhibit remarkable similarity and sensitivity to Irf9 deletion. However, I am uncertain about the extent of consensus on this claim.
- Fig 1e, S1c. Graphs having circles of varying sizes in function of a value are named "bubble plots" and not "dot plots".
- Fig S1b, S3b. The PRO-seq was generated in triplicates, hence these graphs should include the Log₂FC for the individual data points.
- Fig S3c legend. It is mentioned "Graph represents RT-qPCR of genomic Mx2". RT-qPCR usually stands for reverse transcription quantitative PCR, hence we suggest to

change to "ChIP-qPCR" or qPCR. Confusingly, in the literature the term "RT-PCR" is used for real-time PCR and "qPCR" for quantitative PCR. Also, the authors should be specific about the "genomic" region targeted; the graphs mention "promoter", hence it would be appropriate to use the same designation in the legend.

- Fig S3e. The y-axis names are missing.
- In the genomic snapshot shown, only bars or fading triangles are shown in place of the gene body. The authors should provide an accurate gene structure; i.e., exons and introns.
- Raw cells are sometimes spelled as "Raw" and other times as "RAW". Please choose one for consistency.
- In p.10 1.20, the figure number is missing.

2. Significance:

Significance (Required)

Nature and significance of the advance:

The paper presents an investigation of the transcriptional response to IFN β and IFN γ in mouse bone marrow-derived macrophages and identifies key factors controlling the dynamics of interferon-stimulated gene (ISG) expression. The study employs cutting-edge technologies such as PRO-seq and ATAC-seq to assess transcriptional and chromatin accessibility changes, respectively. The results can potentially provide new insights into the transcriptional regulation of ISGs and the factors controlling their expression, which have significant implications for understanding the immune response to viral infection and cancer. Overall, the work could represent a conceptual advance in the field of immunology and epigenetics surrounding the transcriptional regulation of IFN, but validations and further mechanistic results are required.

Contextualization of the work:

The study builds on previous research on the transcriptional response to interferons but provides a more detailed and comprehensive investigation of the underlying mechanisms. Some of the key references that the authors build on include studies on the role of IRF9 in interferon signaling, the regulation of chromatin accessibility during immune activation, and the characterization of interferon-stimulated gene expression. However, the current study goes beyond these previous studies by

integrating multiple approaches to examine the transcriptional and epigenetic changes that occur during interferon signaling of two types, I and II.

Audience and potential impact:

The findings of the study are likely to be of interest to a wide range of researchers in the fields of immunology, molecular biology, and epigenetics, as well as those interested in the transcriptional regulation. The study may also be of interest to clinical researchers investigating the use of interferons as therapies for viral infections and cancer. The identification of factors controlling ISG expression may have implications for the development of new interferon-based therapies, as well as for understanding the mechanisms of resistance to interferon treatment in patients.

Field of expertise:

Overall, the study is contributing to our understanding of the differential transcriptional response to interferons and the factors controlling ISG expression. Upon provide further mechanistic demonstrations and validations, the work could have significant implications for both basic and clinical research.

3. How much time do you estimate the authors will need to complete the suggested revisions:

Estimated time to Complete Revisions (Required)

(Decision Recommendation)

More than 6 months

No

Revision Plan

Manuscript number: RC-2023-01889

Corresponding author(s): Thomas Decker

1. General Statements [optional]

Reply to general assessment of referee #2:

1. **General assessments:** The current study adds some to these observations...some of these observations are incremental...biological significance is limited. While this reviewer does not suggest additional experimentation, this manuscript would be suitable as a resource paper.

Reply: It appears we were not clear enough in explaining the novel aspects of our study. The starting points are two published studies from our lab demonstrating a global increase of ISGF3 association with ISG promoters in IFN γ -treated cells and a remarkable similarity of IFN- γ and type I IFN-induced early transcriptome changes. These findings challenge the notion in the field (as mentioned by the referee) that IFN γ specificity is produced by the predominant deployment of STAT1 homodimers. We thus tested the hypothesis that the specificity of the IFN γ -induced transcriptome is generated over time, rather than during the early response, and relies on secondary responses to transcription factors such as IRF1. In contrast, IRF1 plays no or only a small role in the type I IFN response that utilises ISGF3 and/or unknown secondary factors in the delayed response. We tested this hypothesis with PRO-seq technology to rule out confounding effects of mRNA processing over a 48h period. The data are clear in showing that many genes associated with the antibacterial or anti parasite profile of activated macrophages are indeed much more abundant in late-stage rather than briefly IFN γ -treated macrophages and these delayed changes are to a large extent dependent on IRF1. Our findings are based on the best available technologies, a combination of nascent transcript analysis with genetics and protein interaction studies. In addition, our findings rule out alternative models of sustained or secondary ISG transcription, such as the employment of alternative ISGF3 complexes (such as STAT2-IRF9) or of ISGF3 complexes formed with unphosphorylated STAT1 and STAT2. We provide evidence for higher order waves of transcription caused by unknown transcription factors that are produced by transcriptional activation of ISGF3 or IRF1 target genes and identify candidates among the AP1 and Ets transcription factor families. We agree that some of the data are confirmatory rather than novel (i.e. some of the genes we describe were known from previous literature to be IRF1 targets), but it is the systems approach of our study, and particularly the delineation of conditions under which the largely neglected delayed response diverts the IFN β and IFN γ -induced transcriptomes, that generates a comprehensive and conclusive view of IFN γ acting predominantly as a macrophage activating factor, and IFN β being an essential antiviral cytokine. We do think this main outcome is immunologically meaningful and not incremental. For this reason, we would prefer to publish the paper as a relevant contribution to innate immunology rather than a resource. Emphasizing our point, a paper appeared in 'Cell' while our study was under review, showing that human IRF1 mutations

cause mendelian susceptibility to mycobacterial disease (MSMD), a term coined by JL Casanova and colleagues for immunological defects that reduce the ability of macrophages to cope with intracellular bacteria (new ref. 65). This important study emphasizes the main conclusions of our study about the relevance of IRF1 for macrophage activation. We discuss this paper on p. 14 lines 9-14.

Revision: We tried to better explain the scientific motivation for this study and the significance of the results (p. 4, lines, lines 12-25).

Revision plan: n. a.

2. Description of the planned revisions

Referee #3; major comment 1:

In Fig. 1d is difficult to interpret and misleading for many reasons. First, the cluster numbering is disconnected from the cluster order; why not numbering them based on the hierarchical clustering and writing the cluster number besides the cluster itself? Second, having a 2-color gradient is misleading; negative values shouldn't be in the same color tone than the positive values. Third, the authors did not provide adequate rationale behind using only the top 1,000 most expressed gene? Why not using all the differentially expressed genes in at least one of the condition to provide a comprehensive analysis? Could this potentially lead to bias in the data, and is there any information lost by not using the - lower - expressed genes fraction? Fourth, it is not clear what the color scale is representing and how the data was transformed. Was a mean centering of the expression values of the log₂FC applied to the RNA-seq data to facilitate clustering? Mean centering and z-scoring is a common technique used to adjust expression data, but it can potentially exaggerate differences between samples. More information about the data and analysis should be provided, as it is difficult to determine whether this was a valid approach or not.

Reply:

- To create the heatmap, we used the pheatmap package from R and the cutree_rows option to separate 11 clusters with strikingly different patterns of gene expression based on visual exploration. The numbering was autogenerated by the program.
- The data is now shown in red-blue.
- We restricted our list to only 1000 genes from each comparison as we aimed to analyze the prominent patterns of gene expression across timepoints. Considering all differentially expressed genes based on a padj value would also include genes expressed at very low levels as evident from the low baseMean values obtained from DESeq2. Hence, we applied a selection of 1000 genes which effectively represented the major patterns of gene expression across timepoints.

Revision Plan

- Variance stabilized transformation was applied on read counts obtained from PRO-seq using the DESeq2 package. The transformed reads were z-score normalized and used for performing hierarchical clustering by the "Ward.D2" method using the pheatmap package in R. A total of 3126 genes were used for this analysis. 11 distinct clusters were defined using cutree_rows option. The color scale represents z-score normalized counts. The genes represented in the heatmap were selected based on the following criteria: each timepoint of interferon treatment was compared to the homeostatic condition (untreated sample) in wildtype BMDMs. The differentially expressed genes from each comparison were selected based on the filtering criteria: absolute log2FoldChange ≥ 1 and adjusted p value < 0.01 by Wald test. Following the differential analysis, the first 1000 differentially expressed genes in each treatment condition (ordered based on adjusted p values) were selected for both IFN types and combined and selected for creating a list which consisted of 3126 unique genes. The scale in the heatmap represents z-scores of variance-stabilized reads, calculated across all genotype and treatment conditions, separately for each IFN type.

Revision plan: We will label the clusters with the cluster number next to it in addition to the color codes.

Referee #3; major comment 3:

The large standard deviation bars in the claim that ChIP data confirmed the binding of ISGF3 components to the promoter of Mx2 cast doubt on the validity of the results and conclusions. The authors should consider additional experiments or complementary analyses to validate their findings. Or alternative, to adjust their claims accordingly.

Reply: To demonstrate sufficient quality of the data the ratio of Stat1/ Stat2 was calculated for early (1.5hrs) and late (48h) separately. The unpaired two-tailed t test comparing this ratio between 1.5 hrs and 48hs, shows that they are not significantly different. This indicates that all ISGF3 components are associated with ISG during both early and delayed responses, i. e., that STAT2/IRF9 complexes are unlikely to contribute to delayed ISG control. However, we agree with the referee that the standard deviations of the kinetic ChIP experiment are high and that it would be good to generate additional data.

Revision plan: We will perform additional ChIP experiments to improve the statistical power of the results in fig. S2c.

Referee #3, major comment 6:

The authors interpret their ATAC-seq and ChIP-seq results based on a 2kb window to the TSS of genes, not considering relatively close enhancers or longer range cis-regulatory interactions in their interpretation. For example, they mention on p.7 "Contrasting the strong binding of IRF9 and IRF1 to the Mx2 (cluster 2) and Gbp2 (cluster 9) promoters, respectively, we saw no evidence for direct binding to Lrp11 (cluster 3) and Ptgs2 (cluster 10)", but on Fig 3d they show only the proximal regions. No scale bars are shown either. Moreover, exploring the same

Revision Plan

published IRF1 ChIP-seq dataset, there is a clear IRF1 binding site at the promoter of *Ptgs2*, while the authors report none.

Reply:

- According to the literature (e. g. refs. 11, 27), most IFN-induced accessibility changes occur in the vicinity of the TSS of ISG. This is further strengthened by the data shown in this manuscript. In addition, most functionally validated GAS and ISRE sequences are in the DNA interval chosen for our analysis. While distal ISG enhancers have been reported (e. g. DOI: 10.26508/lsa.202201823), an analysis beyond the placement of most control regions increases the risk of wrong assignments between ISG and their regulatory elements, hence the causality between transcription factor binding and accessibility changes.
- We extended the regions for the analysis of the *Lrp11* and *Ptgs2* regulatory regions and found no evidence for the binding of ISGF3 or IRF1. We find no evidence for a clear peak in the *Ptgs2* promoter. There is a peak called by the Macs2 algorithm, but visual inspection of the track (bigwig file) shows it consists of a minor increase in reads above background that does not suggest a bona fide IRF1 binding site (see below). This view is supported by our inability to find an IRF binding site in the vicinity of the peak.

IRF1 binding indicated by bigWig browser tracks and corresponding peakfiles detected at the locus. We identified the peakfile from *Langlais et al.*, 2016 and identified peaks using MACS2, however using mm10 genome as the analysis in the original paper was done with mm9 genome. The peak identified here appears to be an artefact of the MACS2 program as there is no evident enrichment at the gene promoter region upon inspection of the bigWig files.

Revision plan: Scales will be added to the browser tracks as requested.

Referee #3, major comment 7:

Lack of statistical analysis on chromatin accessibility claims: The authors claim that ATAC-seq data in BMDMs stimulated with IFN β or IFN γ for a short (1.5 hours) or long (48 hours) period reveals a striking similarity between transcription and the general trends of chromatin accessibility at regions up to 1000 bp upstream of the TSS (Fig. 2a), suggesting continuous chromatin remodeling during the transcriptional response. However, I would like to know if this conclusion is well-supported by the correlation between the chromatin accessibility from ATAC-seq data from only one sample and the PRO-seq data.

Revision Plan

Reply: See revision plan.

Revision plan: We will analyze single experiments whether they support the conclusions derived from the z-score of the triplicate samples.

Referee #3, major comment 8:

The need for additional experiments to verify claims such as the dependence of Ifi44 on IRF1 for gaining ATAC signal, as stated in the claim, "Expression required IRF1 for both, but accessibility of the Ifi44 regulatory region depended upon IRF1 whereas that of Gbp2 acquired an open structure independently of IRF1 (Fig. 5c).

Reply: We think the lack of clarity might be related to the size of figures 5a and 5b and the density of the dots in some areas of the plot. We agree it is very difficult to assign our gene labels unambiguously to a single dot.

Fig. 5a combines ATACseq data in wt and IRF1 knockout cells with the expression data from the Pro-seq experiment, Fig. 5b is the same set-up, but IRF9-deficient macrophages are analyzed.

Blue dots show ATACseq signals induced by IFN treatment. Violet dots represent genes that require IRF1 (Fig. 5a) or IRF9 (Fig. 5b) for transcriptional induction. Yellow dots mark genes such as IFI44 requiring IRF1 (Fig. 5a) or IRF9 (Fig. 5b) for both expression and the accessibility change in the promoter region. Fig. 5c visualizes representative examples of genes whose accessibility is coupled to the transcription factor dependence of the transcriptional induction (IFI44), or not (Gbp2). Thus Fig. 5c must be interpreted based on the dot color code in fig. 5a and we admit this has been difficult with the figure in its present form.

Revision plan: We will improve the clarity of figs 5a and 5b in several ways:

- We will label the panels to better indicate the intersected data sets.
- We will increase the size of the panels and figure legends and make sure that the correspondence between gene names and dots are unambiguous.
- We will include trend lines of the Ifi44 and Gbp2 genes to visualize their induction and IRF1 dependence.

Referee #3, major comment 13 (see also section 3):

The authors have not adequately addressed the methodological limitations in their discussion, which extends beyond the aforementioned comments. It is suggested they include a comprehensive discussion of the claims made pertaining to the necessity of IRF1 for accessibility and the potential biases in the interactomes, along with their associated consequences.

Reply: The contribution of IRF1 to the accessibility of ISG promoters emerges from the data in figures 5a, whose clarity will be improved (see reply to point 8). We do not interpret the impact of IRF1 beyond the data, in fact we state a relatively minor effect of IRF1 in the control of

Revision Plan

promoter accessibility (p. 10, lines 20-22) and we have added a reference in agreement with an impact of IRF1 on basal expression of antiviral genes (ref. 39, as suggested by the referee).

We have added discussion on potential limitations of the TurboID approach (p. 11, lines 22-24 and p. 15, lines 3-11).

Revision plan: Improvement of fig 5a (see ref. #3, point 8).

Referee #3, minor comment 2

Fig 1e. The color scales on the GO enrichment graphs are misleading since they use the same blue-to-red gradient for adj p-values ranging from 10⁻²⁵ to 10⁻⁴⁹ and 0.008 to 0.016, which could be considered non significant.

Reply: We agree that this is confusing. It results from automated assignments of the color gradients by the software.

Revision plan: We will investigate possibilities to change color codes for different ranges of p values.

Referee #3, minor comment 4

The incomplete schema in Figure 1a, which only focuses on PRO-seq and does not include the ATAC-seq element.

Reply: We will add a new figure to visualize the set-up of the ATAC seq experiments and their intersection with the Pro-seq data.

Revision plan: We will add a new figure in accordance with the referee's request.

Referee #3, minor comment 6

The clearer labeling of Figure 5a and 5b.

Reply: Please refer to our reply to major point 8.

Referee #3, minor comment 10

Fig S1b, S3b. The PRO-seq was generated in triplicates, hence these graphs should include the Log₂FC for the individual data points.

Reply: The Log₂FC from DESeq2 were calculated from the triplicates, the software does not compute Log₂FC from individual replicates.

Revision plan: We mention the p-values for the Log₂FC to show the degree of consistency (figure legends). We will provide a table with log₂FC and corresponding padj values of the genes represented at each timepoint (table_showing_padj_values_and_log2fc).

Referee #3, minor comment 12

In the genomic snapshot shown, only bars or fading triangles are shown in place of the gene body. The authors should provide an accurate gene structure; i.e., exons and introns.

Reply: We will try to include the exon-intron structure wherever the size of the figure allows this.

Revision: n. a.

Revision plan: If figure size permits, we will add the exon-intron structure of the genes in browser tracks as requested.

3. Description of the revisions that have already been incorporated in the transferred manuscript

Referee #1, major comment 1

Figure 2. Difficult to interpret data as it is presented. Consider quantifying figure 2C in order to make "changes in *Pol II pausing* were more pronounced during IFN β signaling" statement more apparent.

Reply: We presented the pausing data in two different graphic representations (figures 2c and S2) to make the understanding of the information content easier. In hindsight we may have generated more confusion than clarity.

Revision: We removed the original figure 2c and replaced it with original figure S2. This representation is quite intuitive as the graphs represent a direct quantitative logarithmic display whether and how much the relative amount of paused polymerase changes when comparing IFN-treated and untreated cells. The calculation of these ratios is now explained better in the legend to figure 2.

Referee #1, major comment 2

How are you distinguishing autocrine signaling in the BMDMs driven by IFN treatment from late transcripts (for example, at 48 hours are differential genes due to autocrine cytokine signaling or are they truly late transcripts)?

Reply: We do not exclude autocrine effects. In case of ISG, the most likely autocrine factor would be secreted interferon. According to our Proseq data, the differentially expressed genes do not include any interferon genes. That being said, it is possible that the transcription factors from the AP1 family we hypothesize as drivers of secondary or tertiary waves of transcription are activated by non-IFN cytokines secreted from IFN-treated cells (see also reply to comment 3).

Revision Plan

Revision: We now mention that enhanced IFN production is not sustaining ISG responses (p.5 lines 18/20). We mention the possibility that secreted factors may drive secondary or tertiary waves of ISG transcription (p. 8, lines 21/23).

Referee #1, major comment 3

Figure 3D. Authors choose *Gbp2* (as positive control for IFN γ driven gene), but don't show that *Gbp2* is a IFN β independent gene. Consider using IRF1 KO BMDMs in this data as well.

Reply: This is a misunderstanding. *Gbp2* is not shown as an IFN γ -specific gene (it's induction by both IFN types has been shown previously and emerges from our Pro-seq analysis, see also response to minor issue no. 2). It represents the cluster of genes that are sustained specifically after IFN γ treatment in an IRF1-dependent manner. The purpose of fig. 3D is to show that not all ISGF3/IRF9-dependent genes have promoter binding sites for ISGF3 and not all IRF1-dependent genes have binding sites for IRF1. This suggests indirect effects of both transcription factors in sustaining IFN-induced transcription (in line with the referee's comment 1).

Previous figure S3e (now S2f) confirms binding of IRF1 to the GBP2 promoter by CHIP with kinetics correlating to its transcriptional effect. This experiment is normalized with an IgG control. IRF1 knockout cells did not produce a CHIP signal with IRF1 antibody, as expected (data not shown).

Revision: We better explain the rationale behind the experiments shown in figure 3D (text on p8, lines 12-16). In addition, we show the trend line of *Gbp2* expression in WT vs IRF1KO as well as that of additional genes showing delayed/sustained responses in the new Figure S3.

Referee #1, minor comment 2

Define known IFN γ and IFN β driven genes when they are introduced in figure 2 rather than in discussion.

Reply: Following the referee's suggestion we provide the examples of IFN β and IFN γ -controlled genes and the characteristics of their regulation in the context of our description of the results displayed by fig. 2 (p.6 lines 15-21). This includes *Gbp2* (see major issue no. 3).

Revision: The text on p. 6 lines 15-21 has been modified in accordance with the request.

Referee #1, minor comment 4

Unclear whether IRF1 expression in figure 3A is from whole cell lysate or nuclear fraction.

Reply: We indicate in the figure legend that whole cell lysates were used.

Revision: We added a sentence with the relevant information in the legend of figure 3.

Referee #1, minor comment 5

Revision Plan

Authors suggest IFN β treatment induces less IRF1 at later time points, however loading control also seems slightly lower than other considerations. Is it possible that IFN β treated cells are dying at later time points, given that type I IFN signaling can be pro-apoptotic.

Reply: The graph below the blot represents quantified IRF1 signals, normalized to the loading control. It shows that the differences are not generated by unequal loading of the blotted gel. We and others have shown that IFN β may indeed enhance macrophage death, however only when the cells are simultaneously infected with an intracellular pathogen (e.g. new ref. 25). These studies also show that treatment with IFN β alone over periods used in the present study does not affect macrophage viability.

Revision: We added a sentence about the viability of IFN-treated macrophages (p. 4, lines 31-32).

Revision plan: n. a.

Referee #2, major comment 3

The sequencing and BioID data are not submitted to public databases.

Reply: An accession number has been added.

Revision: The accession number was added on p.29, line 25.

Referee #3, major comment 1 (see also revision plan, section 2):

Revision: The rationale for using the top 1.000 genes is explained (p.5, lines 7-9). The description of the pro-seq read count processing has been extended in accordance with our reply to the referee in the legend of figure 1d and in the methods section (p. 33, lines following line 10.)

Referee #3, major comment 2

Fig 2c. The authors claim that RNA Pol II pausing is a major factor in controlling the dynamics of ISG transcription. However, they did not provide sufficient explanation of the results, and in all fairness there is not much variation between the clusters to sustain the claim that this is a major factor in ISG transcriptional control.

Reply: We agree with the referee that we cannot posit RNA pol II pausing as a major factor for the differences of transcriptional control of ISG in individual clusters. We have made sure to remove any statements suggesting this possibility. We also try to better integrate our findings with RNA pol II pausing into the existing literature.

Revision: We added relevant literature on p. 6 lines 28-30 and p. 7, lines 4-6.

Revision Plan

Referee #3, major comment 4

On p.5, the authors mention "Representative browser tracks from the Gbp2 and Slfn1 genes further validate this observation" but they are simply referring to genome browser snapshot, i.e., specific genomic examples, extracting from the same single dataset. Without using an independent dataset, this can not "further validate" the initial findings.

Reply: We agree the wording is incorrect.

Revision: We changed the paragraph describing this experiment (p. 6, lines 15-21).

Referee #3, major comment 5

IRF1 was successfully pulled down with STAT1 bait but not in the reciprocal experiment. The author should discuss this point as it is important for the conclusions. Could it potentially indicate issues with the technique used, and if this could introduce any bias into the results. The statement, "In contrast, interactors of the IRF1 bait did not include STAT1. This discrepancy could result from steric constraints of the tagged proteins due to the limitation of the 10nm distance reached by the biotin ligase," does not seem to be sufficient to explain this discrepancy.

Reply: STAT1 was present in the IRF1 pull-down and the interaction increased significantly after IFN treatment but after normalization to the NLS control it did not conform to our criterium of a 95% confidence interval for the FDR. To be consistent we did not include it in the list of IRF1 interactors. We have observed on several occasions that the significance of proximity is not reciprocal, even for well- documented physical interactions. A prime example for this is the interaction between STAT1 and IRF9 in IFN-treated cells which is recorded in the STAT1 pull-down, but not that with IRF9 (ref. 10). Apart from steric reasons the lack of reciprocity may result from different signal/noise ratios in pull downs with different baits.

Revision: We mention that IRF1 was a STAT1 interactor below the statistical cut-off (p. 11, lines 26-28) as well as the possibility of different signal/noise ratios in the IRF1 and STAT1 pull-downs on p.11, lines 22-24.

Referee #3, major comment 9

In the figure legends, there is missing information about the number of times experiments were replicated, suggesting that some were done a single time. Moreover, some graphs are missing statistical analysis, e.g., in Fig S3cS3e, S3f, the ChIP-qPCR experiments were done on biological triplicates, there is no mention of statistical test performed, it is not mentioned what the error bars represents (SD, SEM, etc.) and the variance is large, but the authors still interpret these results as significant enrichment of the transcription factors to the Mx2 promoter.

Reply: Where missing the relevant information has been added to figure legends. In brief, all experiments represent at least three biological replicates. The only exception is the western blot shown in figure S3a, (no S2a) which represents two independent replicates. Here, the clarity of the difference of IRF1 expression and the fact that the only purpose is to show that Raw264.7

Revision Plan

macrophages behave like bone marrow-derived macrophages in fig. 3a justifies the omission of another replicate (please see also answer to point 3).

Revision: The relevant information has been added to figure legends where necessary (figs. 1, a, 3a, 6a-f, S1, S4, S5).

Referee #3, major comment 10

Another example are the RNA Pol II pausing index ratios, which show minor variations and not are supported by statistics to support a possible significance. Proper description, replication and statistical analyses of the results are critical.

Reply: We agree.

Revision: Statistics underlying the RNA Pol II pausing data are included in supplementary data 2.

Referee #3, major comment 11

The authors used CRISPR-Cas9 genome editing to generate knockout cell lines. However, they did not verify the knockouts at the protein level. Further experiments could confirm that the targeted proteins are not expressed in the knockout cell lines.

Reply: We included a western blot showing the lack of IRF1 and STAT1 expression in the respective cell lines.

Revision: New figure S6.

Referee #3, major comment 12

On p.9, it is mentioned "IRF1 affects chromatin structure ...". Here chromatin structure is related to minor changes in chromatin accessibility, this can not be qualified as changes in chromatin structure.

Reply: 'structure' has been changed in accordance with the request.

Revision: 'structure' has been replaced with 'accessibility'. (p. 10, lines 19 and 21).

Referee #3, major comment 13 (see also section 2, revision plan, major comment 8)

The authors have not adequately addressed the methodological limitations in their discussion, which extends beyond the aforementioned comments. It is suggested they include a comprehensive discussion of the claims made pertaining to the necessity of IRF1 for accessibility and the potential biases in the interactomes, along with their associated consequences.

Reply: The contribution of IRF1 to the accessibility of ISG promoters emerges from the data in figures 5a, whose clarity will be improved (see reply to point 8). We do not interpret the impact

Revision Plan

of IRF1 beyond the data, in fact we state a relatively minor effect of IRF1 in the control of promoter accessibility (p. 10, lines 20-22) and we have added a reference in agreement with an impact of IRF1 on basal expression of antiviral genes (ref. 39, as suggested by the referee).

We have added discussion on potential limitations of the TurboID approach (p. 11, lines 22-24 and p. 15, lines 3-11).

Revision: Change of the discussion section (p. 11, lines 22-24 and p. 15, lines 3-11).

Revision plan: Improvement of fig 5a (see ref. #3, point 8).

Referee #3, major comment 15

The work should be discussed in the context of the demonstrated physiopathological evidence of the IRF1 and IRF9 functions. IRF9 (Hernandez et al., JEM 2018) and more recently IRF1 (Rosain et al Cell, 2023) were identified as causing non overlapping phenotypes in human patients carrying loss-of-function mutations for these genes. The authors must interpret their results in this context.

Reply: We thank the referee for reminding us about the importance of these papers for our work.

Revision: The papers have been mentioned and discussed (p. 13 lines 19-28 and p.14, lines 9-14).

Referee #3, minor comment 3

The inconsistency in the title referring to IFN β as Type 1 but using IFN γ instead of Type 2 nomenclature, perhaps consistency is best.

Reply: We agree about the importance of consistency but find ourselves in yet another quandary. While the use of 'type I IFN' is clearly indicated and widely used as a collective name for this group of cytokines, the use of 'type II IFN' for IFN γ is rare because it is the only member of this type. Hence, we decided for sticking with convention at the expense of a bit of consistency. We agree about the title, though, and have changed type I IFN to IFN β .

Revision: We adapted the title in agreement with the referee's comment.

Referee #3, minor comment 5

Figure 6d includes a color scale of -1 to +3, but it is unclear what these values represent and how they were calculated per interactor. The figure legend should be revised to clarify this information.

Reply: We agree. The relevant information has been added to the figure legend.

Revision: We added information (log₂FC with regard to the NLS control) to the legend of fig. 6d.

Referee #3, minor comment 9

Fig 1e, S1c. Graphs having circles of varying sizes in function of a value are named "bubble plots" and not "dot plots".

Reply: Thank you for pointing this out, we corrected our mistake.

Revision: We changed dot plot to bubble plot in legend to figure S1c.

Referee #3, minor comment 11

Fig S3c legend. It is mentioned "Graph represents RT-qPCR of genomic Mx2". RT-qPCR usually stands for reverse transcription quantitative PCR, hence we suggest to change to "ChIP-qPCR" or qPCR. Confusingly, in the literature the term "RT-PCR" is used for real-time PCR and "qPCR" for quantitative PCR. Also, the authors should be specific about the "genomic" region targeted; the graphs mention "promoter", hence it would be appropriate to use the same designation in the legend.

Reply: We agree and thank the referee for correction of the terminology.

Revision: We changed RT-PCR to qPCR throughout the manuscript. Moreover, we specifically refer to 'promoter region' as the amplified DNA.

Referee #3, minor comment 12

Fig S3e. The y-axis names are missing.

Reply: Thanks for spotting this.

Revision: The y axis in the figure received its proper label.

Referee #3, minor comment 14

Raw cells are sometimes spelled as "Raw" and other times as "RAW". Please choose one for consistency.

Revision: This inconsistency has been corrected

Referee #3, minor comment 15

In p.10 l.20, the figure number is missing.

Revision: We corrected this mistake.

4. Description of analyses that authors prefer not to carry out

Referee #1, minor comment 1

Simplify figure 4B- consider focusing on most differentially expressed genes between clusters

Reply: The purpose of fig. 4B is to provide a visual overview of the kinetics of eRNA transcription in response to both IFN types and of the effects of IRF9 and IRF1 knockouts. This information needs to be given to demonstrate the similarities and differences between the control of eRNA and the corresponding ISG transcripts in the different regulatory clusters (as shown in figs. 1d and 2a).

Simplifying the figure would mean to separate it according to time point, IFN type treatment or knock-out effect. We think this would require to mentally reassemble the figure to understand the interrelationships between these parameters. To our opinion the visual display of the data interrelationship in fig. 4B facilitates the appropriation of the information content.

Revision: n. a. - we hope our reasoning has become sufficiently clear.

Revision plan: n. a.

Referee #1, minor comment 3

Clarify which cell types (IRF1 KO vs IRF9 KO) are used in figure 5 A/B.

Reply: The cell type (bone marrow-derived macrophages) is mentioned in the first sentence of the figure legend. Since all experiments except the Bio-ID experiment were performed with this cell type we decided not to label each figure.

Revision: n. a.

Revision plan: n. a.

Referee #2, major comment 2 and referee #3, major comment 14

Ref #2: Biological significance is limited as this study is largely descriptive and they do not test the hits obtained from BioID.

Ref #3: Although the TurboID experiments identify known STAT1 and IRF1 interactors, the proposed new interactors are numerous, and none are validated through independent co-IP

Revision Plan

experiments. Moreover, the results are very noisy, with little differences between untreated BMDMs (where IRF1 is barely expressed) and IFN-treated conditions.

Reply: The big advantage of BioID or TurboID is the ability to score proximity and very transient interactions. Validating BioID hits with technologies such as coIP is not particularly useful as the two technologies will obviously produce different interactomes. In fact, we show in this manuscript that IRF1 and STAT1 show proximity, but they do not form a stable complex under co-IP conditions. This leaves genetic approaches (LOF or GOF) as alternatives. However, apart from the workload (> 100 genes would have to be knocked out or their products overexpressed), most of our hits are expected to produce very broad effects in such experiments, hard to interpret regarding ISGF3 and IRF1 activities.

In view of this situation, we publish exclusively the high confidence nuclear interactors identified in our screen: biological replicates were performed in triplicate, a stringent internal control (TurboID-NLS) was used, and a stringent statistical cut-off for high-confidence interactors (95% FDR between groups) was applied. We further account for the experimental situation by limiting interpretation of the data to confirmed molecular events. For example, STAT1 dimers and the ISGF3 complex are required for histone acetylation in response to IFN, and ISGF3 is known to contribute to the exchange of the H2AZ histone variant (refs 11, 14, 71, 72). Our data show that IRF1 contributes to promoter accessibility changes and this is in line with its proximity to a remodelling complex. Thus, the BioID data indeed validate previous findings. However, in agreement with the referee's comment, some of the data remain descriptive (such as the intriguing proximity of both STAT1 and IRF1 to nuclear products of ISG). To determine the importance of this molecular proximity is a major undertaking and beyond the scope of this study.

Revision: We added discussion to state the difficulty of validating TurboID-based interactions and the limitations of the TurboID experiments (p.15 lines 3-11).

Referee #3, minor comment 1

In most graphs the expression values or log₂FC are shown separately for IFN β and IFN γ , however in the heatmaps (Fig 1d, S1d) the IFN β and IFN γ results are intercalated keeping them side-by-side for each time point, which makes them more difficult to interpret.

Reply: We are in a quandary about the design of the figure. On the one hand our goal is to visualize gene clusters with distinct behaviors for each IFN type. For this purpose, it would be advantageous to separate the IFN types. On the other hand, we aim at showing similarities and differences between genes induced by each IFN type, for this purpose it is better to maintain the current sample order. While understanding the referee's point, we prefer to keep the figure as it is, because the suggested change will not increase its overall clarity.

Revision: n. a.

Revision plan: n. a.

Referee #3, minor comment 7

The statement that "IFN-I are the more important mediators of antiviral immunity" is not entirely accurate and may be an oversimplification, as there are certainly articles which suggest a larger role for type II IFN elements than type I (ref: Yamane D et al., 2019 Nature microbiology). While yes, IFN-I plays a critical role in the innate immune response to viral infections, IFN γ also has antiviral activity and is involved in the adaptive immune response to viral infections, and in some instances to a larger extent than IFN I.

Reply: The Yamane et al study (now mentioned on p 10, lines 22-25 and referenced) agrees with our findings because it shows that IRF1 contributes to the basal expression of an ISRE-driven ISG subset. Our statement about the predominant role of type I IFN versus IFN γ refers to genetic data in both humans (mainly Casanova's work including effects of autoantibodies against type I IFN, see also the paper about human STAT2 deficiency in the June 15th issue of the JCI, <https://doi.org/10.1172/JCI168321>) and mice (hundreds of papers) showing that disruption of type I IFN synthesis or response causes profound effects of antiviral immunity (i.e. resulting susceptibilities are first and foremost to viral pathogens) whereas susceptibilities as a consequence of disrupting the IFN γ pathway are first and foremost to intracellular nonviral pathogens such a mycobacteria. In fact, the term mendelian susceptibility to mycobacterial disease (MSMD) was coined by Casanova and colleagues to describe a variety of human mutations that include those of the IFN γ , but not the type I IFN pathway.

Maybe more importantly, the Rosain et al. paper mentioned by the referee which appeared in 'Cell' while our study was under review, shows that human IRF1 mutations also fall into the MSMD category (new ref. 65). In contrast, the authors did not observe diminished antiviral immunity. This emphasizes the main conclusions of our study about the relevance of IRF1 for macrophage activation. We discuss this paper on p 14. lines 9-14.

Obviously, this does not exclude a role of type I IFN in nonviral infection or of IFN γ in viral infection, in fact much of our own work has been dedicated to a role of type I IFN in infections with *L. monocytogenes*. Nevertheless, we think that in a generic statement about the difference between type I IFN and IFN γ it is correct to label the former as predominantly antiviral and the latter predominantly as a macrophage activating factor against nonviral, intracellular pathogens.

Revision: We added discussion of Rosain et al. (ref. 65) on p 14. lines 9-14.

Referee #3, minor comment 8

The authors claim that a significant portion of ISG promoters is associated with ISGF3 upon IFN γ receptor engagement and that the transcriptomes of macrophages treated briefly with IFN β or IFN γ exhibit remarkable similarity and sensitivity to Irf9 deletion. However, I am uncertain about the extent of consensus on this claim.

Revision Plan

Reply: The data were surprising but supported by ChIP-seq and RNA-seq in wt and IRF9 ko macrophages (ref 10). Data in a follow-up study (ref. 11) and in this manuscript support our original conclusion by demonstrating the impact of the IRF9 ko on IFN γ responses. Importantly, we don't claim this is true in all cell types, it may well depend on STAT/IRF9 expression levels and tonic IFN signaling.

Revision: n. a.

Revision plan: n. a.

Dear Dr. Decker,

Thank you for submitting your manuscript (EMBOJ-2023-114806; RC-2023-01889) for consideration by the EMBO Journal together with the point-by-point response to the concerns raised during peer-review at Review Commons. I have now carefully read the manuscript, assessed the proposed revision plan, and I also discussed the matter within the editorial team. In addition, we have consulted on the work with an external expert advisor.

We appreciate your findings on the temporal kinetics, as well as magnitude of the response in INF signalling and how it determines the host response to pathogen infection. Also, we agree with the referees that the manuscript addresses a relevant question and proposes an interesting hypothesis. However, we also find that the comments of referee #3 are particularly insightful and raise several major concerns. These include the need for more rigorous support for various core claims including the requirement for additional replication experiments and the request for a more complete annotation of the results. Based on the interest expressed in the reports, together with positive endorsement by an external advisor, I am overall happy to invite you to address these issues in a revised version of the manuscript along the lines sketched in your revision plan. I would like to emphasize that we require i.p. referee #3's concerns to be comprehensively addressed in order to proceed towards publication at the EMBO Journal. I should also add that it is The EMBO Journal policy to allow only a single major round of revision and that it is therefore important to resolve the main concerns at this stage. I would be happy to discuss the revision in more detail via email or phone/videoconferencing.

We generally allow three months as standard revision time, which can be extended to six months in the case of major revisions. As a matter of policy, competing manuscripts published during this period will not negatively impact on our assessment of the conceptual advance presented by your study. However, please contact me as soon as possible upon publication of any related work to discuss the appropriate course of action. Should you foresee a problem in meeting this deadline, please let us know in advance to discuss an extension.

When preparing your letter of response to the referees' comments, please bear in mind that this will form part of the Review Process File and will therefore be available online to the community. For more details on our Transparent Editorial Process, please visit our website: <https://www.embopress.org/page/journal/14602075/authorguide#transparentprocess>. Please also see the attached instructions for further guidelines on preparation of the revised manuscript.

Please feel free to contact me if you have any further questions regarding the revision. Thank you for the opportunity to consider your work for publication. I look forward to discussing your revision.

With best regards,

Cornelius Schneider

Cornelius Schneider, PhD
Editor
The EMBO Journal
c.schneider@embojournal.org

We realize that it is difficult to revise to a specific deadline. In the interest of protecting the conceptual advance provided by the work, we recommend a revision within 3 months (4th Oct 2023). Please discuss the revision progress ahead of this time with the editor if you require more time to complete the revisions. Use the link below to submit your revision:

Link Not Available

Rev_Com_number: RC-2023-01889

New_manu_number: EMBOJ-2023-114806

Corr_author: Decker

Title: Dynamic control of gene expression by ISGF3 and IRF1 during IFN β and IFN γ signaling

Replies to reviewer's comments

Table of content

A Response to referees of RC submission

B Response to referees according to the submitted revision plan

C Material for referee 3

A Response to referees of RC submission

Reviewer #1

We thank the reviewer for her/his helpful comments.

Evidence, reproducibility and clarity

Major issues:

1. Figure 2. Difficult to interpret data as it is presented. Consider quantifying figure 2C in order to make "changes in Pol II pausing were more pronounced during IFN β signaling" statement more apparent.

Reply: We presented the pausing data in two different graphic representations (figures 2c and S2) to make the understanding of the information content easier. In hindsight we may have generated more confusion than clarity.

Revision: We removed the original figure 2c and replaced it with original figure S2. This representation is quite intuitive as the graphs represent a direct quantitative logarithmic display whether and how much the relative amount of paused polymerase changes when comparing IFN-treated and untreated cells. The calculation of these ratios is now explained better in the legend to figure 2.

Revision plan: n. a.

2. How are you distinguishing autocrine signaling in the BMDMs driven by IFN treatment from late transcripts (for example, at 48 hours are differential genes due to autocrine cytokine signaling or are they truly late transcripts)?

Reply: We do not exclude autocrine effects. In case of ISG, the most likely autocrine factor would be secreted interferon. According to our Proseq data, the differentially expressed genes do not include any interferon genes. That being said, it is possible that the transcription factors from the AP1 family we hypothesize as drivers of secondary or tertiary waves of transcription are activated by non-IFN cytokines secreted from IFN-treated cells (see also reply to comment 3).

Revision: We now mention that enhanced IFN production is not sustaining ISG responses (p.5 lines 18/20). We mention the possibility that secreted factors may drive secondary or tertiary waves of ISG transcription (p. 8, lines 21/23).

Revision plan: n. a.

3. Figure 3D. Authors choose Gbp2 (as positive control for IFN γ driven gene), but don't show that Gbp2 is a IFN β independent gene. Consider using IRF1 KO BMDMs in this data as well.

Reply: This is a misunderstanding. Gbp2 is not shown as an IFN γ -specific gene (it's induction by both IFN types has been shown previously and emerges from our Pro-seq analysis, see also response to minor issue no. 2). It represents the cluster of genes that are sustained specifically after IFN γ treatment in an IRF1-dependent manner. The purpose of fig. 3D is to show that not all ISGF3/IRF9-dependent genes have promoter binding sites for ISGF3 and not all IRF1-dependent genes have binding sites for IRF1. This suggests indirect effects of both transcription factors in sustaining IFN-induced transcription (in line with the referee's comment 1).

Previous figure S3e (now S2f) confirms binding of IRF1 to the GBP2 promoter by ChIP with kinetics correlating to its transcriptional effect. This experiment is normalized with an IgG control. IRF1 knockout cells did not produce a ChIP signal with IRF1 antibody, as expected (data not shown).

Revision: We better explain the rationale behind the experiments shown in figure 3D (text on p8, lines 12-16). In addition, we show the trend line of *Gbp2* expression in WT vs IRF1KO as well as that of additional genes showing delayed/sustained responses in the new Figure S3.

Revision plan: n. a.

Minor issues:

1. Simplify figure 4B- consider focusing on most differentially expressed genes between clusters

Reply: The purpose of fig. 4B is to provide a visual overview of the kinetics of eRNA transcription in response to both IFN types and of the effects of IRF9 and IRF1 knockouts. This information needs to be given to demonstrate the similarities and differences between the control of eRNA and the corresponding ISG transcripts in the different regulatory clusters (as shown in figs. 1d and 2a).

Simplifying the figure would mean to separate it according to time point, IFN type treatment or knock-out effect. We think this would require to mentally reassemble the figure to understand the interrelationships between these parameters. To our opinion the visual display of the data interrelationship in fig. 4B facilitates the appropriation of the information content.

Revision: n. a. - we hope our reasoning has become sufficiently clear.

Revision plan: n. a.

2. Define known IFN γ and IFN β driven genes when they are introduced in figure 2 rather than in discussion.

Reply: Following the referee's suggestion we provide the examples of IFN β and IFN γ -controlled genes and the characteristics of their regulation in the context of our description of the results displayed by fig. 2 (p.6 lines 15-21). This includes Gbp2 (see major issue no. 3).

Revision: The text on p. 6 lines 15-21 has been modified in accordance with the request.

Revision plan: n. a.

3. Clarify which cell types (IRF1 KO vs IRF9 KO) are used in figure 5 A/B.

Reply: The cell type (bone marrow-derived macrophages) is mentioned in the first sentence of the figure legend. Since all experiments except the Bio-ID experiment were performed with this cell type we decided not to label each figure.

Revision: n. a.

Revision plan: n. a.

4. Unclear whether IRF1 expression in figure 3A is from whole cell lysate or nuclear fraction.

Reply: We indicate in the figure legend that whole cell lysates were used.

Revision: We added a sentence with the relevant information in the legend of figure 3.

Revision plan: n. a.

5. Authors suggest IFN β treatment induces less IRF1 at later time points, however loading control also seems slightly lower than other considerations. Is it possible that IFN β treated cells are dying at later time points, given that type I IFN signaling can be pro-apoptotic.

Reply: The graph below the blot represents quantified IRF1 signals, normalized to the loading control. It shows that the differences are not generated by unequal loading of the blotted gel. We and others have shown that IFN β may indeed enhance macrophage death, however only when the cells are simultaneously infected with an intracellular pathogen (e.g. new ref. 25). These studies also show that treatment with IFN β alone over periods used in the present study does not affect macrophage viability.

Revision: We added a sentence about the viability of IFN-treated macrophages (p. 4, lines 31-32).

Revision plan: n. a.

Reviewer #2

We thank the referee for comments that allow us to improve our manuscript.

Evidence, reproducibility and clarity

1. **General assessments:** The current study adds some to these observations...some of these observations are incremental...biological significance is limited. While this reviewer does not suggest additional experimentation, this manuscript would be suitable as a resource paper.

Reply: It appears we were not clear enough in explaining the novel aspects of our study. The starting points are two published studies from our lab demonstrating a global increase of ISGF3 association with ISG promoters in IFN γ -treated cells and a remarkable similarity of IFN- γ and type I IFN-induced early transcriptome changes. These findings challenge the notion in the field (as mentioned by the referee) that IFN γ specificity is produced by the predominant deployment of STAT1 homodimers. We thus tested the hypothesis that the specificity of the IFN γ -induced transcriptome is generated over time, rather than during the early response, and relies on secondary responses to transcription factors such as IRF1. In contrast, IRF1 plays no or only a small role in the type I IFN response that utilises ISGF3 and/or unknown secondary factors in the delayed response. We tested this hypothesis with PRO-seq technology to rule out confounding effects of mRNA processing over a 48h period. The data are clear in showing that many genes associated with the antibacterial or anti parasite profile of activated macrophages are indeed much more abundant in late-stage rather than briefly IFN γ -treated macrophages and these delayed changes are to a large extent dependent on IRF1. Our findings are based on the best available technologies, a combination of nascent transcript analysis with genetics and protein interaction studies. In addition, our findings rule out alternative models of sustained or secondary ISG transcription, such as the employment of alternative ISGF3 complexes (such as STAT2-IRF9) or of ISGF3 complexes formed with unphosphorylated STAT1 and STAT2. We provide evidence for higher order waves of transcription caused by unknown transcription factors that are produced by transcriptional activation of ISGF3 or IRF1 target genes and identify candidates among the AP1 and Ets transcription factor families. We agree that some of the data are confirmatory rather than novel (i.e. some of the genes we describe were known from previous literature to be IRF1 targets), but it is the systems approach of our study, and particularly the delineation of conditions under which the largely neglected delayed response diverts the IFN β and IFN γ -induced transcriptomes, that generates a comprehensive and conclusive view of IFN γ acting predominantly as a macrophage activating factor, and IFN β being an essential antiviral cytokine. We do think this main outcome is immunologically meaningful and not incremental. For this reason, we would prefer to publish the paper as a relevant contribution to innate immunology rather than a resource. Emphasizing our point, a paper appeared in 'Cell' while our study was under review, showing that human IRF1 mutations cause mendelian susceptibility to mycobacterial disease (MSMD), a term coined by JL Casanova and colleagues for immunological defects that reduce the ability of macrophages to cope with intracellular bacteria (new ref. 65). This important study emphasizes the main conclusions of our study about the relevance of IRF1 for macrophage activation. We discuss this paper on p. 14 lines 9-14.

Revision: We tried to better explain the scientific motivation for this study and the significance of the results (p. 4, lines, lines 12-25).

Revision plan: n. a.

2. Biological significance is limited as this study is largely descriptive and they do not test the hits obtained from BioID.

The big advantage of BioID or TurboID is the ability to score proximity and very transient interactions. Validating BioID hits with technologies such as coIP is not particularly useful as the two technologies will obviously produce different interactomes. In fact, we show in this manuscript that IRF1 and STAT1 show proximity, but they do not form a stable complex under co-IP conditions. This leaves genetic approaches (LOF or GOF) as alternatives. However, apart from the workload (> 100 genes would have to be knocked out or their products overexpressed), most of our hits are expected to produce very broad effects in such experiments, hard to interpret regarding ISGF3 and IRF1 activities.

In view of this situation, we publish exclusively the high confidence nuclear interactors identified in our screen: biological replicates were performed in triplicate, a stringent internal control (TurboID-NLS) was used, and a stringent statistical cut-off for high-confidence interactors (95% FDR between groups) was applied. We further account for the experimental situation by limiting interpretation of the data to confirmed molecular events. For example, STAT1 dimers and the ISGF3 complex are required for histone acetylation in response to IFN, and ISGF3 is known to contribute to the exchange of the H2AZ histone variant (refs 11, 14, 71, 72). Our data show that IRF1 contributes to promoter accessibility changes and this is in line with its proximity to a remodelling complex. Thus, the BioID data indeed validate previous findings. However, in agreement with the referee's comment, some of the data remain descriptive (such as the intriguing proximity of both STAT1 and IRF1 to nuclear products of ISG). To determine the importance of this molecular proximity is a major undertaking and beyond the scope of this study.

Revision: We added discussion to state the difficulty of validating TurboID-based interactions and the limitations of the TurboID experiments (p.15 lines 3-11).

Revision plan: n. a.

3. The sequencing and BioID data are not submitted to public databases.

Reply: An accession number has been added.

Revision: The accession number was added on p.29, line 25.

Revision plan: n. a.

Reviewer #3

We thank the reviewer for her/his meticulous and insightful assessment of our work.

Evidence, reproducibility and clarity:

Major comments:

1. In Fig. 1d is difficult to interpret and misleading for many reasons. First, the cluster numbering is disconnected from the cluster order; why not numbering them based on the hierarchical clustering and writing the cluster number besides the cluster itself? Second, having a 2-color gradient is misleading; negative values shouldn't be in the same color tone than the positive values. Third, the authors did not provide adequate rationale behind using only the top 1,000 most expressed gene? Why not using all the differentially expressed genes in at least one of the condition to provide a comprehensive analysis? Could this potentially lead to bias in the data, and is there any information lost by not using the - lower - expressed genes fraction? Fourth, it is not clear what the color scale is representing and how the data was transformed. Was a mean centering of the expression values of the log2FC applied to the RNA-seq data to facilitate clustering? Mean centering and z-scoring is a common technique used to adjust expression data, but it can potentially exaggerate differences between samples. More information about the data and analysis should be provided, as it is difficult to determine whether this was a valid approach or not.

Reply:

- To create the heatmap, we used the pheatmap package from R and the cutree_rows option to separate 11 clusters with strikingly different patterns of gene expression based on visual exploration. The numbering was autogenerated by the program.
- The data is now shown in red-blue.
- We restricted our list to only 1000 genes from each comparison as we aimed to analyze the prominent patterns of gene expression across timepoints. Considering all differentially expressed genes based on a padj value would also include genes expressed at very low levels as evident from the low baseMean values obtained from DESeq2. Hence, we applied a selection of 1000 genes which effectively represented the major patterns of gene expression across timepoints.
- Variance stabilized transformation was applied on read counts obtained from PRO-seq using the DESeq2 package. The transformed reads were z-score normalized and used for performing hierarchical clustering by the "Ward.D2" method using the pheatmap package in R. A total of 3126 genes were used for this analysis. 11 distinct clusters were defined using cutree_rows option. The color scale represents z-score normalized counts. The genes represented in the heatmap were selected based on the following criteria: each timepoint of interferon treatment was compared to the homeostatic condition (untreated sample) in wildtype BMDMs. The differentially expressed genes from each comparison were selected based on the filtering criteria: absolute log2FoldChange ≥ 1 and adjusted p value < 0.01 by Wald test. Following the differential analysis, the first 1000 differentially expressed genes in each treatment condition (ordered based on adjusted p values) were selected for both IFN types and combined and selected for creating a list which consisted of 3126 unique genes. The scale in the heatmap represents z-scores of variance-stabilized reads, calculated across all genotype and treatment conditions, separately for each IFN type.

Revision: The rationale for using the top 1.000 genes is explained (p.5, lines 7-9). The description of the pro-seq read count processing has been extended in accordance with our reply to the referee in the legend of figure 1d and in the methods section (p. line)

Revision plan: We will label the clusters with the cluster number next to it in addition to the color codes.

2. Fig 2c. The authors claim that RNA Pol II pausing is a major factor in controlling the dynamics of ISG transcription. However, they did not provide sufficient explanation of the results, and in all fairness there is not much variation between the clusters to sustain the claim that this is a major factor in ISG transcriptional control.

Reply: We agree with the referee that we cannot posit RNA pol II pausing as a major factor for the differences of transcriptional control of ISG in individual clusters. We have made sure to remove any statements suggesting this possibility. We also try to better integrate our findings with RNA pol II pausing into the existing literature.

Revision: We added relevant literature on p. 6 lines 28-30 and p. 7, lines 4-6.

Revision plan: n. a.

3. The large standard deviation bars in the claim that ChIP data confirmed the binding of ISGF3 components to the promoter of Mx2 cast doubt on the validity of the results and conclusions. The authors should consider additional experiments or complementary analyses to validate their findings. Or alternative, to adjust their claims accordingly.

Reply: To demonstrate sufficient quality of the data the ratio of Stat1/ Stat2 was calculated for early (1.5hrs) and late (48h) separately. The unpaired two-tailed t test comparing this ratio between 1.5 hrs and 48hs, shows that they are not significantly different. This indicates that all ISGF3 components are associated with ISG during both early and delayed responses, i. e., that STAT2/IRF9 complexes are unlikely to contribute to delayed ISG control. However, we agree with the referee that the standard deviations of the kinetic ChIP experiment are high and that it would be good to generate additional data.

Revision: n. a.

Revision plan: We will perform additional ChIP experiments to improve the statistical power of the results in fig. S2c.

4. On p.5, the authors mention "Representative browser tracks from the Gbp2 and Sifn1 genes further validate this observation" but they are simply referring to genome browser snapshot, i.e., specific genomic examples, extracting from the same single dataset. Without using an independent dataset, this can not "further validate" the initial findings.

Reply: We agree the wording is incorrect.

Revision: We changed the paragraph describing this experiment (p. 6, lines 15-21).

Revision plan: n. a.

5. IRF1 was successfully pulled down with STAT1 bait but not in the reciprocal experiment. The author should discuss this point as it is important for the conclusions. Could it

potentially indicate issues with the technique used, and if this could introduce any bias into the results. The statement, "In contrast, interactors of the IRF1 bait did not include STAT1. This discrepancy could result from steric constraints of the tagged proteins due to the limitation of the 10nm distance reached by the biotin ligase," does not seem to be sufficient to explain this discrepancy.

Reply: IRF1 was present in the STAT1 pull-down, clearly above background, but the interaction did not conform to our criterium of a 95% confidence interval for the FDR. To be consistent we did not include it in the list of IRF1 interactors. We have observed on several occasions that the significance of proximity is not reciprocal, even for well- documented physical interactions. A prime example for this is the interaction between STAT1 and IRF9 in IFN-treated cells which is recorded in the STAT1 pull-down, but not that with IRF9 (ref. 10). Apart from steric reasons the lack of reciprocity may result from different signal/noise ratios in pull downs with different baits.

Revision: We mention that IRF1 was a STAT1 interactor below the statistical cut-off (p. 11, lines 26-28) as well as the possibility of different signal/noise ratios in the IRF1 and STAT1 pull-downs on p.11, lines 22-24.

Revision plan: n. a.

6. The authors interpret their ATAC-seq and ChIP-seq results based on a 2kb window to the TSS of genes, not considering relatively close enhancers or longer range cis-regulatory interactions in their interpretation. For example, they mention on p.7 "Contrasting the strong binding of IRF9 and IRF1 to the Mx2 (cluster 2) and Gbp2 (cluster 9) promoters, respectively, we saw no evidence for direct binding to Lrp11 (cluster 3) and Ptgs2 (cluster 10)", but on Fig 3d they show only the proximal regions. No scale bars are shown either. Moreover, exploring the same published IRF1 ChIP-seq dataset, there is a clear IRF1 binding site at the promoter of Ptgs2, while the authors report none.

Reply:

- According to the literature (e. g. refs. 11, 27), most IFN-induced accessibility changes occur in the vicinity of the TSS of ISG. This is further strengthened by the data shown in this manuscript. In addition, most functionally validated GAS and ISRE sequences are in the DNA interval chosen for our analysis. While distal ISG enhancers have been reported (e. g. DOI: 10.26508/lsa.202201823), an analysis beyond the placement of most control regions increases the risk of wrong assignments between ISG and their regulatory elements, hence the causality between transcription factor binding and accessibility changes.
- We extended the regions for the analysis of the Lrp11 and Ptgs2 regulatory regions and found no evidence for the binding of ISGF3 or IRF1. We find no evidence for a clear peak in the Ptgs2 promoter. There is a peak called by the Macs2 algorithm, but visual inspection of the track (bigwig file) shows it consists of a minor increase in reads above background that does not suggest a bona fide IRF1 binding site (see below).

IRF1 binding indicated by bigWig browser tracks and corresponding peakfiles detected at the locus. We identified the peakfile from *Langlais et al.*, 2016 and identified peaks using MACS2, however using mm10 genome as the analysis in the original paper was done with mm9 genome. The peak identified here appears to be an artefact of the MACS2 program as there is no evident enrichment at the gene promoter region upon inspection of the bigWig files.

Revision: n. a.

Revision plan: Scales will be added to the browser tracks.

7. Lack of statistical analysis on chromatin accessibility claims: The authors claim that ATAC-seq data in BMDMs stimulated with IFN β or IFN γ for a short (1.5 hours) or long (48 hours) period reveals a striking similarity between transcription and the general trends of chromatin accessibility at regions up to 1000 bp upstream of the TSS (Fig. 2a), suggesting continuous chromatin remodeling during the transcriptional response. However, I would like to know if this conclusion is well-supported by the correlation between the chromatin accessibility from ATAC-seq data from only one sample and the PRO-seq data.

Reply: See revision plan.

Revision: n. a.

Revision plan: We will analyze single experiments whether they support the conclusions derived from the z-score of the triplicate samples.

8. The need for additional experiments to verify claims such as the dependence of *lfi44* on IRF1 for gaining ATAC signal, as stated in the claim, "Expression required IRF1 for both, but accessibility of the *lfi44* regulatory region depended upon IRF1 whereas that of *Gbp2* acquired an open structure independently of IRF1 (Fig. 5c).

Reply: We think the lack of clarity might be related to the size of figures 5a and 5b and the density of the dots in some areas of the plot. We agree it is very difficult to assign our gene labels unambiguously to a single dot.

Fig. 5a combines ATACseq data in wt and IRF1 knockout cells with the expression data from the Pro-seq experiment, Fig. 5b is the same set-up, but IRF9-deficient macrophages are analyzed.

Blue dots show ATACseq signals induced by IFN treatment. Violet dots represent genes that require IRF1 (Fig. 5a) or IRF9 (Fig. 5b) for transcriptional induction. Yellow dots mark genes

such as IFI44 requiring IRF1 (Fig. 5a) or IRF9 (Fig. 5b) for both expression and the accessibility change in the promoter region. Fig. 5c visualizes representative examples of genes whose accessibility is coupled to the transcription factor dependence of the transcriptional induction (IFI44), or not (Gbp2). Thus Fig. 5c must be interpreted based on the dot color code in fig. 5a and we admit this has been difficult with the figure in its present form.

Revision: n. a.

Revision plan: We will improve the clarity of figs 5a and 5b in several ways:

- We will label the panels to better indicate the intersected data sets.
- We will increase the size of the panels and figure legends and make sure that the correspondence between gene names and dots are unambiguous.
- We will include trend lines of the Ifi44 and Gbp2 genes to visualize their induction and IRF1 dependence.

9. In the figure legends, there is missing information about the number of times experiments were replicated, suggesting that some were done a single time. Moreover, some graphs are missing statistical analysis, e.g., in Fig S3cS3e, S3f, the ChIP-qPCR experiments were done on biological triplicates, there is no mention of statistical test performed, it is not mentioned what the error bars represents (SD, SEM, etc.) and the variance is large, but the authors still interpret these results as significant enrichment of the transcription factors to the Mx2 promoter.

Reply: Where missing the relevant information has been added to figure legends. In brief, all experiments represent at least three biological replicates. The only exception is the western blot shown in figure S3a, (no S2a) which represents two independent replicates. Here, the clarity of the difference of IRF1 expression and the fact that the only purpose is to show that Raw264.7 macrophages behave like bone marrow-derived macrophages in fig. 3a justifies the omission of another replicate (please see also answer to point 3).

Revision: The relevant information has been added to figure legends where necessary (figs. 1, a, 3a, 6a-f, S1, S4, S5).

Revision plan: n. a.

10. Another example are the RNA Pol II pausing index ratios, which show minor variations and not are supported by statistics to support a possible significance. Proper description, replication and statistical analyses of the results are critical.

Reply: We agree.

Revision: Statistics underlying the RNA Pol II pausing data are included in supplementary data 2.

Revision plan: n. a.

11. The authors used CRISPR-Cas9 genome editing to generate knockout cell lines. However, they did not verify the knockouts at the protein level. Further experiments could confirm that the targeted proteins are not expressed in the knockout cell lines.

Reply: We included a western blot showing the lack of IRF1 and STAT1 expression in the respective cell lines.

Revision: New figure S6.

Revision plan: n. a.

12. On p.9, it is mentioned "IRF1 affects chromatin structure ...". Here chromatin structure is related to minor changes in chromatin accessibility, this can not be qualified as changes in chromatin structure.

Reply: 'structure' has been changed in accordance with the request.

Revision: ,structure' has been replaced with 'accessibility'. (p. 10, lines 19 and 21).

Revision plan: n. a.

13. The authors have not adequately addressed the methodological limitations in their discussion, which extends beyond the aforementioned comments. It is suggested they include a comprehensive discussion of the claims made pertaining to the necessity of IRF1 for accessibility and the potential biases in the interactomes, along with their associated consequences.

Reply: The contribution of IRF1 to the accessibility of ISG promoters emerges from the data in figures 5a, whose clarity will be improved (see reply to point 8). We do not interpret the impact of IRF1 beyond the data, in fact we state a relatively minor effect of IRF1 in the control of promoter accessibility (p. 10, lines 20-22) and we have added a reference in agreement with an impact of IRF1 on basal expression of antiviral genes (ref. 39, as suggested by the referee).

We have added discussion on potential limitations of the TurboID approach (p. 11, lines 22-24 and p. 15, lines 3-11).

Revision: Change of the discussion section (p. 11, lines 22-24 and p. 15, lines 3-11).

Revision plan: Improvement of fig 5a (see ref. #3, point 8).

14. Although the TurboID experiments identify known STAT1 and IRF1 interactors, the proposed new interactors are numerous, and none are validated through independent co-IP experiments. Moreover, the results are very noisy, with little differences between untreated BMDMs (where IRF1 is barely expressed) and IFN-treated conditions.

Reply (see also ref #2, point 2): The big advantage of BioID or TurboID is the score of proximity and very transient interactions. Validating BioID hits with technologies such as coIP is not particularly useful as the two technologies will obviously produce different interactomes. In fact, we show in this manuscript that IRF1 and STAT1 show proximity, but they do not form a stable complex under co-IP conditions. This leaves genetic approaches (LOF or GOF) as alternatives. However, apart from the workload (> 100 genes would have to be knocked out or their products overexpressed), most of our hits are expected to produce very broad effects in such experiments, hard to interpret regarding ISGF3 and IRF1 activities.

In view of this situation, we publish exclusively the high confidence nuclear interactors identified in our screen: biological replicates were performed in triplicate, a stringent internal control (TurboID-NLS) was used, and a stringent statistical cut-off for high-confidence interactors (95% FDR between groups) was applied. We further account for the experimental situation by limiting interpretation of the data to confirmed molecular events. For example, STAT1 dimers and the ISGF3 complex are required for histone acetylation in response to IFN and ISGF3 is known to contribute to the exchange of the H2AZ histone variant (refs 11, 14, 71, 72). Our data show that IRF1 contributes to promoter accessibility changes and this is in line with its proximity to a remodelling complex. Thus, the BioID data indeed validate previous findings. However, in agreement with the referee's comment, some of the data remain descriptive (such as the intriguing proximity of both STAT1 and IRF1 to nuclear products of ISG). To determine the importance of this molecular proximity is a major undertaking and beyond the scope of this study.

Revision: We added discussion to state the difficulty of validating TurboID-based interactions and the limitations of the TurboID experiments (p.15 lines 3-11).

Revision plan: n. a.

15. The work should be discussed in the context of the demonstrated physiopathological evidence of the IRF1 and IRF9 functions. IRF9 (Hernandez et al., JEM 2018) and more recently IRF1 (Rosain et al Cell, 2023) were identified as causing non overlapping phenotypes in human patients carrying loss-of-function mutations for these genes. The authors must interpret their results in this context.

Reply: We thank the referee for reminding us about the importance of these papers for our work.

Revision: The papers have been mentioned and discussed (p. 13 lines 19-28 and p.14, lines 9-14).

Revision plan: n. a.

Minor comments:

- In most graphs the expression values or log2FC are shown separately for IFN β and IFN γ , however in the heatmaps (Fig 1d, S1d) the IFN β and IFN γ results are

intercalated keeping them side-by-side for each time point, which makes them more difficult to interpret.

Reply: We are in a quandary about the design of the figure. On the one hand our goal is to visualize gene clusters with distinct behaviors for each IFN type. For this purpose, it would be advantageous to separate the IFN types. On the other hand, we aim at showing similarities and differences between genes induced by each IFN type, for this purpose it is better to maintain the current sample order. While understanding the referee's point, we prefer to keep the figure as it is, because the suggested change will not increase its overall clarity.

Revision: n. a.

Revision plan: n. a.

- Fig 1e. The color scales on the GO enrichment graphs are misleading since they use the same blue-to-red gradient for adj p-values ranging from 10⁻²⁵ to 10⁻⁴⁹ and 0.008 to 0.016, which could be considered non significant.

Reply: We agree that this is confusing. It results from automated assignments of the color gradients by the software.

Revision: n. a.

Revision plan: We will investigate possibilities to change color codes for different ranges of p values.

- The inconsistency in the title referring to IFN β as Type 1 but using IFN γ instead of Type 2 nomenclature, perhaps consistency is best.

Reply: We agree about the importance of consistency but find ourselves in yet another quandary. While the use of 'type I IFN' is clearly indicated and widely used as a collective name for this group of cytokines, the use of 'type II IFN' for IFN γ is rare because it is the only member of this type. Hence, we decided for sticking with convention at the expense of a bit of consistency. We agree about the title, though, and have changed type I IFN to IFN β .

Revision: We adapted the title in agreement with the referee's comment.

Revision plan: n. a.

- The incomplete schema in Figure 1a, which only focuses on PRO-seq and does not include the ATAC-seq element.

Reply: We will add a new figure to visualize the set-up of the ATAC seq experiments and their intersection with the Pro-seq data.

Revision: n. a.

Revision plan: We will add a new figure in accordance with the referee's request.

- Figure 6d includes a color scale of -1 to +3, but it is unclear what these values represent and how they were calculated per interactor. The figure legend should be revised to clarify this information.

Reply: We agree. The relevant information has been added to the figure legend.

Revision: We added information (log2FC with regard to the NLS control) to the legend of fig. 6d.

Revision plan: n. a.

- The clearer labeling of Figure 5a and 5b.

Reply: Please refer to our reply to point 8.

- The statement that "IFN-I are the more important mediators of antiviral immunity" is not entirely accurate and may be an oversimplification, as there are certainly articles which suggest a larger role for type II IFN elements than type I (ref: Yamane D et al., 2019 Nature microbiology). While yes, IFN-I plays a critical role in the innate immune response to viral infections, IFN γ also has antiviral activity and is involved in the adaptive immune response to viral infections, and in some instances to a larger extent than IFN I.

Reply: The Yamane et al study (now mentioned on p 10, lines 22-25 and referenced) agrees with our findings because it shows that IRF1 contributes to the basal expression of an ISRE-driven ISG subset. Our statement about the predominant role of type I IFN versus IFN γ refers to genetic data in both humans (mainly Casanova's work including effects of autoantibodies against type I IFN, see also the paper about human STAT2 deficiency in the June 15th issue of the JCI, <https://doi.org/10.1172/JCI168321>) and mice (hundreds of papers) showing that disruption of type I IFN synthesis or response causes profound effects of antiviral immunity (i.e. resulting susceptibilities are first and foremost to viral pathogens) whereas susceptibilities as a consequence of disrupting the IFN γ pathway are first and foremost to intracellular nonviral pathogens such a mycobacteria. In fact, the term mendelian susceptibility to mycobacterial disease (MSMD) was coined by Casanova and colleagues to describe a variety of human mutations that include those of the IFN γ , but not the type I IFN pathway.

Maybe more importantly, the Rosain et al. paper mentioned by the referee which appeared in 'Cell' while our study was under review, shows that human IRF1 mutations also fall into the MSMD category (new ref. 65). In contrast, the authors did not observe diminished antiviral immunity. This emphasizes the main conclusions of our study about the relevance of IRF1 for macrophage activation. We discuss this paper on p 14. lines 9-14.

Obviously, this does not exclude a role of type I IFN in nonviral infection or of IFN γ in viral infection, in fact much of our own work has been dedicated to a role of type I IFN in

infections with *L. monocytogenes*. Nevertheless, we think that in a generic statement about the difference between type I IFN and IFN γ it is correct to label the former as predominantly antiviral and the latter predominantly as a macrophage activating factor against nonviral, intracellular pathogens.

Revision: We added discussion of Rosain et al. (ref. 65) on p 14. lines 9-14.

Revision plan: n. a.

- The authors claim that a significant portion of ISG promoters is associated with ISGF3 upon IFN γ receptor engagement and that the transcriptomes of macrophages treated briefly with IFN β or IFN γ exhibit remarkable similarity and sensitivity to Irf9 deletion. However, I am uncertain about the extent of consensus on this claim.

Reply: The data were surprising but supported by ChIP-seq and RNA-seq in wt and IRF9 ko macrophages (ref 10). Data in a follow-up study (ref. 11) and in this manuscript support our original conclusion by demonstrating the impact of the IRF9 ko on IFN γ responses. Importantly, we don't claim this is true in all cell types, it may well depend on STAT/IRF9 expression levels and tonic IFN signaling.

Revision: n. a.

Revision plan: n. a.

- Fig 1e, S1c. Graphs having circles of varying sizes in function of a value are named "bubble plots" and not "dot plots".

Reply: Thank you for pointing this out, we corrected our mistake.

Revision: We changed dot plot to bubble plot in legend to figure S1c.

Revision plan: n. a.

- Fig S1b, S3b. The PRO-seq was generated in triplicates, hence these graphs should include the Log2FC for the individual data points.

Reply: The Log2FC from DESeq2 were calculated from the triplicates, the software does not compute Log2FC from individual replicates.

Revision: n. a.

We mention the p-values for the Log2FC to show the degree of consistency (figure legends).

Revision plan: We will provide a table with log2FC and corresponding padj values of the genes represented at each timepoint (table_showing_padj_values_and_log2fc).

- Fig S3c legend. It is mentioned "Graph represents RT-qPCR of genomic Mx2". RT-qPCR usually stands for reverse transcription quantitative PCR, hence we suggest to change to "ChIP-qPCR" or qPCR. Confusingly, in the literature the term "RT-PCR" is

used for real-time PCR and "qPCR" for quantitative PCR. Also, the authors should be specific about the "genomic" region targeted; the graphs mention "promoter", hence it would be appropriate to use the same designation in the legend.

Reply: We agree and thank the referee for correction of the terminology.

Revision: We changed RT-PCR to qPCR throughout the manuscript. Moreover, we specifically refer to 'promoter region' as the amplified DNA.

Revision plan: n.a.

- Fig S3e. The y-axis names are missing.

Reply: Thanks for spotting this.

Revision: The y axis in the figure received its proper label.

Revision plan: n. a.

- In the genomic snapshot shown, only bars or fading triangles are shown in place of the gene body. The authors should provide an accurate gene structure; i.e., exons and introns.

Reply: We will try to include the exon-intron structure wherever the size of the figure allows this.

Revision: n. a.

Revision plan: If figure size permits, we will add the exon-intron structure of the genes in browser tracks as requested.

- Raw cells are sometimes spelled as "Raw" and other times as "RAW". Please choose one for consistency.

Reply: We agree.

Revision: This inconsistency has been corrected

Revision plan: n. a.

- In p.10 l.20, the figure number is missing.

Reply: We agree.

Revision: We corrected this mistake.

Revision plan: n. a.

B Response to referees according to the submitted revision plan
Point-by-point-reply

1. Ref. 3, comment 1:

In response to the referee's request, we labelled each of the clusters in Fig 1D with the numbers that reference the clusters in the manuscript text.

2. Ref. 3, comment 3:

In accordance with the referee's request, we performed three additional biological replicates of the ChIP experiments (Fig EV2C, D). The data derived from IFN α -treated cells confirm the interpretation that there are no significant changes in the ratios of STAT1 and STAT2 binding at early and late time points of the transcriptional response. Together with Fig EV2F measuring the phosphorylated STAT1 and STAT2, the interpretation is supported that the canonical ISGF3 complex mediates both the early and late responses to IFN α .

The difference ISGF3 subunit binding between the untreated cells and 24h after IFN α treatment did not reach statistical significance. This is consistent with a weaker and more transient response of ISGF3 target genes in the IFN α compared to the IFN β response.

3. Ref 3, comment 7:

In accordance with the referee's request we show data to confirm the correlation between the overall trends of ISG transcription (Pro-seq) and ISG chromatin accessibility (ATAC-seq) between the biological replicates. New Appendix Fig S1 shows the Pearson correlations of z-scores calculated across treatment times (untreated, early, late) of vst-normalized reads derived from ATAC-Seq vs PRO-seq for (a) IFN β and (b) IFN γ stimulation.

4. Ref 3, comments 8 and 13, minor comment 4:

We modified Fig 5 to comply with the referees' request to better demonstrate the transcription factor requirement for chromatin accessibility changes.

The labeling of the figure was improved to show that

- the volcano plot represents PRO-seq data
- the color code of individual dots representing significantly induced ISG indicates whether the genes are induced independently of IRF1 (Fig 5B, blue dots), whether their transcriptional induction requires IRF1 (Fig 5B, purple

dots) or whether both transcription and accessibility changes require IRF1 (Fig 5B, yellow dots).

- In Fig 5C the same color code indicates IRF9 requirement, or the lack thereof. Fig. 5d shows browser tracks of the genes mentioned in the referees' comment (Ifi44 and Gbp2). The tracks clearly demonstrate that the increase in accessibility of the Ifi44 gene upon IFN treatment is completely lost in absence of IRF1, whereas the increase of the Gbp2 gene remains the same. For direct visualization of the corresponding transcriptional response the trend lines of Ifi44 and Gbp2 transcription according to PRO-seq were added to figure EV3.

The size of the panels has been increased for better resolution of individual dots in the volcano plots. In addition, panel A shows the work flow of the experiment as requested by the referee in a minor comment.

5. Ref 3, comment 14:

The referee asks to validate our Turbo ID data with IP experiments. We will restate our skepticism about the usefulness of such an undertaking below. Nonetheless, we precipitated Turbo-ID samples using Streptavidin, followed by western blot and compared the data to IP-western blots. The results of our data are appended for the referee.

Of the tested proteins, only STAT2 was precipitated via STAT1 in both situations, suggesting proximity and physical interaction. The other tested protein, Usp22, shows clear proximity to STAT1 in the Streptavidin pull-down, but does not co-immunoprecipitate. This situation is also reported for the interaction between STAT1 and IRF1 which is found in the TurboID-MS approach, but not by coIP with either IRF1 or STAT1 antibodies (fig. EV5).

It is important to point out that these examples are a minority of those we have tested. The frustrating experience has been that with most antibodies (if at all available), small changes in the interaction with STAT1 or IRF1 are lost in the background of the western blot. This is to be expected as only a minor fraction of any interactor will be proximal to STAT1 or IRF1. For example, components of the histone acetylase complexes we identified mediate histone modification changes on very many genes, and they cannot be expected to interact with our baits to any major extent. It takes a sensitive technology such as MS to identify such interactions. Moreover, we have intentionally analyzed nuclear extracts to have a higher chance of revealing DNA-dependent interactions between transcription factors or chromatin modifiers. DNA-dependent interactions are impossible to reproduce in a co-IP experiment.

The problem of using western blots for the validation of MS data are well known (DOI: 10.1002/pmic.201800222) and, in line with our thoughts, are refuted by some as validating a better technology with one that is worse (<https://doi.org/10.1038/s41477-022-01314-8>). A better strategy would be to use an orthogonal technique for validation. The only proximity labelling technology that has

the potential to address the same questions as TurboID is APEX (doi: 10.1002/wdev.272). However, APEX is known to produce high backgrounds and is usually coupled with SILAC for interpretable results (see introduction in doi: 10.3390/cells9051070).

We would once more like to point out the diligent control of, and high level of confidence in our data:

- Independent biological triplicates.
- Stringent statistical cut-off.
- Sufficiently high number of peptides identified for most interactors.
- Known interactors have been identified
- Several components of chromatin modifying complexes were identified for both STAT1 and IRF1, making it highly unlikely that all interactions are artefacts.

To underfeed these statements, we include the QC from the mass spectrometry data.

In essence, we are aware of the fact that TurboID may identify serendipitous proximity. However, based on the above statements, we are confident that our experimental procedure rules out such randomness as much as possible and that the majority of the reported interactions are real.

Ref 3, minor comment 13

Exon-Intron structures are provided as requested by the referee (Figs 2B, 3D, 4A, C, 5D).

PRIDE accession number.

Project Name: Identifying interactors of IRF1 and STAT1 during type I and type II interferon treatment using proximity-dependent labeling

Project accession: PXD040337

Project DOI: Not applicable

Reviewer account details:

Username: reviewer_pxd040337@ebi.ac.uk

Password: w1V3828B

Reviewer links for the associated genomics data sets SRA BioProject

PROseq

PRJNA931836

ATACseq

PRJNA937204

Chromatin accessibility during prolonged interferon treatment

<https://dataview.ncbi.nlm.nih.gov/object/PRJNA937204?reviewer=gphu8g3oopede8m8esg4nm8q60>

ChIP-seq:

PRJNA947574

IRF9 binding in BMDMs during early versus later stages of interferon stimulation

<https://dataview.ncbi.nlm.nih.gov/object/PRJNA947574?reviewer=86kmgoo6ackdfe6fum0t3372i2>

Dear Dr. Decker,

Thank you for your message. I apologize for the delay in the assessment. I have unfortunately, despite numerous reminders only received feedback from referee #3. This referee thinks that in principle most of the concerns have been addressed and that also the proposed additional experiments appear appropriate. I would agree with this referee that the proposed revisions (including the attempted co-IP conformation of the bio-ID experiments) are appropriate.

I have now again taken some time to go through the manuscript and your point-by-point reply. I must admit that I find the chosen format rather confusing, and it was not always clear which of the proposed experiments or text modifications are already in the manuscript and which of the changes are still pending. We usually do not consider partial revisions at the EMBO Journal as it is impossible for both the referees and the editors to properly evaluate the manuscript under these circumstances. I obviously should have realized this right away and therefore will not hold this against you here.

Anyhow, I do not think it makes sense to wait any longer for referees #1 and #2 at this point and I would therefore ask you to revise your manuscript according to the remaining revisions detailed in the revision plan.

Thank you for the opportunity to consider your work for publication. I look forward to your revision.

Yours sincerely,

Cornelius Schneider, PhD
Editor
The EMBO Journal
c.schneider@embojournal.org

We realize that it is difficult to revise to a specific deadline. In the interest of protecting the conceptual advance provided by the work, we recommend a revision within 3 months (28th Apr 2024). Please discuss the revision progress ahead of this time with the editor if you require more time to complete the revisions. Use the link below to submit your revision:

Referee #3:

The authors have addressed most of the comments and concerns raised by this reviewer. For the outstanding points, an appropriate revision plan has been provided.

A Response to referees of RC submission (black font) and EMBO J submission (red font).

Reviewer #1

1. Figure 2. Difficult to interpret data as it is presented. Consider quantifying figure 2C in order to make "changes in *Pol II pausing* were more pronounced during IFN β signaling" statement more apparent.

Revision RC: We removed the original figure 2c and replaced it with original figure S2. This representation is quite intuitive as the graphs represent a direct quantitative logarithmic display whether and how much the relative amount of paused polymerase changes when comparing IFN-treated and untreated cells. The calculation of these ratios is now explained better in the legend to figure 2.

2. How are you distinguishing autocrine signaling in the BMDMs driven by IFN treatment from late transcripts (for example, at 48 hours are differential genes due to autocrine cytokine signaling or are they truly late transcripts)?

Revision RC: We now mention that enhanced IFN production is not sustaining ISG responses (p.5 lines 18/20). We mention the possibility that secreted factors may drive secondary or tertiary waves of ISG transcription (p. 8, lines 21/23).

3. Figure 3D. Authors choose *Gbp2* (as positive control for IFN γ driven gene), but don't show that *Gbp2* is a IFN β independent gene. Consider using IRF1 KO BMDMs in this data as well.

Revision RC: We better explain the rationale behind the experiments shown in figure 3D (text on p8, lines 12-16). In addition, we show the trend line of *Gbp2* expression in WT vs IRF1KO as well as that of additional genes showing delayed/sustained responses in the new Figure S3.

Minor issues:

1. Simplify figure 4B- consider focusing on most differentially expressed genes between clusters

Reply RC: Simplifying the figure would mean to separate it according to time point, IFN type treatment or knock-out effect. We think this would require to mentally reassemble the figure to understand the interrelationships between these parameters. To our opinion the visual display of the data interrelationship in fig. 4B facilitates the impropriation of the information content.

2. Define known IFN γ and IFN β driven genes when they are introduced in figure 2 rather than in discussion.

Revision RC: Following the referee's suggestion we provide the examples of IFN β and IFN γ -controlled genes and the characteristics of their regulation in the context of our description of the results displayed by fig. 2 (p.6 lines 15-21). This includes *Gbp2* (see major issue no. 3).

3. Clarify which cell types (IRF1 KO vs IRF9 KO) are used in figure 5 A/B.

Reply RC: The cell type (bone marrow-derived macrophages) is mentioned in the first sentence of the figure legend. Since all experiments except the Bio-ID experiment were performed with this cell type we decided not to label each figure.

4. Unclear whether IRF1 expression in figure 3A is from whole cell lysate or nuclear fraction.

Revision RC: We added a sentence with the relevant information in the legend of figure 3.

5. Authors suggest IFN β treatment induces less IRF1 at later time points, however loading control also seems slightly lower than other considerations. Is it possible that IFN β treated cells are dying at later time points, given that type I IFN signaling can be pro-apoptotic.

Reply and Revision RC: The graph below the blot represents quantified IRF1 signals, normalized to the loading control. It shows that the differences are not generated by unequal loading of the blotted gel. We and others have shown that IFN β may indeed enhance macrophage death, however only when the cells are simultaneously infected with an intracellular pathogen (e.g. new ref. 25). These studies also show that treatment with IFN β alone over periods used in the present study does not affect macrophage viability. We added a sentence about the viability of IFN-treated macrophages (p. 4, lines 31-32).

Reviewer #2

1. General assessments: The current study adds some to these observations...some of these observations are incremental...biological significance is limited. While this reviewer does not suggest additional experimentation, this manuscript would be suitable as a resource paper.

Reply and Revision RC: We tried to better explain the scientific motivation for this study and the significance of the results (p. 4, lines, lines 12-25).

It appears we were not clear enough in explaining the novel aspects of our study. The starting points are two published studies from our lab demonstrating a global increase of ISGF3 association with ISG promoters in IFN γ -treated cells and a remarkable similarity of IFN- γ and type I IFN-induced early transcriptome changes. These findings challenge the notion in the field (as mentioned by the referee) that IFN γ specificity is produced by the predominant deployment of STAT1 homodimers. We thus tested the hypothesis that the specificity of the IFN γ -induced transcriptome is generated over time, rather than during the early response, and relies on secondary responses to transcription factors such as IRF1. In contrast, IRF1 plays no or only a small role in the type I IFN response that utilises ISGF3 and/or unknown secondary factors in the delayed response. We tested this hypothesis with PRO-seq technology to rule out confounding effects of mRNA processing over a 48h period. The data are clear in showing that many genes associated with the antibacterial or anti parasite profile of activated macrophages are indeed much more abundant in late-stage rather than briefly IFN γ -treated macrophages and these delayed changes are to a large extent

dependent on IRF1. Our findings are based on the best available technologies, a combination of nascent transcript analysis with genetics and protein interaction studies. In addition, our findings rule out alternative models of sustained or secondary ISG transcription, such as the employment of alternative ISGF3 complexes (such as STAT2-IRF9) or of ISGF3 complexes formed with unphosphorylated STAT1 and STAT2. We provide evidence for higher order waves of transcription caused by unknown transcription factors that are produced by transcriptional activation of ISGF3 or IRF1 target genes and identify candidates among the AP1 and Ets transcription factor families. We agree that some of the data are confirmatory rather than novel (i.e. some of the genes we describe were known from previous literature to be IRF1 targets), but it is the systems approach of our study, and particularly the delineation of conditions under which the largely neglected delayed response diverts the IFN β and IFN γ -induced transcriptomes, that generates a comprehensive and conclusive view of IFN γ acting predominantly as a macrophage activating factor, and IFN β being an essential antiviral cytokine. We do think this main outcome is immunologically meaningful and not incremental. For this reason, we would prefer to publish the paper as a relevant contribution to innate immunology rather than a resource. Emphasizing our point, a paper appeared in 'Cell' while our study was under review, showing that human IRF1 mutations cause mendelian susceptibility to mycobacterial disease (MSMD), a term coined by JL Casanova and colleagues for immunological defects that reduce the ability of macrophages to cope with intracellular bacteria (new ref. 65). This important study emphasizes the main conclusions of our study about the relevance of IRF1 for macrophage activation. We discuss this paper on p. 14 lines 9-14.

2. **Biological significance is limited as this study is largely descriptive and they do not test the hits obtained from BioID.**

Reply and Revision RC: We added discussion to about the difficulty of validating TurboID-based interactions and the limitations of the TurboID experiments (p.15 lines 3-11).

The big advantage of BioID or TurboID is the ability to score proximity and very transient interactions. Validating BioID hits with technologies such as coIP is not particularly useful as the two technologies will obviously produce different interactomes. In fact, we show in this manuscript that IRF1 and STAT1 show proximity, but they do not form a stable complex under co-IP conditions. This leaves genetic approaches (LOF or GOF) as alternatives. However, apart from the workload (> 100 genes would have to be knocked out or their products overexpressed), most of our hits are expected to produce very broad effects in such experiments, hard to interpret regarding ISGF3 and IRF1 activities.

In view of this situation, we publish exclusively the high confidence nuclear interactors identified in our screen: biological replicates were performed in triplicate, a stringent internal control (TurboID-NLS) was used, and a stringent statistical cut-off for high-confidence interactors (95% FDR between groups) was applied. We further account for the experimental situation by limiting interpretation of the data to confirmed molecular events. For example, STAT1 dimers and the ISGF3 complex are required for histone acetylation in response to IFN, and ISGF3 is known to contribute to the exchange of the H2AZ histone variant (refs 11, 14, 71, 72). Our data show that IRF1 contributes to promoter accessibility changes and this is in line with its proximity to a remodelling complex. Thus, the BioID data indeed validate previous findings. However, in agreement with the referee's comment, some of the data remain descriptive (such as the intriguing proximity of both STAT1 and IRF1 to nuclear products of

ISG). To determine the importance of this molecular proximity is a major undertaking and beyond the scope of this study.

3. The sequencing and BioID data are not submitted to public databases.

Revision RC: The accession number was added on p.29, line 25.

Reviewer #3

Major comments:

1. In Fig. 1d is difficult to interpret and misleading for many reasons. First, the cluster numbering is disconnected from the cluster order; why not numbering them based on the hierarchical clustering and writing the cluster number besides the cluster itself? Second, having a 2-color gradient is misleading; negative values shouldn't be in the same color tone than the positive values. Third, the authors did not provide adequate rationale behind using only the top 1,000 most expressed gene? Why not using all the differentially expressed genes in at least one of the condition to provide a comprehensive analysis? Could this potentially lead to bias in the data, and is there any information lost by not using the - lower - expressed genes fraction? Fourth, it is not clear what the color scale is representing and how the data was transformed. Was a mean centering of the expression values of the log2FC applied to the RNA-seq data to facilitate clustering? Mean centering and z-scoring is a common technique used to adjust expression data, but it can potentially exaggerate differences between samples. More information about the data and analysis should be provided, as it is difficult to determine whether this was a valid approach or not.

Reply and Revision RC:

- To create the heatmap, we used the pheatmap package from R and the cutree_rows option to separate 11 clusters with strikingly different patterns of gene expression based on visual exploration. The numbering was autogenerated by the program.
- The data is now shown in red-blue.
- We restricted our list to only 1000 genes from each comparison as we aimed to analyze the prominent patterns of gene expression across timepoints. Considering all differentially expressed genes based on a padj value would also include genes expressed at very low levels as evident from the low baseMean values obtained from DESeq2. Hence, we applied a selection of 1000 genes which effectively represented the major patterns of gene expression across timepoints.
- Variance stabilized transformation was applied on read counts obtained from PRO-seq using the DESeq2 package. The transformed reads were z-score normalized and used for performing hierarchical clustering by the "Ward.D2" method using the pheatmap package in R. A total of 3126 genes were used for this analysis. 11 distinct clusters were defined using cutree_rows option. The color scale represents z-score normalized counts. The genes represented in the heatmap were selected based on the following criteria: each timepoint of interferon treatment was compared to the homeostatic condition (untreated sample) in wildtype BMDMs. The differentially expressed genes from each comparison were selected based on the filtering criteria: absolute log2FoldChange ≥ 1 and adjusted p value < 0.01 by Wald test. Following the differential analysis, the first 1000 differentially expressed genes in each treatment condition (ordered based on adjusted p values) were selected for both IFN

types and combined and selected for creating a list which consisted of 3126 unique genes. The scale in the heatmap represents z-scores of variance-stabilized reads, calculated across all genotype and treatment conditions, separately for each IFN type.

- The rationale for using the top 1.000 genes is explained (p.5, lines 7-9). The description of the pro-seq read count processing has been extended in accordance with our reply to the referee in the legend of figure 1d and in the methods section (p. line)

Revision EMBO J: In response to the referee's request, we labelled each of the clusters in Fig 1D with the numbers that reference the clusters in the manuscript text.

2. Fig 2c. The authors claim that RNA Pol II pausing is a major factor in controlling the dynamics of ISG transcription. However, they did not provide sufficient explanation of the results, and in all fairness there is not much variation between the clusters to sustain the claim that this is a major factor in ISG transcriptional control.

Revision RC: We agree with the referee that we cannot posit RNA pol II pausing as a major factor for the differences of transcriptional control of ISG in individual clusters. We have made sure to remove any statements suggesting this possibility. We also try to better integrate our findings with RNA pol II pausing into the existing literature. We added relevant literature on p. 6 lines 28-30 and p. 7, lines 4-6.

3. The large standard deviation bars in the claim that ChIP data confirmed the binding of ISGF3 components to the promoter of Mx2 cast doubt on the validity of the results and conclusions. The authors should consider additional experiments or complementary analyses to validate their findings. Or alternative, to adjust their claims accordingly.

Reply RC: To demonstrate sufficient quality of the data the ratio of Stat1/ Stat2 was calculated for early (1.5hrs) and late (48h) separately. The unpaired two-tailed t test comparing this ratio between 1.5 hrs and 48hs, shows that they are not significantly different. This indicates that all ISGF3 components are associated with ISG during both early and delayed responses, i. e., that STAT2/IRF9 complexes are unlikely to contribute to delayed ISG control. However, we agree with the referee that the standard deviations of the kinetic ChIP experiment are high and that it would be good to generate additional data.

Revision EMBO J In accordance with the referee's request, we performed three additional biological replicates of the ChIP experiments (Fig EV2C, D). The data derived from IFN β -treated cells confirm the interpretation that there are no significant changes in the ratios of STAT1 and STAT2 binding at early and late time points of the transcriptional response. Together with Fig EV2F measuring the phosphorylated STAT1 and STAT2, the interpretation is supported that the canonical ISGF3 complex mediates both the early and late responses to IFN β .

The difference of ISGF3 subunit binding between the untreated cells and 24h after IFN γ treatment did not reach statistical significance. This is consistent with a weaker and more transient response of ISGF3 target genes in the IFN γ compared to the IFN β response.

4. On p.5, the authors mention "Representative browser tracks from the Gbp2 and Slfn1 genes further validate this observation" but they are simply referring to genome browser snapshot, i.e., specific genomic examples, extracting from the same single dataset. Without using an independent dataset, this can not "further validate" the initial findings.

Revision RC: We changed the paragraph describing this experiment (p. 6, lines 15-21).

5. IRF1 was successfully pulled down with STAT1 bait but not in the reciprocal experiment. The author should discuss this point as it is important for the conclusions. Could it potentially indicate issues with the technique used, and if this could introduce any bias into the results. The statement, "In contrast, interactors of the IRF1 bait did not include STAT1. This discrepancy could result from steric constraints of the tagged proteins due to the limitation of the 10nm distance reached by the biotin ligase," does not seem to be sufficient to explain this discrepancy.

Reply and Revision RC: IRF1 was present in the STAT1 pull-down, clearly above background, but the interaction did not conform to our criterium of a 95% confidence interval for the FDR. To be consistent we did not include it in the list of IRF1 interactors. We have observed on several occasions that the significance of proximity is not reciprocal, even for well- documented physical interactions. A prime example for this is the interaction between STAT1 and IRF9 in IFN-treated cells which is recorded in the STAT1 pull-down, but not that with IRF9 (ref. 10). Apart from steric reasons the lack of reciprocity may result from different signal/noise ratios in pull downs with different baits.

We now mention that IRF1 was a STAT1 interactor below the statistical cut-off (p. 11, lines 26-28) as well as the possibility of different signal/noise ratios in the IRF1 and STAT1 pull-downs on p.11, lines 22-24.

6. The authors interpret their ATAC-seq and ChIP-seq results based on a 2kb window to the TSS of genes, not considering relatively close enhancers or longer range cis-regulatory interactions in their interpretation. For example, they mention on p.7 "Contrasting the strong binding of IRF9 and IRF1 to the Mx2 (cluster 2) and Gbp2 (cluster 9) promoters, respectively, we saw no evidence for direct binding to Lrp11 (cluster 3) and Ptgs2 (cluster 10)", but on Fig 3d they show only the proximal regions. No scale bars are shown either. Moreover, exploring the same published IRF1 ChIP-seq dataset, there is a clear IRF1 binding site at the promoter of Ptgs2, while the authors report none.

Reply and Revision RC:

- According to the literature (e. g. refs. 11, 27), most IFN-induced accessibility changes occur in the vicinity of the TSS of ISG. This is further strengthened by the data shown in this manuscript. In addition, most functionally validated GAS and ISRE sequences are in the DNA interval chosen for our analysis. While distal ISG enhancers have been reported (e. g. DOI: 10.26508/lsa.202201823), an analysis beyond the placement of most control regions increases the risk of wrong assignments between ISG and their regulatory elements, hence the causality between transcription factor binding and accessibility changes.

- We extended the regions for the analysis of the Lrp11 and Ptgs2 regulatory regions and found no evidence for the binding of ISGF3 or IRF1. We find no evidence for a clear peak in the Ptgs2 promoter. There is a peak called by the Macs2 algorithm, but visual inspection of the track (bigwig file) shows it consists of a minor increase in reads above background that does not suggest a bona fide IRF1 binding site (see below).

IRF1 binding indicated by bigWig browser tracks and corresponding peakfiles detected at the locus. We identified the peakfile from *Langlais et al.*, 2016 and identified peaks using MACS2, however using mm10 genome as the analysis in the original paper was done with mm9 genome. The peak identified here appears to be an artefact of the MACS2 program as there is no evident enrichment at the gene promoter region upon inspection of the bigWig files.

Revision EMBO J: Scales have been added to the browser tracks.

7. Lack of statistical analysis on chromatin accessibility claims: The authors claim that ATAC-seq data in BMDMs stimulated with IFN β or IFN γ for a short (1.5 hours) or long (48 hours) period reveals a striking similarity between transcription and the general trends of chromatin accessibility at regions up to 1000 bp upstream of the TSS (Fig. 2a), suggesting continuous chromatin remodeling during the transcriptional response. However, I would like to know if this conclusion is well-supported by the correlation between the chromatin accessibility from ATAC-seq data from only one sample and the PRO-seq data.

Reply RC: We will analyze single experiments whether they support the conclusions derived from the z-score of the triplicate samples.

Revision EMBO J: In accordance with the referee's request we show data to confirm the correlation between the overall trends of ISG transcription (Pro-seq) and ISG chromatin accessibility (ATAC-seq) between the biological replicates. New Appendix Fig S1 shows the Pearson correlations of z-scores calculated across treatment times (untreated, early, late) of vst-normalized reads derived from ATAC-Seq vs PRO-seq for (a) IFN β and (b) IFN γ stimulation.

8. The need for additional experiments to verify claims such as the dependence of Irf44 on IRF1 for gaining ATAC signal, as stated in the claim, "Expression required IRF1 for both, but accessibility of the Irf44 regulatory region depended upon IRF1 whereas that of Gbp2 acquired an open structure independently of IRF1 (Fig. 5c). **Reply RC:** We think the lack of clarity might be related to the size of figures 5a and 5b and the density of the dots in some areas of the plot. We agree it is very difficult to assign our gene labels unambiguously to a single dot.

Fig. 5a combines ATACseq data in wt and IRF1 knockout cells with the expression data from the Pro-seq experiment, Fig. 5b is the same set-up, but IRF9-deficient macrophages are analyzed.

Blue dots show ATACseq signals induced by IFN treatment. Violet dots represent genes that require IRF1 (Fig. 5a) or IRF9 (Fig. 5b) for transcriptional induction. Yellow dots mark genes such as IFI44 requiring IRF1 (Fig. 5a) or IRF9 (Fig. 5b) for both expression and the accessibility change in the promoter region. Fig. 5c visualizes representative examples of

genes whose accessibility is coupled to the transcription factor dependence of the transcriptional induction (IFI44), or not (Gbp2). Thus Fig. 5c must be interpreted based on the dot color code in fig. 5a and we admit this has been difficult with the figure in its present form.

Revision EMBO J: We modified Fig 5 to comply with the referees' request to better demonstrate the transcription factor requirement for chromatin accessibility changes.

The labeling of the figure was improved to show that

- the volcano plot represents PRO-seq data
- the color code of individual dots representing significantly induced ISG indicates whether the genes are induced independently of IRF1 (Fig 5B, blue dots), whether their transcriptional induction requires IRF1 (Fig 5B, purple dots) or whether both transcription and accessibility changes require IRF1 (Fig 5B, yellow dots).
- In Fig 5C the same color code indicates IRF9 requirement, or the lack thereof.

Fig. 5d shows browser tracks of the genes mentioned in the referees' comment (Ifi44 and Gbp2). The tracks clearly demonstrate that the increase in accessibility of the Ifi44 gene upon IFN treatment is completely lost in absence of IRF1, whereas the increase of the Gbp2 gene remains the same. For direct visualization of the corresponding transcriptional response the trend lines of Ifi44 and Gbp2 transcription according to PRO-seq were added to figure EV3. The size of the panels has been increased for better resolution of individual dots in the volcano plots. In addition, panel A shows the work flow of the experiment as requested by the referee in a minor comment.

9. In the figure legends, there is missing information about the number of times experiments were replicated, suggesting that some were done a single time. Moreover, some graphs are missing statistical analysis, e.g., in Fig S3cS3e, S3f, the ChIP-qPCR experiments were done on biological triplicates, there is no mention of statistical test performed, it is not mentioned what the error bars represents (SD, SEM, etc.) and the variance is large, but the authors still interpret these results as significant enrichment of the transcription factors to the Mx2 promoter.

Reply and Revision RC: Where missing the relevant information has been added to figure legends. In brief, all experiments represent at least three biological replicates. The only exception is the western blot shown in figure S3a, (no S2a) which represents two independent replicates. Here, the clarity of the difference of IRF1 expression and the fact that the only purpose is to show that Raw264.7 macrophages behave like bone marrow-derived macrophages in fig. 3a justifies the omission of another replicate (please see also answer to point 3). The relevant information has been added to figure legends where necessary (figs. 1, a, 3a, 6a-f, S1, S4, S5).

10. Another example are the RNA Pol II pausing index ratios, which show minor variations and not are supported by statistics to support a possible significance. Proper description, replication and statistical analyses of the results are critical.

Reply and Revision RC: Statistics underlying the RNA Pol II pausing data are included in supplementary data 2.

11. The authors used CRISPR-Cas9 genome editing to generate knockout cell lines. However, they did not verify the knockouts at the protein level. Further experiments could confirm that the targeted proteins are not expressed in the knockout cell lines.

Revision RC: We included a western blot showing the lack of IRF1 and STAT1 expression in the respective cell lines (new figure S6).

12. On p.9, it is mentioned "IRF1 affects chromatin structure ...". Here chromatin structure is related to minor changes in chromatin accessibility, this can not be qualified as changes in chromatin structure.

Revision RC: ,structure' has been replaced with 'accessibility'. (p. 10, lines 19 and 21).

13. The authors have not adequately addressed the methodological limitations in their discussion, which extends beyond the aforementioned comments. It is suggested they include a comprehensive discussion of the claims made pertaining to the necessity of IRF1 for accessibility and the potential biases in the interactomes, along with their associated consequences.

Reply and Revision RC: The contribution of IRF1 to the accessibility of ISG promoters emerges from the data in figures 5a, whose clarity will be improved (see reply to point 8). We do not interpret the impact of IRF1 beyond the data, in fact we state a relatively minor effect of IRF1 in the control of promoter accessibility (p. 10, lines 20-22) and we have added a reference in agreement with an impact of IRF1 on basal expression of antiviral genes (ref. 39, as suggested by the referee).

We have added discussion on potential limitations of the TurboID approach (p. 11, lines 22-24 and p. 15, lines 3-11).

14. Although the TurboID experiments identify known STAT1 and IRF1 interactors, the proposed new interactors are numerous, and none are validated through independent co-IP experiments. Moreover, the results are very noisy, with little differences between untreated BMDMs (where IRF1 is barely expressed) and IFN-treated conditions.

Reply (see also ref #2, point 2): The big advantage of BioID or TurboID is the score of proximity and very transient interactions. Validating BioID hits with technologies such as coIP is not particularly useful as the two technologies will obviously produce different interactomes. In fact, we show in this manuscript that IRF1 and STAT1 show proximity, but they do not form a stable complex under co-IP conditions. This leaves genetic approaches (LOF or GOF) as alternatives. However, apart from the workload (> 100 genes would have to be knocked out or their products overexpressed), most of our hits are expected to produce very broad effects in such experiments, hard to interpret regarding ISGF3 and IRF1 activities.

In view of this situation, we publish exclusively the high confidence nuclear interactors identified in our screen: biological replicates were performed in triplicate, a stringent internal control (TurboID-NLS) was used, and a stringent statistical cut-off for high-confidence interactors (95% FDR between groups) was applied. We further account for the experimental situation by limiting interpretation of the data to confirmed molecular events. For example, STAT1 dimers and the ISGF3 complex are required for histone acetylation in response to IFN

and ISGF3 is known to contribute to the exchange of the H2AZ histone variant (refs 11, 14, 71, 72). Our data show that IRF1 contributes to promoter accessibility changes and this is in line with its proximity to a remodelling complex. Thus, the BioID data indeed validate previous findings. However, in agreement with the referee's comment, some of the data remain descriptive (such as the intriguing proximity of both STAT1 and IRF1 to nuclear products of ISG). To determine the importance of this molecular proximity is a major undertaking and beyond the scope of this study.

Revision RC: We added discussion to state the difficulty of validating TurboID-based interactions and the limitations of the TurboID experiments (p.15 lines 3-11).

Reply and Revision EMBO J: The referee asks to validate our Turbo ID data with IP experiments. We will restate our skepticism about the usefulness of such an undertaking below. Nonetheless, we precipitated Turbo-ID samples using Streptavidin, followed by western blot and compared the data to IP-western blots. The results of our data are appended for the referee.

Of the tested proteins, only STAT2 was precipitated via STAT1 in both situations, suggesting proximity and physical interaction. The other tested protein, Usp22, shows clear proximity to STAT1 in the Streptavidin pull-down, but does not co-immunoprecipitate. This situation is also reported for the interaction between STAT1 and IRF1 which is found in the TurboID-MS approach, but not by coIP with either IRF1 or STAT1 antibodies (fig. EV5).

It is important to point out that these examples are a minority of those we have tested. The frustrating experience has been that with most antibodies (if at all available), small changes in the interaction with STAT1 or IRF1 are lost in the background of the western blot. This is to be expected as only a minor fraction of any interactor will be proximal to STAT1 or IRF1. For example, components of the histone acetylase complexes we identified mediate histone modification changes on very many genes, and they cannot be expected to interact with our baits to any major extent. It takes a sensitive technology such as MS to identify such interactions. Moreover, we have intentionally analyzed nuclear extracts to have a higher chance of revealing DNA-dependent interactions between transcription factors or chromatin modifiers. DNA-dependent interactions are impossible to reproduce in a co-IP experiment.

The problem of using western blots for the validation of MS data are well known (DOI: 10.1002/pmic.201800222) and, in line with our thoughts, are refuted by some as validating a better technology with one that is worse (<https://doi.org/10.1038/s41477-022-01314-8>). A better strategy would be to use an orthogonal technique for validation. The only proximity labelling technology that has the potential to address the same questions as TurboID is APEX (doi: 10.1002/wdev.272). However, APEX is known to produce high backgrounds and is usually coupled with SILAC for interpretable results (see introduction in doi: 10.3390/cells9051070).

We would once more like to point out the diligent control of, and high level of confidence in our data:

- Independent biological triplicates.
- Stringent statistical cut-off.
- Sufficiently high number of peptides identified for most interactors.
- Known interactors have been identified
- Several components of chromatin modifying complexes were identified for both STAT1 and IRF1, making it highly unlikely that all interactions are artefacts.

To underfeed these statements, we include the QC from the mass spectrometry data.

In essence, we are aware of the fact that TurboID may identify serendipitous proximity. However, based on the above statements, we are confident that our experimental procedure rules out such randomness as much as possible and that the majority of the reported interactions are real.

15. The work should be discussed in the context of the demonstrated physiopathological evidence of the IRF1 and IRF9 functions. IRF9 (Hernandez et al., JEM 2018) and more recently IRF1 (Rosain et al Cell, 2023) were identified as causing non overlapping phenotypes in human patients carrying loss-of-function mutations for these genes. The authors must interpret their results in this context.

Revision RC: The papers have been mentioned and discussed (p. 13 lines 19-28 and p.14, lines 9-14).

Minor comments:

- In most graphs the expression values or log2FC are shown separately for IFN β and IFN γ , however in the heatmaps (Fig 1d, S1d) the IFN β and IFN γ results are intercalated keeping them side-by-side for each time point, which makes them more difficult to interpret.

Reply RC: We are in a quandary about the design of the figure. On the one hand our goal is to visualize gene clusters with distinct behaviors for each IFN type. For this purpose, it would be advantageous to separate the IFN types. On the other hand, we aim at showing similarities and differences between genes induced by each IFN type, for this purpose it is better to maintain the current sample order. While understanding the referee's point, we prefer to keep the figure as it is, because the suggested change will not increase its overall clarity.

- Fig 1e. The color scales on the GO enrichment graphs are misleading since they use the same blue-to-red gradient for adj p-values ranging from 10⁻²⁵ to 10⁻⁴⁹ and 0.008 to 0.016, which could be considered non significant.

Reply RC and EMBO J: We agree that this is confusing. It results from automated assignments of the color gradients by the software. Unfortunately, **we have not found a way to change this.**

- The inconsistency in the title referring to IFN β as Type 1 but using IFN γ instead of Type 2 nomenclature, perhaps consistency is best.

Reply and Revision RC: We agree about the importance of consistency but find ourselves in yet another quandary. While the use of 'type I IFN' is clearly indicated and widely used as a collective name for this group of cytokines, the use of 'type II IFN' for IFN γ is rare because it is the only member of this type. Hence, we decided for sticking with convention at the expense of a bit of consistency. We agree about the title, though, and have changed type I IFN to IFN β .

- The incomplete schema in Figure 1a, which only focuses on PRO-seq and does not include the ATAC-seq element.

Reply: We will add a new figure to visualize the set-up of the ATAC seq experiments and their intersection with the Pro-seq data.

Revision EMBO J: Fig. 5A, shows the work flow of the experiment as requested by the referee.

- Figure 6d includes a color scale of -1 to +3, but it is unclear what these values represent and how they were calculated per interactor. The figure legend should be revised to clarify this information.

Revision RC: We added information (log₂FC with regard to the NLS control) to the legend of fig. 6d.

- The clearer labeling of Figure 5a and 5b.

Reply EMBO J: Please refer to our reply to point 8.

- The statement that "IFN-I are the more important mediators of antiviral immunity" is not entirely accurate and may be an oversimplification, as there are certainly articles which suggest a larger role for type II IFN elements than type I (ref: Yamane D et al., 2019 Nature microbiology). While yes, IFN-I plays a critical role in the innate immune response to viral infections, IFN γ also has antiviral activity and is involved in the adaptive immune response to viral infections, and in some instances to a larger extent than IFN I.

Reply and Revision RC: The Yamane et al study (now mentioned on p 10, lines 22-25 and referenced) agrees with our findings because it shows that IRF1 contributes to the basal expression of an ISRE-driven ISG subset. Our statement about the predominant role of type I IFN versus IFN γ refers to genetic data in both humans (mainly Casanova's work including effects of autoantibodies against type I IFN, see also the paper about human STAT2 deficiency in the June 15th issue of the JCI, <https://doi.org/10.1172/JCI168321>) and mice (hundreds of papers) showing that disruption of type I IFN synthesis or response causes profound effects of antiviral immunity (i.e. resulting susceptibilities are first and foremost to viral pathogens) whereas susceptibilities as a consequence of disrupting the IFN γ pathway are first and foremost to intracellular nonviral pathogens such a mycobacteria. In fact, the term mendelian susceptibility to mycobacterial disease (MSMD) was coined by Casanova and colleagues to describe a variety of human mutations that include those of the IFN γ , but not the type I IFN pathway.

Maybe more importantly, the Rosain et al. paper mentioned by the referee which appeared in 'Cell' while our study was under review, shows that human IRF1 mutations also fall into the

MSMD category (new ref. 65). In contrast, the authors did not observe diminished antiviral immunity. This emphasizes the main conclusions of our study about the relevance of IRF1 for macrophage activation. We discuss this paper on p 14. lines 9-14.

Obviously, this does not exclude a role of type I IFN in nonviral infection or of IFN γ in viral infection, in fact much of our own work has been dedicated to a role of type I IFN in infections with *L. monocytogenes*. Nevertheless, we think that in a generic statement about the difference between type I IFN and IFN γ it is correct to label the former as predominantly antiviral and the latter predominantly as a macrophage activating factor against nonviral, intracellular pathogens.

- The authors claim that a significant portion of ISG promoters is associated with ISGF3 upon IFN γ receptor engagement and that the transcriptomes of macrophages treated briefly with IFN β or IFN γ exhibit remarkable similarity and sensitivity to Irf9 deletion. However, I am uncertain about the extent of consensus on this claim.

Reply RC: The data were surprising but supported by ChIP-seq and RNA-seq in wt and IRF9 ko macrophages (ref 10). Data in a follow-up study (ref. 11) and in this manuscript support our original conclusion by demonstrating the impact of the IRF9 ko on IFN γ responses. Importantly, we don't claim this is true in all cell types, it may well depend on STAT/IRF9 expression levels and tonic IFN signaling.

- Fig 1e, S1c. Graphs having circles of varying sizes in function of a value are named "bubble plots" and not "dot plots".

Revision RC: We changed dot plot to bubble plot in legend to figure S1c.

- Fig S1b, S3b. The PRO-seq was generated in triplicates, hence these graphs should include the Log2FC for the individual data points.

Reply and Revision RC: The Log2FC from DESeq2 were calculated from the triplicates, the software does not compute Log2FC from individual replicates. We mention the p-values for the Log2FC to show the degree of consistency (figure legends).

- Fig S3c legend. It is mentioned "Graph represents RT-qPCR of genomic Mx2". RT-qPCR usually stands for reverse transcription quantitative PCR, hence we suggest to change to "ChIP-qPCR" or qPCR. Confusingly, in the literature the term "RT-PCR" is used for real-time PCR and "qPCR" for quantitative PCR. Also, the authors should be specific about the "genomic" region targeted; the graphs mention "promoter", hence it would be appropriate to use the same designation in the legend.

Revision RC: We changed RT-PCR to qPCR throughout the manuscript. Moreover, we specifically refer to 'promoter region' as the amplified DNA.

- Fig S3e. The y-axis names are missing.

Revision RC: The y axis in the figure received its proper label.

- In the genomic snapshot shown, only bars or fading triangles are shown in place of the gene body. The authors should provide an accurate gene structure; i.e., exons and introns.

Reply RC: We will try to include the exon-intron structure wherever the size of the figure allows this.

Revision EMBO J: Exon-Intron structures are provided as requested by the referee (Figs 2B, 3D, 4A, C, 5D).

- Raw cells are sometimes spelled as "Raw" and other times as "RAW". Please choose one for consistency.

Revision RC: This inconsistency has been corrected

- In p.10 I.20, the figure number is missing.

Revision RC: We corrected this mistake.

Additional Information added to the EMBO J manuscript PRIDE accession number.

Project Name: Identifying interactors of IRF1 and STAT1 during type I and type II interferon treatment using proximity-dependent labeling

Project accession: PXD040337

Project DOI: Not applicable

Reviewer account details:

Username: reviewer_pxd040337@ebi.ac.uk

Password: w1V3828B

Reviewer links for the associated genomics data sets SRA BioProject

PROseq

PRJNA931836

ATACseq

PRJNA937204

Chromatin accessibility during prolonged interferon treatment

<https://dataview.ncbi.nlm.nih.gov/object/PRJNA937204?reviewer=gphu8g3oopede8m8esg4nm8g60>

ChIP-seq:

PRJNA947574

IRF9 binding in BMDMs during early versus later stages of interferon stimulation

<https://dataview.ncbi.nlm.nih.gov/object/PRJNA947574?reviewer=86kmgoo6ackdfe6fum0t3372i2>

Dear Dr. Decker,

Thank you for submitting a revised version of your manuscript. Your study has now been seen again by the referee, who finds that his previous concerns have been addressed and now recommends publication of the manuscript. There remain only a few mainly editorial points that have to be addressed before I can extend formal acceptance of the manuscript:

1. Please submit up to five keywords.
2. FUNDING INFO: "Funding" should be included in the "Acknowledgement" section.
3. REFERENCE FORMAT: numbered, but should be alphabetical, 1 author + et al., but should be 10 authors + et al.
4. DATA AVAILABILITY SECTION: in, the section should be renamed to "Data Availability"
5. COI: title needs renaming to "DISCLOSURE AND COMPETING INTERESTS STATEMENT"
6. AC/CRedit: section needs to be removed.
7. APPENDIX 1 FILE WITH ToC: page numbers missing in ToC; figure legends should be placed under the corresponding figures and above the tables.
8. Please upload the SD checklist (we have uploaded a blank SD checklist for completion)
9. For EV and/or appendix figures, please ZIP together all source data, and please double-check the figure labels in SD files for EV and Appendix figures as they don't match.
10. Please resize the synopsis image (it should be 550x300-600 pixels large (width x height, jpeg or png format)
11. and also add a 'Synopsis' which consists of A) a short (1-2 sentences) summary of the findings and their significance, B) 3-4 bullet points highlighting key results.
12. Please recheck the section order which should be: title page with complete author information, abstract, keywords, introduction, results, discussion, materials & methods, data availability section, acknowledgements, disclosure and competing interests statement, references, main figure legends, tables, expanded figure legends.
13. - Author email bounced:
 - Katrin Fischer - katrin.fischer@univie.ac.at
 - Olga Babadei - olga.babadaei@univie.ac.at
 - Ekaterini Platanitis - ekaterini.platanitis@univie.ac.at

With best regards,

Cornelius Schneider

Cornelius Schneider, PhD
Editor
The EMBO Journal
c.schneider@embojournal.org

We realize that it is difficult to revise to a specific deadline. In the interest of protecting the conceptual advance provided by the work, we recommend a revision within 3 months (5th Jun 2024). Please discuss the revision progress ahead of this time with the editor if you require more time to complete the revisions. Use the link below to submit your revision:

Referee #1:

My comments were addressed by the authors to a satisfactory level.

All editorial and formatting issues were resolved by the authors.

Dear Prof. Decker,

I am pleased to inform you that your manuscript has been accepted for publication in the EMBO Journal.

Yours sincerely,

Cornelius Schneider, PhD
Editor
The EMBO Journal
c.schneider@embojournal.org
